# Posterior scleral birefringence measured by triple-input polarization-sensitive imaging as a biomarker of myopia progression

Xinyu Liu [1,2,3], Liqin Jiang[1,2], Mengyuan Ke [1,4], Ian A. Sigal[5,6], Jacqueline Chua[1,2,3], Quan V. Hoang[1,2,7,8], Audrey WI. Chia[1,2], Raymond P. Najjar [1,2,7], Bingyao Tan[1,3,9], Jocelyn Cheong [1,2], Valentina Bellemo[3,10], Rachel S. Chong[1,2], Michaël J. A. Girard[1,2,11], Marcus Ang[1,2], Mengyang Liu[1,4], Gerhard Garhöfer[12], Veluchamy A. Barathi[2,7,13], Seang-Mei Saw[1,2,14], Martin Villiger [15] & Leopold Schmetterer [1,2,3,4,9,10,11,12] ✉

In myopic eyes, pathological remodelling of collagen in the posterior sclera has mostly been observed ex vivo. Here we report the development of triple-input polarization-sensitive optical coherence tomography (OCT) for measuring posterior scleral birefringence. In guinea pigs and humans, the technique offers superior imaging sensitivities and accuracies than dual-input polarization-sensitive OCT. In 8-week-long studies with young guinea pigs, scleral birefringence was positively correlated with spherical equivalent refractive errors and predicted the onset of myopia. In a cross-sectional study involving adult individuals, scleral birefringence was associated with myopia status and negatively correlated with refractive errors. Triple-input polarization-sensitive OCT may help establish posterior scleral birefringence as a non-invasive biomarker for assessing the progression of myopia.

Myopia (near-sightedness) is a prevalent vision disorder that can be corrected by eyeglasses, contact lenses or refractive surgery. However, unmitigated progression to high myopia exposes patients to an increased risk of developing vision-threatening complications[1,2]. Recent studies have reported that 10–30% of patients with high myopia develop associated pathological complications later in life[3,4], including myopic maculopathy and optic neuropathy, which lead to irreversible visual impairment[5,6]. Clinical interventions for retarding the progression of early-stage myopia and rescuing eyes with pathological complications are available[7,8]. However, reliable biomarkers guiding the timing of treatment are lacking. Specifically, for early-stage myopia, topical atropine has been proved effective in controlling myopia progression[9], yet its adverse effects preclude universal application to all patients[10,11]. Currently, treatment decisions are based on documented myopia

[1]Singapore Eye Research Institute, Singapore National Eye Centre, Singapore, Singapore. [2]Academic Clinical Program, Duke-NUS Medical School, Singapore, Singapore. [3]SERI-NTU Advanced Ocular Engineering (STANCE) programme, Singapore, Singapore. [4]Center for Medical Physics and Biomedical Engineering, Medical University of Vienna, Vienna, Austria. [5]Department of Bioengineering, University of Pittsburgh, Pittsburgh, PA, USA. [6]Department of Ophthalmology, University of Pittsburgh, Pittsburgh, PA, USA. [7]Department of Ophthalmology, Yong Loo Lin School of Medicine National University of Singapore, Singapore, Singapore. [8]Department of Ophthalmology, Columbia University, New York, NY, USA. [9]School of Chemistry, Chemical Engineering and Biotechnology, Nanyang Technological University, Singapore, Singapore. [10]Lee Kong Chian School of Medicine, Nanyang Technological University, Singapore, Singapore. [11]Institute of Molecular and Clinical Ophthalmology, Basel, Switzerland. [12]Department of Clinical Pharmacology, Medical University of Vienna, Vienna, Austria. [13]Translational Pre-Clinical Model Platform, Singapore Eye Research Institute, Singapore, Singapore. [14]Saw Swee Hock School of Public Health, ,National University of Singapore, National University Health System, Singapore, Singapore. [15]Wellman Center for Photomedicine, Harvard Medical School and Massachusetts General Hospital, Boston, MA, USA. ✉e-mail: leopold.schmetterer@ntu.edu.sg

progression[12], that is, the baseline and deterioration of the spherical equivalent refractive error (SE) in the past year. However, large fluctuations in SE during myopia development in childhood and the scarcity of documented records pose practical issues in decision making[13]. For end-stage myopia, posterior scleral reinforcement (PSR) surgery, including macular buckling, is a clinically available therapy to strengthen the posterior sclera and arrest the continued elongation of the eye[14,15]. But standards for whether and when to perform PSR surgeries are controversial and inconclusive[15,16]. To guide treatment decision-making, there is a compelling need for biomarkers that reliably predict myopia progression and indicate early pathological changes in myopic eyes.

Owing to its pivotal role in defining eye shape, the sclera has been extensively studied in animal models and humans with myopia or pathologic myopia[17–20]. The sclera is a dense, collagen-rich and mechanically strong tissue that coats the eye and protects its internal structures[21]. During the development and progression of myopia, the posterior segment of the sclera undergoes a remodelling process that includes thinning[22], weakening[23] and enlargement in surface area[24], resulting in an excessive axial elongation of the eye that impairs its optical function. Furthermore, extensive scleral remodelling may predispose patients to staphyloma, an irregular outpouching of the posterior eyewall, which is a defining characteristic of pathologic myopia. Staphyloma may create shear forces across the retina and is one of the main pathophysiological factors of myopia-associated vision-threatening complications[25]. At present, staphyloma is diagnosed through the observation of an irregular eye shape using ultrasonography or wide-field optical coherence tomography (OCT)[6]. However, eye-shape deformation may be a secondary result of extensive scleral remodelling, which, by this point, may already have caused irreversible retinal damage[26]. From the early stages to the late stages of myopia, scleral collagen is constantly remodelling at the microscopic level; these changes include a decrease in collagen fibre diameter[27,28], a shift towards disordered architecture with a reduction in the number of interwoven fibres[22,29] and alterations in fibre direction[30]. Imaging techniques including polarization light microscopy (PLM)[31,32] and transmission electron microscopy (TEM) have been essential in identifying these changes associated with scleral remodelling but are suitable only for ex vivo samples. Currently, no tool is commercially available to inspect posterior scleral collagen in vivo. On the basis of knowledge on scleral remodelling in myopic eyes, we envision that a tool enabling in vivo imaging of collagen in the posterior sclera could enable evaluation of the status of myopia, prediction of its progression, identification of scleral weakening and prospective evaluation of the risk of pathological changes.

Polarization-sensitive OCT (PS-OCT) derives image contrast from tissue birefringence[33,34] and has been demonstrated to be a promising tool for scleral collagen imaging in small animals in vivo[35,36]. Collagen fibres exhibit a combination of form and intrinsic birefringence and confer birefringence to scleral tissue, whereby light polarized along or orthogonal to the fibre direction experiences slightly different refractive indices. Unlike the anterior sclera that can be directly accessed[37], imaging the posterior sclera in humans is far more challenging and requires high detection sensitivity and accuracy, because the probing light is attenuated[38] and the input polarization state is altered when passing through the eye[39]. Recently, posterior sclera imaging using PS-OCT has been shown in seven healthy volunteers and the architecture of scleral collagen fibres in normal eyes has been revealed[40]. However, the clinical value of scleral collagen imaging is still unclear and further investigations of scleral collagen imaging in preclinical and clinical settings would benefit from further improvements in detection sensitivity and system robustness.

In this study, we investigated posterior scleral birefringence (PSB) in an animal model and in patients with myopia or pathologic myopia using triple-input PS-OCT (TRIPS-OCT), a modulation and reconstruction strategy for PS-OCT that increases imaging sensitivity, accuracy and system robustness. Current electro-optic modulator (EOM)-based

PS-OCT instruments using sequential dual-input polarization states[41,42] assume that the measurements contain only pure retardance and that the impact of sample diattenuation is negligible[43]. Depth-encoding PS-OCT[44–46] can measure diattenuation but reduces the imaging range due to multiplexing of the images of two polarization input states along depth. TRIPS-OCT measures diattenuation and corrects for depolarization while maintaining the simplicity of dual-input systems. We demonstrated that TRIPS-OCT improved birefringence sensitivity and accuracy of optic axis measurement compared with dual-input PS-OCT. Moreover, using histological sections of the posterior sclera of pig and guinea pig eyes, we validated in vivo TRIPS-OCT birefringence imaging with PLM and TEM.

To examine PSB as a biomarker for myopia, first, we used a guinea pig myopia model[47] (42 eyes) to longitudinally evaluate the correlation between PSB and development of refractive errors in animals of 2–8 weeks age. PSB measured at 2 weeks of age was an effective predictive biomarker for the onset of myopia at the ages of 4 and 8 weeks, better than baseline SE, which has been reported as the best single predictor for myopia onset[48,49]. Next, in our human cross-sectional study, we found a strong, albeit negative, correlation between PSB and myopia status within eyes with emmetropia (normal vision) and low myopia (69 eyes, $-6D < SE \le 3D$). To the best of our knowledge, there have been no previous clinical studies focusing on PSB in patients with myopia. Moreover, in patients with pathologic myopia, we observed a spatial association of PSB with staphyloma. We next determined that PSB was a better classifier than axial length to differentiate eyes with pathologic myopia (15 eyes) from those with high myopia (16 eyes). In eyes with high myopia, we found increased PSB to be associated with the presence of peripapillary atrophy (PPA)[50,51] and to possibly indicate an increased risk of progression to a more advanced stage of myopia. Overall, in this study, we demonstrated the potential of PSB, measured with TRIPS-OCT, as a predictive biomarker for myopia, from childhood myopia development to late-life complications.

## Results

### Sensitivity and accuracy of TRIPS-OCT

We developed TRIPS-OCT (Extended Data Fig. 1) to address the challenge of performing reliable birefringence measurements in the clinical setting. To demonstrate the improved birefringence sensitivity of TRIPS-OCT compared with the dual-input reconstruction method, we imaged a guinea pig retina in vivo (Fig. 1a) and reconstructed the local birefringence images using the dual-input method and the proposed method. In this comparison, we ensured that the sampling time of the signals used by the two methods were identical (Extended Data Fig. 2a). As the inner retina of the guinea pig exhibits low birefringence, distributions of its birefringence (Fig. 1b) can approximate the characteristics of the birefringence background noise. The standard deviation of the noise, or noise floor, of TRIPS-OCT was 48% lower than that of the conventional dual-input method (Fig. 1b and Extended Data Fig. 2b,c).

We observed that the dual-input reconstruction method suffers from edge artefacts that are associated with variations in the sample scattering profile (Extended Data Fig. 3a,b). The edge artefact (Supplementary Discussion 1) is a dominant source of birefringence noise that is induced by a shift in the point-spread-functions (PSF) due to the polarization mode dispersion of preceding tissue layers, including the cornea. Because this shift leads to apparent diattenuation, the resulting artefacts are markedly suppressed by correctly accounting for diattenuation in the Mueller matrices of the sample. TRIPS-OCT isolates the sample retardance from the Mueller matrices, properly separating the effects of polarization-dependent scattering, sample diattenuation and apparent diattenuation, and thus is almost free from edge artefacts (Extended Data Fig. 3c).

To test whether compensation for corneal retardance and diattenuation improves the accuracy of optic axis measurements, we scanned the Henle's fibre layer (HFL) in the retina of a healthy volunteer

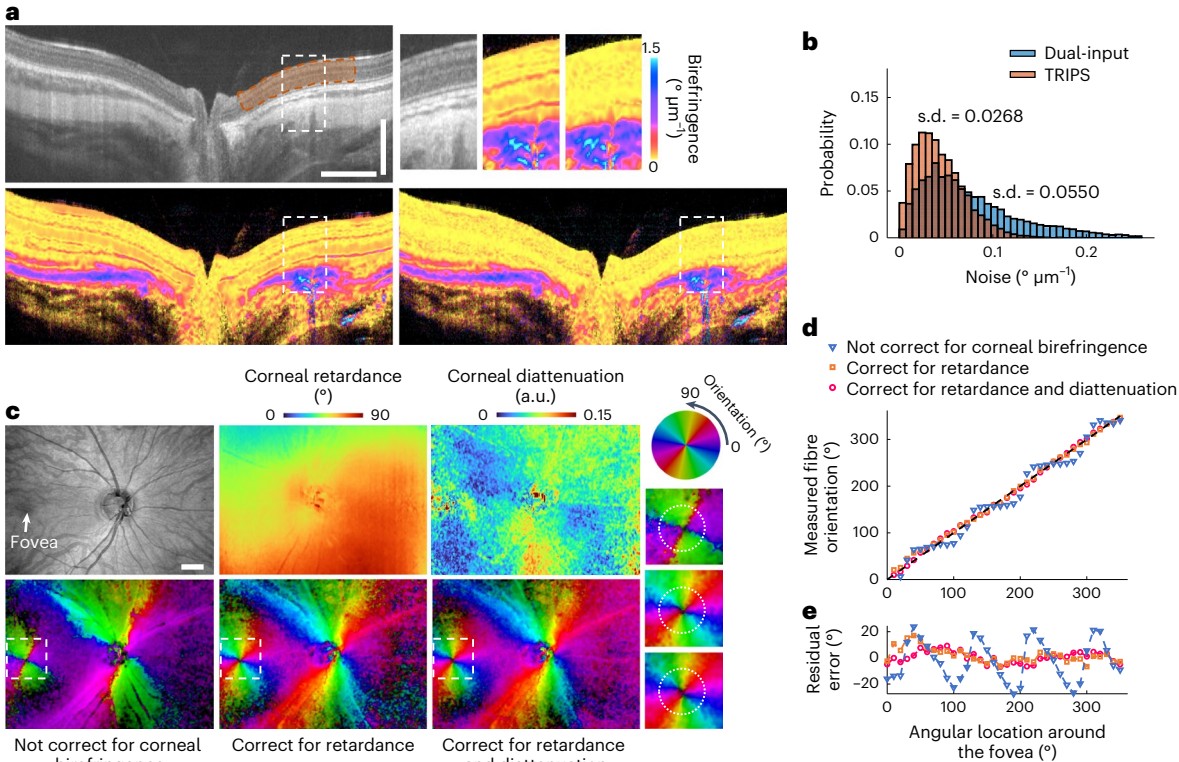

**Fig. 1 | Technical advantages of TRIPS-OCT. a**, TRIPS and dual-input reconstruction methods on guinea pig retina in vivo. The intensity image (upper left) and corresponding birefringence images are reconstructed from the dual-input method (lower left) and the proposed triple-input method (lower right). White boxes indicate the location of the zoomed-in views (upper right). The reddish stripes in the inner retina that are present in the dual-input reconstruction are induced by edge artefacts (Extended Data Fig. 3a,b). Of note, the artefacts disappear in the TRIPS reconstruction. The orange area indicates a region in the inner retina that is used to characterize the birefringence noise. **b**, Histograms of birefringence noise calculated from the region in **a** indicated by the orange area (pixel number $n$ = 5,117 from 1 cross-sectional image). **c**, Two-dimensional correction of corneal retardance and diattenuation. The en face intensity image (upper left) from a healthy human subject (32-yr-old male,

OD, Asian) is rendered from a volume scan of the posterior eye. The corresponding corneal retardance (upper middle) and diattenuation (upper right) maps are extracted from the retinal surface and the optic axis images are reconstructed without (lower left) and with the correction for corneal retardance (lower middle), and for both retardance and diattenuation (lower right). The position of the fovea is indicated by a white arrow. The magnitude of diattenuation $Dia$ is defined as the relative difference between the maximum $p_1^2$ and minimum $p_2^2$ attenuation coefficients, where $Dia = (p_1^2 - p_2^2)/(p_1^2 + p_2^2)$. Zoomed-in images (right) indicated by white boxes highlight the HFL. **d**, Measured in-plane HFL fibre orientation against angular location on a circle (indicated by white dotted circles in **c**) centred on the fovea with an eccentricity of 2°. **e**, Optic axis measurement error without/with corneal correction. Scale bars: **a**, vertical: 300 μm, horizontal: 1 mm; **c**, 1 mm.

(32-yr-old male, oculus dexter (OD), Asian) (Fig. 1c). The in-plane orientation of the HFL is approximately radially distributed around the fovea[52]. We used the orientation of the HFL to assess the accuracy of the optic axis measurements. We extracted the two-dimensional corneal retardance and diattenuation maps from the surface of the retina (Supplementary Method 6). In the optic axis images of the HFL, the measured orientations were plotted against the angular location on a circle centred on the foveal pit (Fig. 1d). An offset existed in the measured optic axis orientation due to the unmeasurable circular birefringence of the system and the anterior segment of the eye. The offset can be estimated by minimizing the difference between the measured optic axis orientation and the assumed orientation of the HFL, that is, radial around the fovea. Without corneal correction, the measured fibre orientation deviated markedly from the assumed radial profile. After applying the correction for corneal retardance and diattenuation, the mean error of the measurement, characterized by the residual difference between the measured optic axis orientation and the assumed orientation (Fig. 1e), was 35% lower than that of the uncorrected results.

## Scleral collagen architecture under TRIPS-OCT

To demonstrate imaging of the posterior scleral birefringence (PSB), we scanned the eye of a healthy volunteer (28-yr-old female,

oculus sinister (OS), −6.75 dioptre ($D$), Caucasian) with TRIPS-OCT in a three-dimensional volume and reconstructed the birefringence and optic axis images (Supplementary Video 1). The cross-sectional intensity image (Fig. 2a) obscured the complex scleral collagen fibre structure, which was clearly visualized in the cross-sectional optic axis image (Fig. 2b). Consistent with previous reports[40,53], en face optic axis images (Fig. 2c) of the peripapillary sclera showed a two-layer architecture, where the inner and outer layers were dominated by radial and circumferential fibres, respectively.

We used the eye of a pig, which was similar in size to a human eye, to further validate TRIPS-OCT against polarized light microscopy (PLM), an established tool for birefringence imaging. We first imaged the pig's eye in vivo with TRIPS-OCT and collected the eye for PLM histological analysis immediately after imaging and killing the animal. The TRIPS-OCT volume scan was rotated and resliced to register to the PLM images. The radial and circumferential fibre distributions in the inner and outer sclera layers indicated close agreement between TRIPS-OCT (Fig. 2d,f) and PLM (Fig. 2e,g). The good co-location between the two methods in the locations of fine structures, including the annular collagen around the optic nerve head (ONH) and the tree-like stalk (Fig. 2h) of the pig lamina cribrosa[53], further validated the accurate registration between these in vivo and ex vivo imaging modalities.

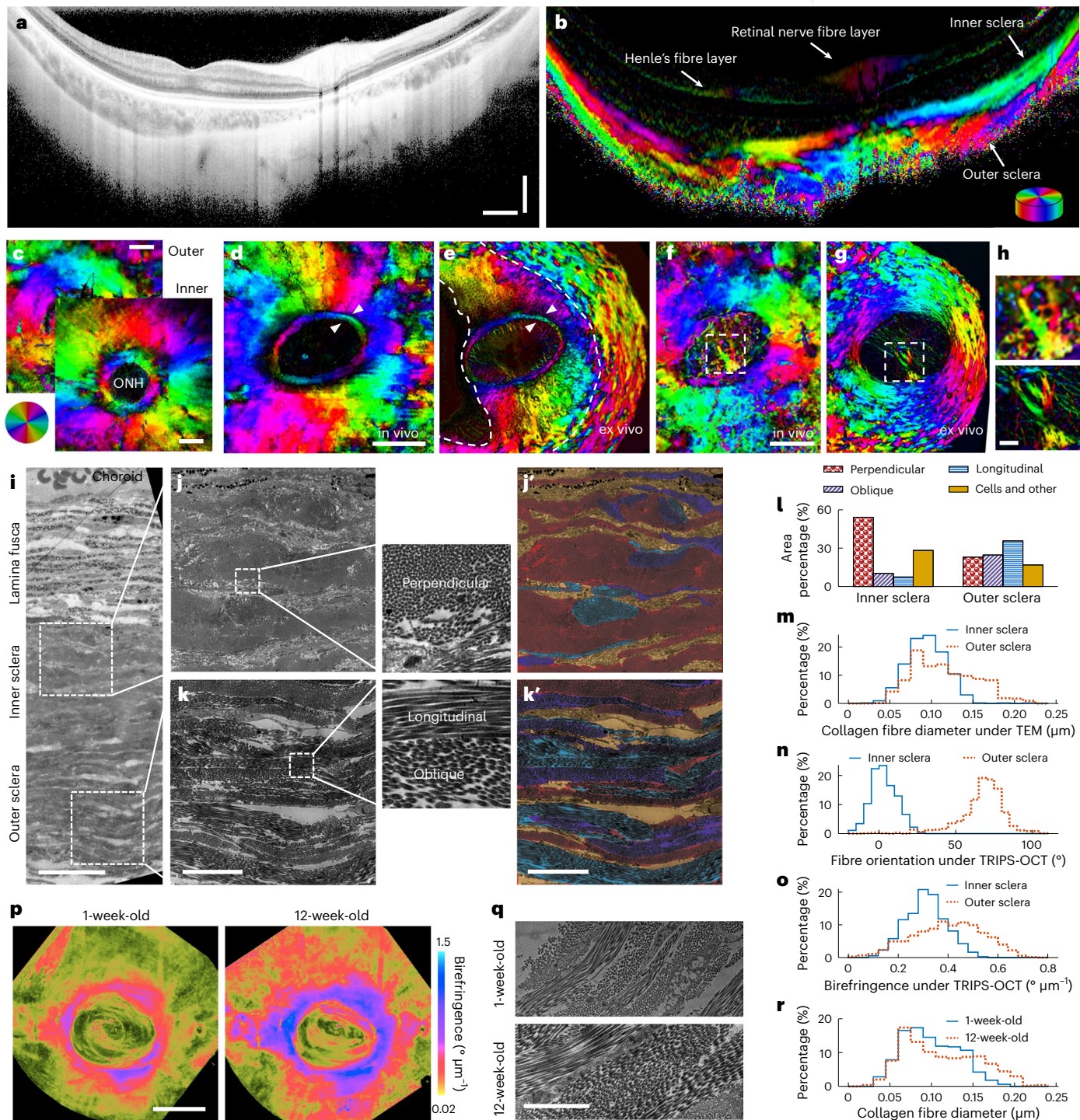

**Fig. 2 | Validation and interpretation of TRIPS-OCT images. a–c,** TRIPS-OCT scan on a healthy participant (28-yr-old female, −6.75 D, Caucasian) showing a representative cross-sectional intensity image (**a**) and the corresponding optic axis image (**b**), and the en face optic axis images (**c**) centred at the ONH at two depths. **d–g,** Pig en face scleral optic axis images at two depths under TRIPS-OCT in vivo (**d,f**) and registered images under PLM ex vivo (**e,g**). White arrows in **d** and **e** indicate the annular collagen around the ONH. White boxes in **f** and **g** indicate the tree-like stalk of the pig lamina cribrosa. Eyeball deformation during tissue fixation creates discrepancy areas in **d** and **e** indicated by white dashed lines. **h**, Magnified views of dashed boxes in **f** and **g**. **i–k**, Guinea pig TEM images of the sclera (**i**), the inner (**j**) and outer (**k**) sclera and zoomed-in views to observe the fibres. Perpendicular, longitudinal and oblique fibre orientations are colour-coded in **j′** and **k′**. **l,m**, Percentages of the fibre orientations (**l**) and diameter (**m**) measured from TEM images. Each histogram is calculated from $n = 600$ individual fibres from 1 sclera sample. **n,o,** Distributions of fibre orientation (**n**) and birefringence (**o**) measured from TRIPS-OCT in vivo (Supplementary Data Fig. 1) at roughly the same location as TEM. Each histogram is calculated from $n = 500$ pixels in a cuboidal region from 1 volume scan. **p**, Guinea pig en face images in one eye at the ages of 1 week and 12 weeks. **q**, TEM images at the ages of 1 week and 12 weeks. **r**, Outer scleral collagen fibre diameter distributions measured from TEM images from 2 guinea pigs at the ages of 1 week and 12 weeks. Histogram equalization was applied to **a**. Scale bars: **a**, vertical: 300 μm, horizontal: 1 mm; **c,d,f,p**, 1 mm; **h**, 150 μm; **i**, 30 μm; **k,k′**, 10 μm; **q**, 2 μm.

### Birefringence reflects the arrangement and diameter of fibres

To further interpret the birefringence images, we compared TRIPS-OCT with transmission electron microscopy (TEM), a nanometre-scale resolution tool that can clearly visualize collagen bundles and individual fibres. A 16-week-old guinea pig's eye was imaged in vivo via TRIPS-OCT (Supplementary Data Fig. 1) and then collected for TEM analysis after killing the animal. The TEM section was sampled parallel to the temporal–nasal plane in the superior region located 2 mm from the ONH. From the TEM images (Fig. 2i–k) of the inner and outer sclera, the fibre orientations with respect to the sectioning plane were classified as perpendicular, longitudinal and oblique (Fig. 2j',k'). We calculated the percentages of the different fibre orientations (Fig. 2l) and measured the fibre diameter distribution (Fig. 2m). For comparison, in roughly the same location, we evaluated the distributions of optic axis orientation (Fig. 2n) and birefringence (Fig. 2o) measured with TRIPS-OCT in vivo. We qualitatively observed that the measured optic axis orientations corresponded to the average orientation of all fibres within the TRIPS-OCT resolution volume. By inspecting the birefringence images, we found that higher birefringence corresponded to more aligned fibres within the TRIPS-OCT resolution volume, explaining the ring-like pattern around the ONH in the en face birefringence image (Fig. 2p), where most of the fibres were circumferentially arranged around the ONH.

To qualitatively investigate the relationship between birefringence and collagen fibre diameter, we imaged the same eye of one guinea pig in vivo at the ages of 1 week and 12 weeks using TRIPS-OCT and reconstructed the en face birefringence images (Fig. 2p). A significant increase in scleral birefringence was observed in the older animal due to physiological eye growth. We then killed two guinea pigs at the ages of 1 week and 12 weeks for TEM analysis. TEM sectioning (Fig. 2q) was performed at the outer sclera from the same location in both animals and the distributions of scleral fibre diameters (Fig. 2r) were calculated. We observed that in the older animal, the distribution of fibre diameter was skewed to larger values. Overall, in addition to more aligned fibres, a larger average collagen fibre diameter, corresponding to a higher collagen content and thicker lamellae (Supplementary Discussion 2), led to the increased birefringence in TRIPS-OCT images.

### Development of refractive errors in guinea pigs

To explore the correlation between posterior scleral birefringence (PSB) and the development of refractive errors, we used a cohort of guinea pigs ($N = 21$), mixed albino ($N = 17$) and pigmented ($N = 4$) strains, which were previously reported to have spontaneous myopia rates of 70% (albino) and 29% (pigmented)[47]. We imaged the animals weekly with TRIPS-OCT from birth to 8 weeks (Fig. 3a, raw data in Supplementary Figs. 2 and 3). As the optical aberration of the eye at the age of 1 week hindered TRIPS-OCT imaging, some measurements were not available at the age of 1 week. Refraction was measured as spherical equivalent refractive error (SE) with retinoscopy from 1 to 8 weeks. Notably, we observed strong correlations between SE and PSB from the ages of 2 to 8 weeks (Fig. 3b), with the highest Pearson correlation coefficient observed at the age of 4 weeks ($r = 0.78$, $P = 3.6 \times 10^{-5}$).

We also performed a repeatability test of the birefringence measurements (Extended Data Fig. 4) within a cross-sectional imaging experiment of guinea pig eyes in vivo. Excellent measurement repeatability of PSB was demonstrated between repeat measurements at the same imaging angle ($r = 0.999$, 1.96 s.d. = 2.43%, 12 eyes) and between measurements at two different imaging angles ($r = 0.995$, 1.96 s.d. = 6.09%, 12 eyes).

### PSB predicts the onset of myopia in guinea pigs

To evaluate the PSB measured at the age of 2 weeks and myopia onset at the age of 4 weeks, we used the data obtained from the longitudinal model described in the previous subsection. TRIPS-OCT images measured at the age of 2 weeks were assigned to two groups (Fig. 4a,b) on the basis of the SE measurements at the age of 4 weeks by a threshold of $0D$

(myopia group: SE < $0D$, emmetropia and hyperopia group: SE ≥ $0D$). We observed that guinea pig eyes in the myopia group showed significantly lower PSB than those in the emmetropia and hyperopia group ($P = 0.0054$, Fig. 4c).

We hypothesized that PSB could be a predictor for the onset of myopia and compared it to baseline SE, which has been reported as the best single predictor for myopia onset[48,49]. We assessed the correlation between the baseline SE, PSB measured at the age of 2 weeks and the SE at the subsequent ages of 2–8 weeks (Fig. 4d). Notably, from the age of 4 weeks onwards, the baseline SE was less correlated with refraction status than the PSB.

We next used the baseline SE and PSB at the age of 2 weeks to predict myopia onset (defined as SE < $0D$) at the ages of 4 and 8 weeks. Receiver operating characteristic (ROC) curves (Fig. 4e) of prediction outcomes showed that PSB achieved better performance than the baseline SE (week 4: PSB area under the curve (AUC), 0.89; baseline SE AUC, 0.74; week 8: PSB AUC, 0.85; baseline SE AUC, 0.73).

### Birefringence is correlated with myopia status in humans

To investigate whether a correlation between PSB and myopia status exists in humans, 80 participants without pathological ocular conditions were recruited (Extended Data Fig. 5). TRIPS-OCT scanning and measurements of SE and axial length (AL) were performed on both eyes of every participant. Due to the requirement for a sufficient signal from the sclera, we excluded 75 eyes (47%) on the basis of quality criteria (Supplementary Discussion 3), composed of images with suboptimal positioning (50 eyes, 31%) and insufficient signal-to-noise ratio (average scleral SNR < 4.6 dB) from the sclera (25 eyes, 16%). In TRIPS-OCT images (Fig. 5a–h and Supplementary Video 2) of a typical emmetropic eye (SE = $0D$, Fig. 5a,c,e) and a myopic one (SE = −6.75$D$, Fig. 5b,d,f–h), we observed that the myopic eye presented increased PSB, both in the outer peripapillary area and the posterior pole area.

We then investigated the correlation between PSB and myopia status in participants with emmetropia or low degree of myopia. Using SE as a threshold, we grouped the eyes into two groups: the emmetropia or low myopia group (−6$D$ < SE ≤ 3$D$) and the high myopia group (SE ≤ −6$D$). From the 69 eyes of 42 participants (Supplementary Table 1) in the emmetropia or low myopia group, we calculated the scleral birefringence at two different locations on the fundus; these values were the outer peripapillary scleral birefringence (OPSB) and the posterior pole scleral birefringence (PPSB) (Fig. 5e). We assessed the correlation between SE and both OPSB and PPSB (Fig. 5i,j) and found that SE was significantly correlated with PPSB and to a lesser degree with OPSB (PPSB vs SE: $r = −0.60$, $P = 1.2 \times 10^{-4}$, OPSB vs SE: $r = −0.42$, $P = 0.012$). We further assessed the correlation between PPSB and AL (Fig. 5k) to confirm the strong correlation between myopia status and PPSB (PPSB vs AL: $r = 0.54$, $P = 0.001$).

To further investigate the location dependence of PSB and myopia status on the fundus, the outer peripapillary area of the eyes from the emmetropia or low myopia group was divided into 12 segments by polar coordinates (Fig. 5e) and the mean birefringence value within each segment was correlated with the SE and AL (Fig. 5l and Supplementary Data Fig. 4). Notably, the birefringence in the 4 segments close to the fovea exhibited a significant correlation with myopia status (temporal-inferior: −60°, −30°; temporal: 0°; temporal-superior: 30°; $P ≤ 0.01$) and the highest Pearson correlation coefficient was observed in the segment located between the ONH and the fovea (temporal: 0°), which served as the definition of PPSB.

In addition, to evaluate the correlation among other biometrics within the eyes in the emmetropia or low myopia group, we additionally measured the choroidal thickness and the inner sclera thickness from the TRIPS-OCT images. We evaluated the correlation among age, AL, SE, choroidal thickness, inner scleral thickness and PPSB (Fig. 5m; model parameters in Extended Data Table 1, raw data in Supplementary Data Fig. 5). We found that in addition to the known strong correlation

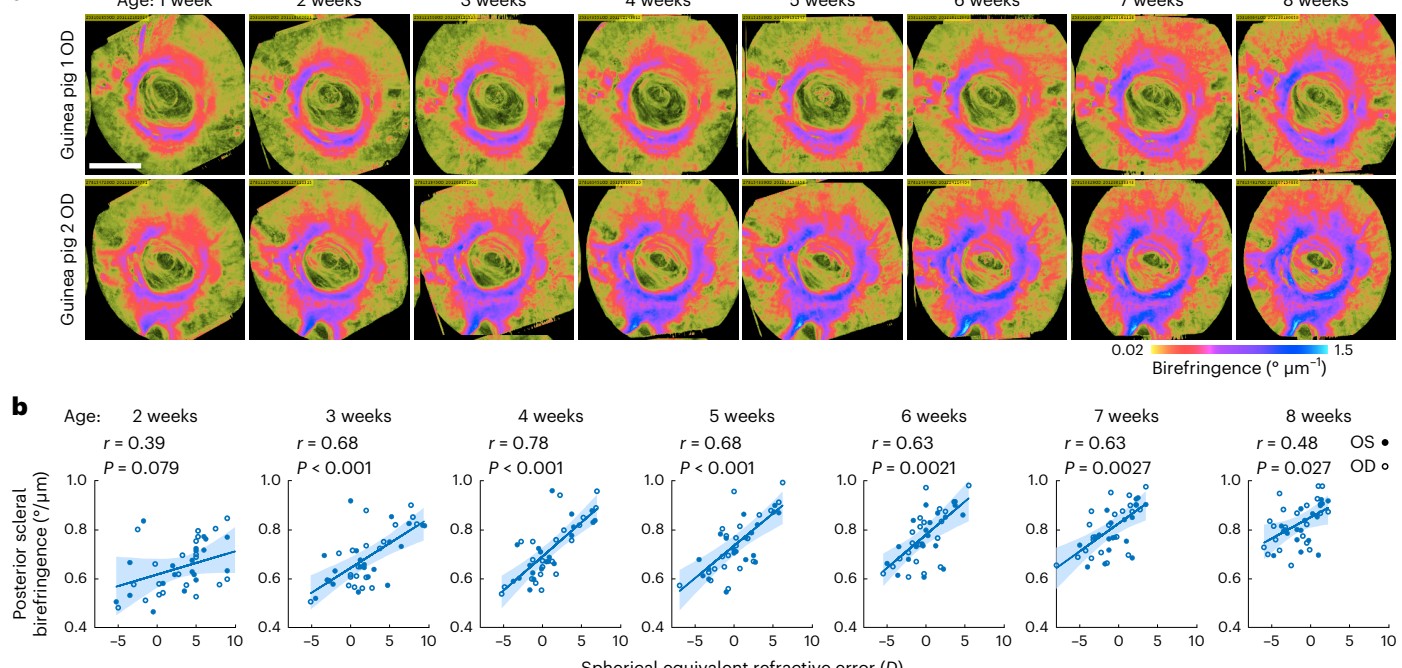

**Fig. 3 | Correlation between the development of refractive errors and posterior scleral birefringence as measured using TRIPS-OCT in a guinea pig model. a**, Representative guinea pig en face scleral birefringence images of two eyes longitudinally measured weekly from 1 week to 8 weeks of age. Scale bar, 1 mm. **b**, Correlation analysis of posterior scleral birefringence measured from the images and SE from 2 weeks to 8 weeks of age. Scatterplots show 42 individual eyes from 21 animals, regression (lines) and 95% confidence intervals (shaded areas). $r$ values were calculated using Pearson correlation. $P$ values were calculated using $F$-test against a constant model. Inter-eye correlation was addressed by cluster bootstrapping.

between SE and AL, the correlation between PPSB and myopia status (SE, AL) was significantly higher than that between other biometrics.

### PSB is associated with pathological changes in humans

We hypothesized that PSB would be an indicator of scleral pathological changes. To test this hypothesis, we recruited 10 patients with both eyes diagnosed with myopia-associated staphyloma. TRIPS-OCT and AL measurements were performed on both eyes, but no SE measurements were performed due to the low function of the eyes. Fifteen eyes (75%) were scanned and included in the pathologic myopia group for further analysis (Supplementary Table 1).

In eyes with pathologic myopia, we observed a spatial association between PSB and staphyloma. From the TRIPS-OCT images (Fig. 6a–c) of a typical eye with staphyloma, we further reconstructed the three-dimensional eye shape (Fig. 6d and Supplementary Video 3) from the volume scan and identified the edges of staphyloma. We observed that increased PSB was spatially correlated with the staphylomatous outpouching regions. To observe the PSB in eyes with various stages of myopia, we compared the eyes from the high myopia group reported in the previous subsection and the eyes in the pathologic group (Fig. 6e). We observed that PPSB markedly increased in eyes with pathologic myopia even when staphyloma edges were not located in the posterior pole area.

To further investigate the correlation between PPSB and myopia status, we combined the eyes from the emmetropia or low myopia group, the high myopia group and the pathologic myopia group and assessed the correlation between PPSB and AL (Fig. 6f). A strong correlation was found between PPSB and AL in these eyes ($r = 0.55$, $P = 3.9 \times 10^{-5}$, 100 eyes), verifying that the increase in PPSB is associated with scleral changes in pathologic myopia.

To further evaluate the potential of PPSB as a marker to differentiate eyes with pathologic changes, we combined the eyes from the high myopia group and the pathologic myopia group and compared PPSB to AL as classifiers to identify pathological eyes (Fig. 6g). The PPSB showed better performance than the AL in terms of the AUC (PPSB AUC: 0.94, 95% CI (0.72, 1); AL AUC: 0.82, 95% CI (0.53, 1)).

To evaluate whether PPSB indicates potential progression of high myopia, we further divided the high myopia group according to the presence of peripapillary atrophy (PPA), which has been reported as a factor associated with progressive myopia[50,51]. We observed that PPSB was significantly higher in eyes with PPA than in those without PPA ($P = 0.011$, Fig. 6h). The PPSB of eyes in the pathologic myopia group was higher than that of eyes in the high myopia group, albeit with statistical significance only for eyes without PPA ($P = 9.1 \times 10^{-5}$ vs $P = 0.18$, Fig. 6h). Consistent with the diagnostic implication of PPA, increased PPSB may prospectively suggest the potential for progression from high myopia to pathologic myopia.

### Discussion

The prevalence of myopia is increasing globally. It has been predicted that myopia will affect almost 5 billion people by 2050. Although only a fraction thereof would develop pathologic myopia, this will amount to approximately 300 million people[1,2]. Patients with pathologic myopia have reduced quality of life due to its economic and societal impact[54]. Myopia is now recognized as an immense future healthcare problem that needs to be addressed today[55]. While genetic, environmental, biochemical and physiological factors have been reported to contribute to the development and progression of myopia[1], the physical elongation of the eye is ultimately related to the remodelling of the posterior sclera[56]. In this study, we measured PSB, which relates to the architecture and diameter of collagen fibres in the posterior sclera, and investigated its potential as a biomarker for predictively evaluating the risk of myopia progression. To enable clinical PSB measurements, we developed TRIPS-OCT which offers benefits for birefringence imaging in terms

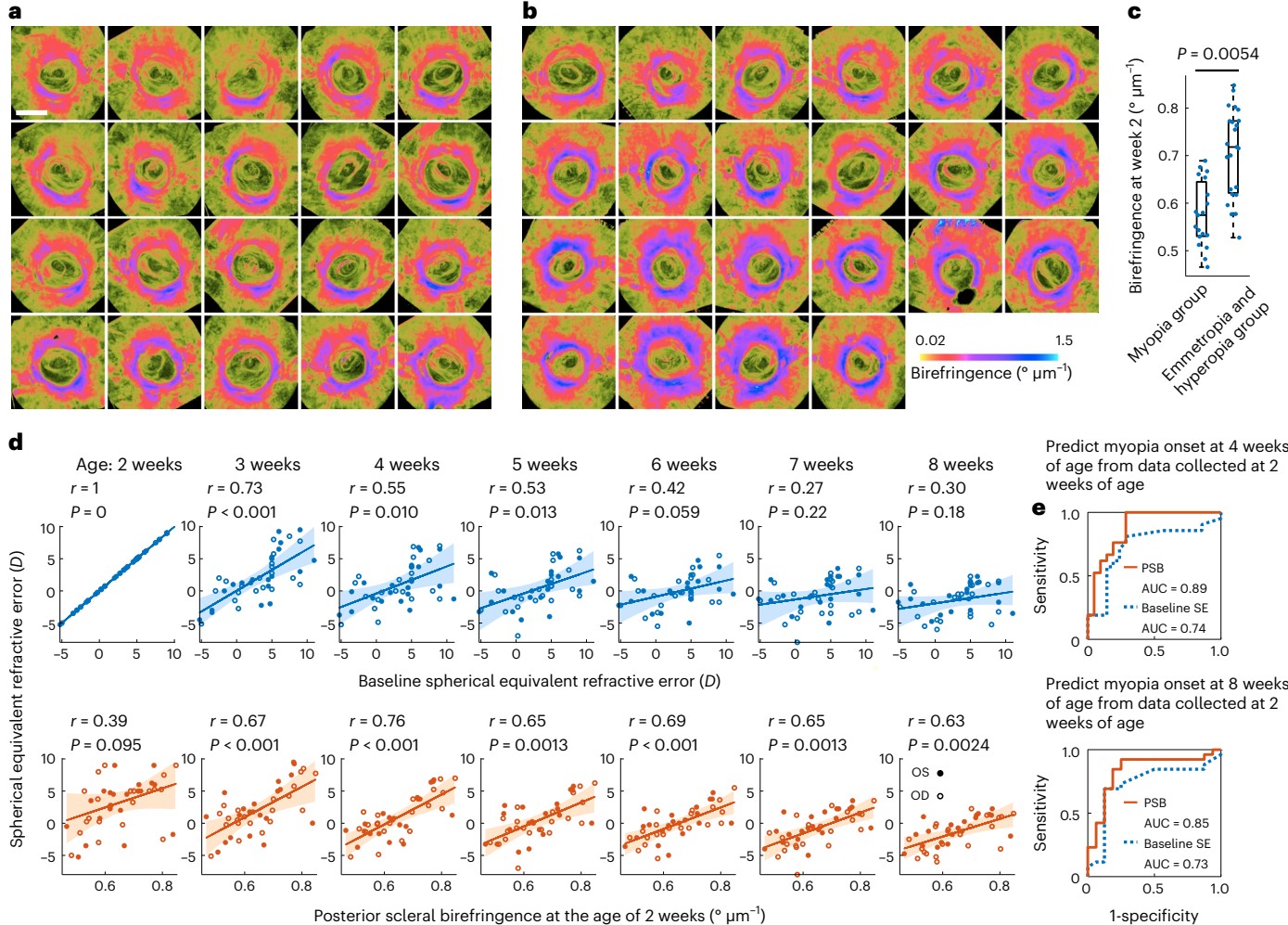

**Fig. 4 | Prediction of myopia onset using TRIPS-OCT in guinea pig model.**
**a,b**, En face scleral birefringence images measured at the age of 2 weeks from the entire cohort of guinea pigs. Images are grouped by myopia outcome at the age of 4 weeks, defined as SE < 0*D*. Group 1 (**a**), myopic eyes. Group 2 (**b**), emmetropic and hyperopic eyes. Scale bar, 1 mm. **c**, PSB values measured from the images in the two groups of guinea pig eyes. Dots represent *n* = 42 eyes from 21 guinea pigs, central line indicates median, box shows interquartile range and whiskers show range. *P* value was calculated using two-sided Wilcoxon rank-sum test with cluster bootstrapping to correct inter-eye correlation. **d**, Correlation between the predictors, baseline SE (upper panel) and PSB at the age of 2 weeks

(lower panel), and the outcome (SE at the ages of 2–8 weeks). Scatterplots show 42 individual eyes from 21 animals, regression (lines) and 95% confidence intervals (shaded areas). *r* values were calculated using Pearson correlation. *P* values were calculated using *F*-test against a constant model. Inter-eye correlation was addressed by cluster bootstrapping. **e**, Predictions of myopia outcomes at the ages of 4 weeks (upper panel) and 8 weeks (lower panel) from the data measured at the age of 2 weeks, using PSB (orange line) and baseline SE (blue line) as predictors. PSB: AUC, 0.89; 95% CI (0.70, 1); SE: AUC, 0.74, 95% CI (0.48, 0.94); Week 8, PSB: AUC, 0.85, 95% CI (0.61, 1); Baseline SE: AUC, 0.73, 95% CI (0.46, 0.95).

of sensitivity, accuracy, robustness and imaging range, compared with previous PS-OCT implementations.

In our guinea pig model, we showed that the development of refractive errors from the ages of 2–8 weeks was correlated with PSB. Myopia onset at the ages of 4 and 8 weeks can be predicted by PSB measured at 2 weeks. Various studies have demonstrated that the establishment of refraction is controlled through the modulation of scleral extracellular matrix growth and remodelling[57]. Specifically, in response to myopiagenic visual signals (that is, defocus), the activity of scleral fibroblasts, chondrocytes and myofibroblasts (regulated by gene expression or biochemical factors) is altered, sequentially altering extracellular matrix synthesis and organization[19,58]. The PSB measured at the age of 2 weeks is related to the average collagen fibre diameter in the posterior sclera. As a possible explanation of our results, PSB is indicative of the level of scleral collagen synthesis[19] and relatedly, the proliferation of myofibroblasts[58]. Therefore, the rapid increase in PSB in young animals during eye growth may indicate the establishment

of an effective regulation from vision to scleral development[18], which reduces the animals' susceptibility to myopia.

To control early-stage myopia, increasing outdoor activities can be unequivocally encouraged for all children. However, more targeted interventions, such as orthokeratology and atropine eye drops, have to be applied selectively on the basis of the individual risk of developing high myopia[8,13,59]. The prediction of myopia progression is particularly important for guiding treatment decisions. Various studies have reported that baseline SE is the best single predictor for myopia onset[48,49], outperforming other risk factors, including outdoor time and parental myopia. In contrast, in our guinea pig model, we demonstrated that PSB performed better than baseline SE with higher AUC values. Our results imply that PSB may serve as a predictive biomarker for predicting myopia progression in children; however, further clinical studies focusing on children participants are needed to validate the association between PSB and childhood myopia. Considering active research into various myopia control strategies, such as atropine with different

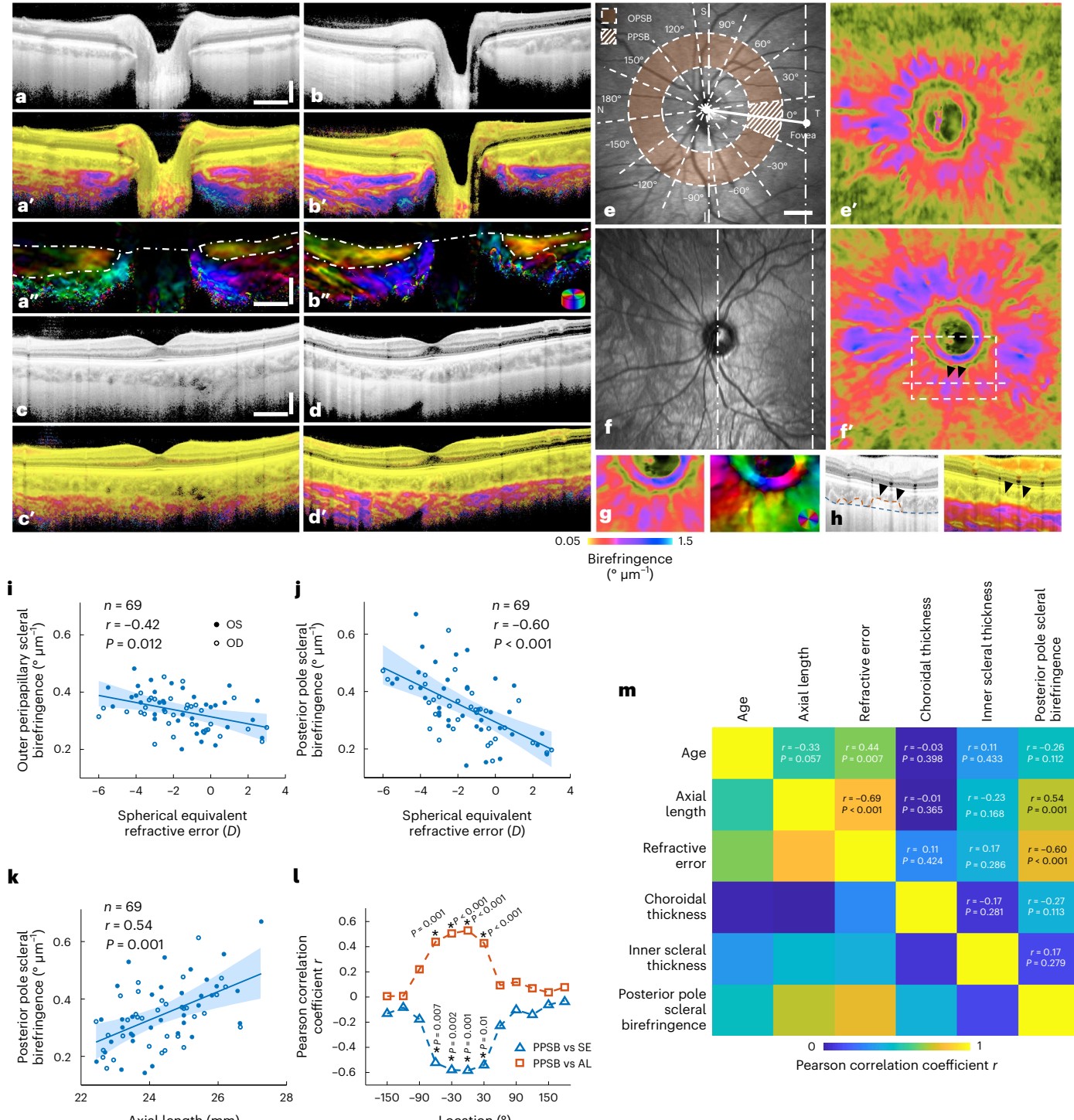

**Fig. 5 | Scleral birefringence in non-pathologic patients with myopia.**
**a–f**, Representative TRIPS-OCT images of an emmetropic eye (40-yr-old female, OS, 0 D, Caucasian) (**a,a′,a″,c,c′,e,e′**) and a myopic eye (28-yr-old female, OS, −6.75 D, Caucasian) (**b,b′,b″,d,d′,f,f′**). Cross-sectional images containing the ONH (**a,b**), fovea (**c,d**) and en face images (**e,f**) are shown in intensity (**a–d,e,f**), birefringence (**a′–d′,e′,f′**) and optic axis (**a″,b″**) contrasts. Dotted lines in **e** and **f** indicate locations of cross-sectional images. **g**, Zoomed-in view of the box in **f′** and corresponding optic axis image showing the circumferential ring-like structure. **h**, Cross-sectional intensity (left) and birefringence (right) images indicated by the white dashed line in **f′** showing the uneven choroidal–scleral interface (orange dashed line) resulting in the petaloid birefringence pattern (black arrowheads in **f′** and **h**). En face birefringence and optic axis images were obtained from an average projection of a 200 μm slab centred at the manually labelled choroidal–scleral interface (blue dashed line). Scleral birefringence values were calculated by averaging specific fundus areas indicated by orange annular (OPSB) and hatched segment (PPSB) in **e**. **i**–**k**, Correlation analysis of OPSB vs SE (**i**), PPSB vs SE (**j**) and PPSB vs AL (**k**). **l**, Pearson correlation coefficients of PPSB vs SE and PPSB vs AL at different fundus locations. *$P \le 0.01$. **m**, Correlation matrix of biometrics of the eyes in the emmetropia or low myopia group. Scatterplots (**i,j,k**) show 69 eyes from 42 individuals, regression (lines) and 95% confidence intervals (shaded areas). *r* values were calculated using Pearson correlation. *P* values were calculated using *F*-test against a constant model. Inter-eye correlation was addressed by cluster bootstrapping. Histogram equalization was applied to **a**, **b**, **c**, **d** and **h** (left panel). S, superior; I, inferior; N, nasal; T, temporal. Scale bars: **a,a″,c**, vertical: 300 μm, horizontal: 1 mm; **e**, 1 mm.

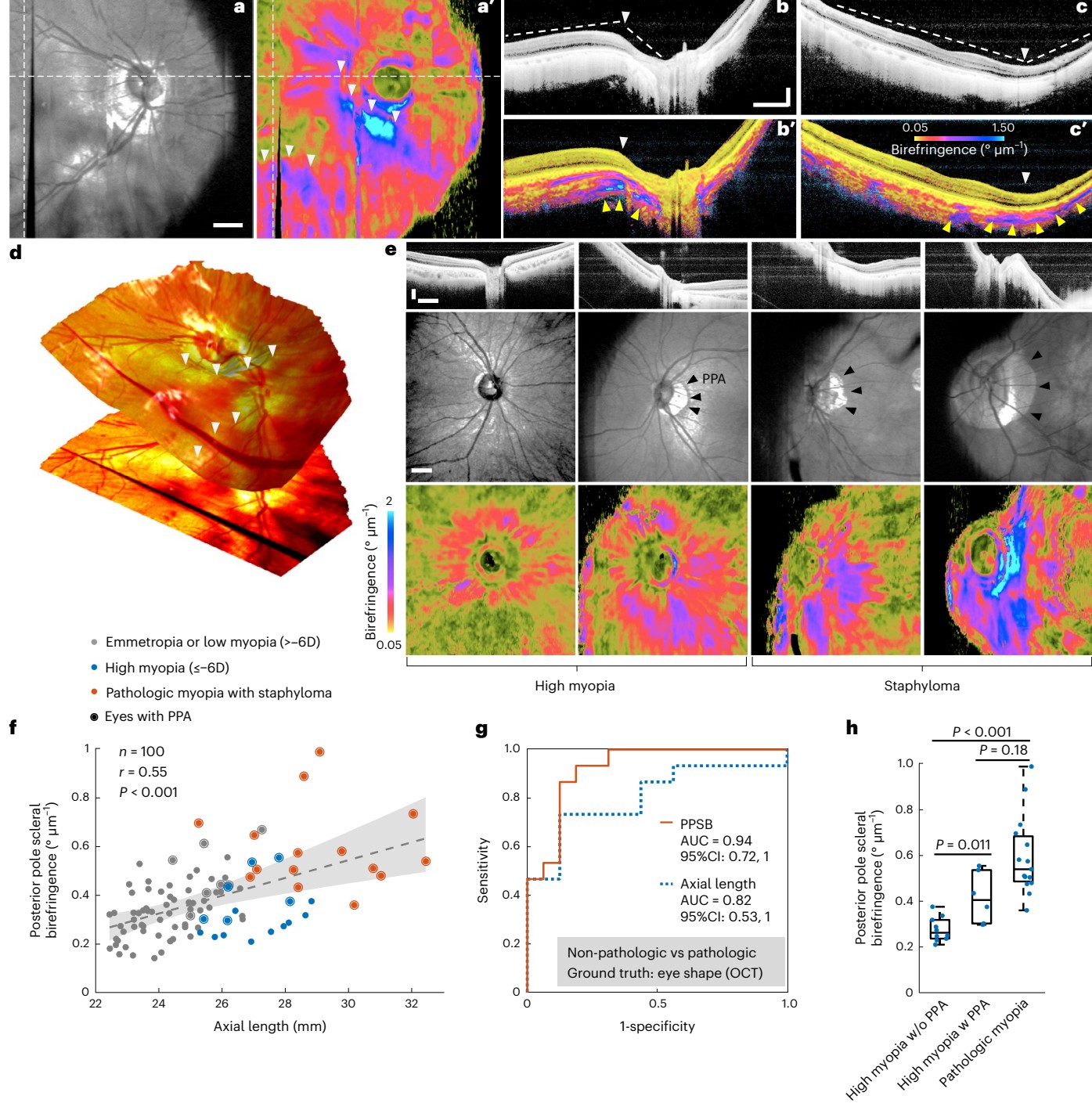

**Fig. 6 | Scleral birefringence in patients with high and pathologic myopia.**
**a**–**d**, Representative TRIPS-OCT images of a patient (61-yr-old female, OD, AL: 27.8 mm, Asian) with pathologic myopia, diagnosed by the presence of staphyloma, shown in contrasts of en face intensity (**a**), en face birefringence (**a'**), cross-sectional intensity (**b,c**), cross-sectional birefringence (**b',c'**) and three-dimensional reconstruction of the eyeball shape (**d**). The dashed lines in **a** and **a'** indicate the locations of the cross-sectional images (**b,b',c,c'**). White arrowheads (**a',b,b',c,c'**) indicate the edges of staphyloma. Yellow arrowheads (**b',c'**) indicate regions of scleral birefringence increase. **e**, Representative images of eyes with high myopia and pathologic myopia at various stages (from left to right: 32-yr-old male, OD, AL: 28.8 mm; 32-yr-old female, OS, AL: 26.9 mm; 49-yr-old female, OS, AL: 26.9 mm; 62-yr-old male, OS, AL: 30.2 mm, all Asian). Images are shown in contrasts of cross-sectional intensity (upper panel), en face intensity (middle panel) and en face birefringence (lower panel). Black arrows indicate

PPA. **f**, Correlation of PPSB and AL in all eyes (non-pathologic $n = 85$, pathologic $n = 15$). The scatterplot shows 100 eyes from 59 individuals, regression (line) and 95% confidence intervals (shaded area). The $r$ value was calculated using Pearson correlation. The $P$ value was calculated using $F$-test against a constant model. Inter-eye correlation was addressed by cluster bootstrapping. **g**, Performance of using PPSB and AL as classifiers to differentiate eyes with pathologic myopia from eyes with high myopia. **h**, PPSB values in eyes with high myopia without PPA ($n = 10$), high myopia with PPA ($n = 6$) and pathologic myopia ($n = 15$). Dots represent eyes, central line indicates median, box shows interquartile range and whiskers show range. The $P$ value was calculated using two-sided Wilcoxon rank-sum test with cluster bootstrapping to correct inter-eye correlation. Histogram equalization was applied to **b**, **c** and **e** (upper panels). Scale bars: **a**, 1 mm; **b,e**, vertical: 300 μm, horizontal: 1 mm.

doses and combinative treatment of atropine and orthokeratology[60], PSB could potentially complement the widely used SE for optimizing the treatment strategy and tailoring it to individual needs.

In our human cross-sectional study, we showed that in eyes with emmetropia and low myopia, PPSB increased on average by 0.03° μm⁻¹ as myopia progressed by −1 D, and increased by 0.05° μm⁻¹ as AL increased by 1 mm. Furthermore, in patients with pathologic myopia, we observed a spatial association between PSB and staphyloma in the sclera. In eyes with high myopia, we found that PSB increased when PPA was present and might predict further progression of the disease. In the posterior pole area, we observed the most pronounced correlation of PSB with SE and AL. This observation is in agreement with previous observations in animals, which identified that the primary myopic change in the sclera started from the posterior pole area[61,62]. The increase in PSB in myopic eyes can be explained by changes in the collagen fibre structure from interwoven to aligned, as observed in ex vivo studies, including (1) unfolding of microstructural crimps[22,29], (2) a change in collagen fibre direction from circumferential to radial in the peripapillary sclera[30], (3) reorganization of the collagen fibres into a lamellar arrangement (rather than interwoven)[63] or (4) a combination of all aforementioned phenomena[21].

In the clinical management of pathologic myopia, PSR surgery (for example, macular buckling) has been an option to arrest myopia progression. In general, due to its invasive nature and concerns about post-operative complications, PSR is only considered when ocular pathologies such as retinal detachment and myopic maculopathy (caused by progressive thinning of the sclera) are threatening or already impacting vision[60]. There is a need to detect scleral weakness as early as possible and for close monitoring to make a timely treatment decision[7]. Our results in humans showed that PSB was correlated with myopia degree and may be a sensitive indicator of scleral degeneration towards pathological conditions. TRIPS-OCT is sensitive to scleral changes in moderate and low myopia, which are not generally observed with current fundus photography or conventional OCT[64]. We speculate that TRIPS-OCT can detect subtle changes in the sclera that precede an obvious change in thickness or morphology accessible with current imaging methods, and can thus provide better guidance on the necessity and timing of PSR treatment.

Comparing the correlations between PSB and the degree of myopia in the young guinea pigs and adult humans, the PSB decreased in animals but increased in adult humans with the severity of the disease. We additionally imaged two adult guinea pigs without and with high myopia (Extended Data Fig. 6) and found the PSB was higher in the myopic eye, which is consistent with the data obtained in humans. We suppose that there is a fundamental difference in the scleral changes between the young guinea pig model and adult humans representing different stages of myopia development within the lifespan. Our data indicate that the early eye growth at younger age and the myopic elongation in adults lead to opposing effects on PSB related to scleral collagen fibre arrangement and diameter. To understand the difference between these two stages of myopia development, we analysed the collagen fibre orientation and birefringence within the regions of interest in guinea pigs and humans. In the young guinea pigs at the ages of 2 and 8 weeks (Extended Data Fig. 7), the average scleral birefringence increased with age, in conjunction with an increase in the interweaving of the collagen fibres as evidenced by a reduction in the local maxima in the angular histograms of fibre orientation. In the adult guinea pigs, the average birefringence was higher in the myopic eye, however, in conjunction with a reduction in the interweaving of the collagen fibres (Extended Data Fig. 6). The latter phenomenon was also observed in the submacular sclera of adult humans (Extended Data Fig. 8). As such, increased PSB in young guinea pigs may indicate the enlargement of scleral fibre diameter, associated with increased scleral collagen synthesis to achieve the required scleral stiffness during eye growth. In contrast, increased PSB in adult guinea pigs and

humans with myopia may indicate alterations in the arrangement of scleral collagen and reductions in interwoven fibres, associated with an extended elongation of the eyeball. However, these suppositions are solely based on TRIPS-OCT measurements and still require further histological investigations.

In this study, we introduced TRIPS-OCT, a new polarimetric modulation and reconstruction strategy for posterior scleral imaging in vivo. Today, there are three major types of PS-OCT instruments, including single-input PS-OCT[65,66], depth-encoding PS-OCT[45,46,67] and dual-input EOM-based PS-OCT[41,42]. Single-input PS-OCT offers a simplified setup but limits its use to lowly birefringent samples. When the polarization state of the local probing light occasionally coincides with the sample's optic axis, the depth-resolved reconstruction of birefringence metrics may be frustrated. Depth-encoding PS-OCT offers reliable depth-resolved measurements independent of the sample but requires a doubled ranging depth to achieve the same imaging depth as a time multiplexing system and a k-clock device to ensure phase stability[68]. The dual-input EOM-based PS-OCT is robust to environmental variation as it does not require phase stability for birefringence reconstruction, but it assumes measurements to represent pure retardance, hence it is readily impacted by the presence of diattenuation and edge artefacts induced by both the system components and the sample. TRIPS-OCT does not have the aforementioned limitations but requires a longer acquisition time because of triple repetitive scans at the same location of the sample. However, due to the development of faster lasers and OCT-angiography techniques[69], repetitive scanning has become a standard in current ophthalmic imaging. TRIPS-OCT is not sensitive to sample motion within a few micrometres, as the measurements of modulated frames are based on Stokes vectors, and the phase variation between repetitive scans does not influence the reconstruction. Overall, TRIPS-OCT mitigates some of the drawbacks of traditional PS-OCT implementations and makes it more suitable for clinical translation.

This study has several limitations that we hope can be addressed. First, TRIPS-OCT measurements are fundamentally limited by the intensity SNR. As the sclera is a dense and highly scattering structure, only a 100-μm band from the upper sclera can be reliably measured. In this study, we estimated the intensity SNR of the 100-μm scleral band and excluded approximately 16% of all imaged eyes due to insufficient intensity SNR. We found that a thick choroid (roughly >450 μm) was one of the factors limiting light penetration to the sclera. In addition, we observed that Caucasian eyes provide slightly better penetration of the sclera than Asian eyes, perhaps due to less scattering and absorption caused by lesser pigmentation. Second, as both the diameter and the alignment of the collagen fibres determine the birefringence measured by TRIPS-OCT, the explanation of increased PSB requires further analysis of the collagen fibre orientation and previous knowledge of the underlying structures. Lastly, the scleral birefringence measurement highly depends on the segmentation of the choroidal–scleral interface, which is performed manually at present. The issue with manual segmentation is that the interface between the choroid and the sclera is not well defined. There are fine petaloid scleral structures unevenly connecting to the above choroid tissues and large blood vessels going through the sclera from the choroid. To minimize the truncation of the inner parts of the sclera, we used a 200-μm band centred on the choroidal–scleral interface to produce the en face images and quantify birefringence. Incorporating an area including the choroid does not affect the PSB measurement because no structure in the choroid is observed to be birefringent. In addition, manual labelling is a time-consuming and subjective process, whereas an automated and reliable image segmentation algorithm will improve the accuracy and efficiency of TRIPS-OCT analysis.

We have reported the development of a polarimetric imaging technique, TRIPS-OCT, and revealed PSB as a biomarker for myopia by imaging the posterior sclera in eyes with different presentations of myopia. We anticipate TRIPS-OCT to be potentially useful in the

diagnosis of other ocular conditions that are related to scleral anomalies. Furthermore, TRIPS-OCT may be applied to imaging systems based on fibre probes, thus bringing new opportunities in intravascular and endoscopic applications.

## Methods

### Triple-input modulation

A swept-source optical coherence tomography (SS-OCT) system previously developed by our group[70] was modified to achieve TRIPS-OCT. The OCT system employed a swept-source laser (1,060 nm, sweep rate 200 kHz, tuning range 100 nm, Axsun Technologies). The measured axial full-width-at-half-maximum of the intensity PSF in air was 6.1 μm. In a swept cycle, the digitization was triggered by the laser trigger signal with a constant sampling rate of 1 GHz, and the measured 3 dB roll-off ranging depth was 3.5 mm. The beam size entering the pupil was 0.67 mm, corresponding to an optical lateral resolution of 44 μm and a depth of focus of 2.9 mm in a normal human eye with an axial length of 23 mm. The spatial averaging in the birefringence reconstruction, including filtering of Stokes vectors and spectral binning, reduced the resolution to 150 μm in the lateral and 30 μm in the axial directions for birefringence imaging. The laser power entering the eye, which was controlled by a motorized aperture placed in the free space before the triple-input modulator, was set to 1 mW for alignment and 4 mW for retinal volume scan.

We replaced the polarization-depth-encoding unit with a triple-input modulator (Extended Data Fig. 1b) consisting of a polarizer and an EOM (4104NF, Newport), inspired by the previous dual-input configuration[71]. A triple-step voltage driving signal (Extended Data Fig. 1c) and an angle of 27.37° between the optic axis of the EOM and its preceding linear polarizer (Extended Data Fig. 1d) allowed for the generation of three mutually orthogonal polarization states on the Poincaré sphere (Extended Data Fig. 1e and Supplementary Method 3). Modulating the polarization states between three repeated frames enabled the reconstruction of the Mueller matrix at each location within the triple-measured frames. The reconstruction involved an algorithm (Supplementary Method 1) fitting the measured Stokes vectors to Mueller matrices that respect physical polarization constraints and describe cumulative diattenuation and retardance. From the reconstructed Mueller matrices, we isolated the cumulative sample retardance and computed local depth-resolved tissue birefringence and optic axis orientation.

### Mueller matrix reconstruction

To remove wavelength-dependent polarization effects, spectral binning was performed by dividing the spectral fringe into 9 overlapping bins. The signals from the 2 detection channels were converted to Stokes vectors and filtered along both fast and slow lateral scan directions (kernel size: 30 μm for guinea pigs and 150 μm for pigs and humans) for each of the three input states. The Stokes vectors of each spectral bin were modelled as $\mu = DLs$, where $s$ is the probing matrix of the three input states, and the $4 \times 3$ $\mu$ matrix is composed of the three averaged Stokes vectors. $D$ is a general depolarizing matrix. $L$ is a nondepolarizing, so-called Jones-derived Mueller or pure Mueller matrix, representing the retardance and diattenuation to be recovered. Note that $L$ has only 7 free independent parameters, while $\mu$ has 12. Although insufficient to fully recover $D$, we corrected its estimated effect on $\mu$ by polarizing the Stokes vectors composing $\mu$ to recover $L$, as described in detail in Supplementary Method 1.

### Combining spectral bins

For each spectral bin, the Mueller matrix $L$ describing the cumulative round-trip through the sample and the system was recovered as described above. Next, the polarization reciprocal symmetry of round-trip optical transmission was recovered for each bin (Supplementary Method 4). The remaining constant alignment of these matrices to the central spectral bin, described by similarity transformation with a pure $4 \times 4$ Mueller matrix, was solved by minimizing the alignment error of adjacent bins using 10 frames randomly sampled from a volume scan (Supplementary Method 5). After the alignment, the reconstructed Mueller matrices from the 9 bins were averaged elementwise to obtain a general Mueller matrix image $M(x, z)$, where $x$ and $z$ are the coordinates along the fast scan axis and depth, respectively. Although the initial matrices of the individual spectral bins are pure Mueller matrices, averaging introduces depolarization, which was removed using polar decomposition[72]: $M(x,z) = M_\triangle(x,z)M_R(x,z)M_D(x,z)$, where $M_\triangle(x,z)$ is a depolarizer, $M_R(x,z)$ is a retarder and $M_D(x,z)$ is a diattenuator. Combined, $M_P(x,z) = M_R(x,z)M_D(x,z)$ defines the pure Mueller matrix of the cumulative round-trip to sample depth $z$.

### System and corneal birefringence compensation

The cumulative round-trip pure Mueller matrix of the retinal surface, $S(x)$, was identified as a function of the lateral position. The matrix $C(x)$, representing the single-pass linear retardance and diattenuation effect of transmission through the cornea and the anterior eye to the retinal surface, was obtained by taking the square root of $S(x)$. It is critical to unwrap the exponential generator of $S(x)$ to force continuity of the corneal retarder and diattenuator not only in the $x$ direction but also in the $x$–$y$ plane. The transmission through the system and the cornea was then compensated by $M_{PC}(x, z) = C^{-1}(x)M_P(x, z)C^{-1}(x)$, where $M_{PC}(x, z)$ is the compensated cumulative round-trip Mueller matrix (Supplementary Method 6). Any single-pass circular retardance and diattenuation cancels on the round-trip, evading S($x$), and remains uncompensated.

### Local birefringence reconstruction

Polar decomposition was further applied to remove diattenuation $M_{DC}(x, z)$ as $M_{PC}(x, z) = M_{RC}(x, z)M_{DC}(x, z)$. The local Mueller matrix image $m(x, z)$ was reconstructed recursively along depth from $M_{RC}(x, z)$[73] as follows:

$$m(x, z) =$$
$$\sqrt{m^{-1}(x, z+\mathrm{dz})\cdots m^{-1}(x,z_0) M_{RC}(x, z) m^{-1}(x, z_0) \cdots m^{-1}(x, z+dz)},$$

where $m(x, z_0) = \sqrt{M_{RC}(x, z_0)}$ represents the first row of pixels of the local Mueller matrix image. The depth-resolved optic axis orientation and birefringence were then extracted from $m(x, z)$. We used an acrylonitrile butadiene styrene phantom to validate the depth-resolved optic axis reconstruction (Supplementary Method 7).

### TRIPS-OCT imaging of pig and guinea pig eyes

The use of animals for these studies was approved by the Institutional Animal Care and Use Committee of SingHealth (AAALAC Accredited; 2018/SHS/1441, IACUC 1290). All procedures adhered to the ARVO Statement for the Use of Animals in Ophthalmic and Vision Research. In the studies using guinea pig and pig models, the animals were anaesthetized with an intramuscular injection of a cocktail of ketamine hydrochloride (27 mg kg$^{-1}$) and xylazine hydrochloride (0.6 mg kg$^{-1}$). The scan was performed with a field of view of 22°, corresponding to a ~9 × 9 mm area in the pig eye and a 4 × 4 mm area in the guinea pig eyes. The scanning was performed centred on the ONH by positioning according to a preview of the OCT B-scans. The volume scan comprises 1,000 × 1,000 × 3 A-scans over a square area, with 3 repetitive B-scans on the same location for the triple-input modulation.

### Pig eye TRIPS-OCT imaging and PLM histology

The left eye of a 1-yr-old pig (Yorkshire-Landrace cross, male, National Large Animal Research Facility, Singapore) was scanned using TRIPS-OCT. After TRIPS-OCT imaging, the pig was euthanized with

an overdose of sodium pentobarbital (80–100 mg kg⁻¹) and the left eye globe was collected. The entire globe was fixed in 10% formalin for 24 h. After fixation, the eye was transferred to phosphate buffered saline (PBS). The posterior pole region centred on the ONH was cryosectioned transversely into 30-μm-thick sections and mounted on glass slides without staining. Fifty sections were obtained and imaged using a customized polarized light microscope.

## Longitudinal guinea pig model

Twenty-one guinea pigs (Elm Hill Labs, female $n = 13$, Chelmsford), including albino ($n = 17$) and pigmented ($n = 4$) strains, were bred on site. The animals were reared under a 12 h light/12 h dark cycle with lights on at 08:00 in the animal-centre facilities. The animals had free access to standard food and water. Fresh vegetables were provided twice a day. Refraction data were collected from 1 to 8 weeks using streak retinoscopy and were reported as the spherical equivalent refractive error (SE). Retinoscopy was performed on cycloplegic eyes in alert animals. TRIPS-OCT imaging was performed weekly on both eyes of the animals.

## Adult guinea pig model

Two albino guinea pigs (Elm Hill Labs, female $n = 1$, Chelmsford) were selected for TRIPS-OCT imaging from a group of breeders in our animal facility. The selection criteria for these animals were as follows: (1) older than 1.5 yr, (2) with clear and healthy eyes without any evidence of anterior segment or retinal changes and (3) emmetropia (SE = 0$D$) or high myopia (SE ≤ −6$D$). Retinoscopy was performed on cycloplegic eyes in alert animals. TRIPS-OCT imaging was performed on the eyes meeting these inclusion criteria.

## Guinea pig TEM

Three guinea pigs (Elm Hill Labs, male $n = 3$, Chelmsford) aged 1, 12 and 16 weeks were killed for TEM histology analysis. The animals were euthanized with an overdose of sodium pentobarbital (80–100 mg kg⁻¹). After in vivo TRIPS-OCT imaging, the eye globes were collected and immersed in 0.05 M cacodylic acid sodium and 2.5% glutaraldehyde with PBS (pH 7.4) for 2 h. Then, the cornea and lens were dissected. A 2 × 2 mm section of scleral tissue from the superior region of the ONH of each eye was sampled and postfixed in 1% osmium tetroxide with PBS (pH 7.4) for 1 h at 4 °C, stained with 1% uranyl acetate with double distilled water for 2 h, rinsed and dehydrated in graded acetone before embedding in Araldite. Micrographs of histologic 100-nm-thick cross-sections were imaged using a transmission electron microscope (JEM-2100, Jeol). Electron micrographs were taken from the sclera region at magnifications of ×100, ×1,000 and ×10,000.

## Human recruitment

All procedures performed were in adherence with the ethical standards of the SingHealth Centralized Institutional Review Board (CIRB Ref No. 2021/2592). Written informed consent was obtained from all participants in accordance with the Declaration of Helsinki. The recruitment was conducted under two cohorts. From the cohort of normal participants, 80 normal adult volunteers without any ocular diseases were recruited. The inclusion criteria were as follows: age 21 yr and above; no diabetes and free from clinically relevant eye disease that interferes with the aim of the study including glaucoma, diabetic retinopathy, age-related macular degeneration, uveitis or vascular occlusive diseases.

From the cohort of participants with pathologic myopia, 10 adult patients diagnosed with pathologic myopia with staphyloma were recruited. The inclusion criteria were as follows: age > 21 yr; staphyloma observed in both eyes on wide-field OCT imaging. The exclusion criteria consisted of eye conditions that might result in poor-quality imaging scans (severe cataract, corneal haze/opacity).

## Autorefraction, axial length and TRIPS-OCT imaging

In the cohort of normal participants, cycloplegia was achieved using 3 drops of 1% cyclopentolate administered 5 min apart, and cycloplegic autorefraction was measured 30 min after the last drop using a Canon RK-F1 autorefractor (Canon). Five readings, all of which were less than 0.25 $D$ apart, were averaged. The SE was calculated as the sphere plus half cylindrical power. For those who had undergone refractive surgery, data were obtained from records before the surgery. The AL was obtained using a Zeiss IOL Master (Carl Zeiss Meditec). Five readings, all within 0.05 mm or less, were averaged. TRIPS-OCT scans were performed on both eyes with 700 × 700 × 3 A-lines in a region of 8 × 8 mm centred on the ONH.

In the cohort of participants with pathologic myopia, AL was obtained using Zeiss IOL Master (Carl Zeiss Meditec). Autorefraction was not performed due to low accuracy in such patients. TRIPS-OCT scans of both eyes with 700 × 700 × 3 A-lines in a region of 8 × 8 mm centred on the ONH were performed twice, with the vertical and horizontal directions as the fast and slow scan directions, respectively.

## TRIPS-OCT image processing

TRIPS-OCT data acquisition was controlled by an interface software developed using NI LabVIEW (2020, National Instruments). TRIPS-OCT images, including those of animal models and humans, were reconstructed with contrasts of the intensity, birefringence and optic axis. The cross-sectional intensity images of human eyes were averaged over 12 adjacent B-scans, followed by histogram equalization to enhance the contrast of the sclera. Cross-sectional birefringence images were synthesized in the hue, saturation, value (HSV) colour model, with the H and S channels encoding the birefringence value and the V channel encoding the intensity value. En face birefringence images were constructed as single-channel images cast into the birefringence colour map with a constant V channel. Cross-sectional and en face optic axis images were synthesized in the HSV colour model, with the H and S channels encoding the optic axis orientation and the V channel encoding the birefringence value. In optic axis images, pixels with intensity SNR lower than 1 dB were set to background and were replaced with black colour.

## Birefringence quantification

**Guinea pigs.** To obtain en face birefringence images in guinea pigs, the depolarization index was calculated from the reconstructed general Mueller matrix $\mathbf{M}(x, z)$ and a threshold of 0.9 to define a mask for removing the background (Supplementary Method 8). A 30 μm vertical-line kernel was used to filter each cross-sectional birefringence image, followed by a maximum projection along the depth. The en face image was converted to polar coordinates around the ONH. The overall PSB value was obtained by a maximum birefringence projection along the radial direction, followed by averaging along the circumferential direction.

**Humans.** In each cross-sectional image, 20 points were initially manually placed at the choroidal–scleral interface with a spline interpolation to define the segmentation. The labeller was free to add more points and define a finer segmentation by means of an interactive interface (Supplementary Method 7). Labelling was solely based on the intensity image and performed by a postdoc OCT expert. A 200 μm slab on the birefringence image, centred on the choroidal–scleral interface 100 μm above and 100 μm below to include fine structures on the scleral surface, was summed to project onto the en face direction (Supplementary Method 8). In the projected en face image, the ONH and fovea were manually labelled. In polar coordinates with the ONH as the origin, 0° was defined as the vector pointing to the fovea. OPSB was defined as the mean birefringence value of the annular area centred on the ONH, with inner and outer circle radii defined as 0.3 and 0.7 of the ONH–fovea distance, respectively. The annulus was evenly divided into 12 radial segments. The PPSB was defined as the mean value of the segment located between the ONH and the fovea.

**Statistical analysis.** As this study of PSB was a pilot study, no sample size calculation was performed for the animal experiments owing to the lack of previous studies. Human sample size estimation was based on the evaluation of the correlation between refractive error and TRIPS-OCT measurements with 90% statistical power using preliminary parameters from the longitudinal guinea pig study. Analyses of the correlations between scleral birefringence and other biometrics were performed by univariate linear regression. Correlation significance analyses were performed by conducting an *F*-test on the linear model. Significance analyses for scleral birefringence changes were performed by the Mann–Whitney *U* test. Inter-eye correlation in the same participant was addressed by cluster bootstrapping. Specifically, to determine the CIs and *P* values of related statistics, a random resampling process with the original sample size was performed with replacement at the participant level, and the process was repeated 5,000 times, generating distributions of related statistics. Estimation of statistics was derived from the median of the generated distribution, and the 95% CI of the statistics was derived from the 2.5th and 97.5th percentiles. All analyses were performed using MATLAB (R2019b, R2020b, R2021b, MathWorks).

### Reporting summary

Further information on research design is available in the Nature Portfolio Reporting Summary linked to this article.

## Data availability

Processed animal data (shown in Figs. 3 and 4), including en face images and refractive errors, are available from figshare[74]. Additionally, one example guinea pig B-scan modulated by triple polarization states (shown in Fig. 1) is also available from figshare[74]. The entire raw dataset of animal experiments is more than 25 TB in size and can be shared upon request with appropriate data transfer methods. For the clinical study, the raw data acquired during the study are available for at least 5 years from the corresponding author on reasonable request, subject to approval from the SingHealth Centralised Institutional Review Board. A request will be processed within 3 months.

## Code availability

The central algorithm to reconstruct TRIPS-OCT images from triple-measured Stokes vectors can be found at https://github.com/DrXinyu/TRIPS-OCT.

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

## Acknowledgements

We thank J. M. Busoy and K. J. V. Catbagan of the Singapore Eye Research Institute for keeping and handling the animals; Q. Hu, B. Kulantayan, H. Chye, J. L. H. Tay and S. B. J. Pow of the Singapore Eye Research Institute for coordinating with patients; J. Gnalian of the University of Pittsburgh for performing the PLM histology;

L. Liu of Nanyang Technological University, J. Ren of Harvard Medical School and T. Ling of Nanyang Technological University for discussion on data presentation; and C. Zhang of Tsinghua University for discussion on statistics. This work was funded by grants from the Industry Alignment Fund - Industry Collaboration Projects (IAF-ICP) Grant (I1901E0038, L.S., Q.V.H., A.W.C., R.P.N., V.A.B., M.A. and S.-M.S.) and Johnson & Johnson Vision. We also acknowledge the support of the National Medical Research Council (CG/C010A/2017_SERI, L.S.; OFLCG/004c/2018-00, L.S.; MOH-000249-00, J. Chua; MOH-000647-00, L.S.; MOH-001001-00, L.S.; MOH-001015-00, L.S.; MOH-000500-00, L.S.; MOH-000707-00, L.S.; MOH-001072-06, L.S.; NMRC/CSIRG/MOH-000531/2021, Q.V.H.); the National Research Foundation Singapore (NRF2019-THE002-0006, L.S. and NRF-CRP24-2020-0001, L.S.), A*STAR (A20H4b0141, L.S., J. Chua), the Singapore Eye Research Institute & Nanyang Technological University (SERI-NTU Advanced Ocular Engineering (STANCE) Program, L.S.), the SERI-Lee Foundation (LF1019-1, J. Chua), the US National Institutes of Health (P41EB-015903, M.V. and R01 EY023966, I.A.S.), the EU (H2020-MSCA-IF-2019 program 894325, M.L.) and the Singapore Eye Research Institute & National University of Singapore ASPIRE Program (NUHSRO/2022/038/Startup/08, R.P.N.).

## Author contributions

X.L. developed the TRIPS-OCT system. L.J., Q.V.H., X.L., M.K., R.P.N., M.L., L.S. and V.A.B. designed and conducted the animal studies. I.A.S. and X.L. conducted the PLM analysis. X.L., M.K., J. Chua, Q.V.H., A.W.C., B.T., J. Cheong, V.B., R.S.C., M.J.A.G., M.A., S.-M.S. and L.S. designed and conducted the human studies. M.V. and X.L. developed the TRIPS-OCT reconstruction method. L.S., X.L., M.V., R.P.N., G.G. and I.A.S. interpreted the data. L.S. conceived the overall idea, obtained funding and supervised the entire study. All authors read and edited the manuscript.

## Competing interests

X.L. and L.S. are inventors on a provisional patent (10202009128V, Singapore, 2020) related to TRIPS-OCT technology. The other authors declare no competing interests.

## Additional information

**Extended data** is available for this paper at https://doi.org/10.1038/s41551-023-01062-w.

**Correspondence and requests for materials** should be addressed to Leopold Schmetterer.

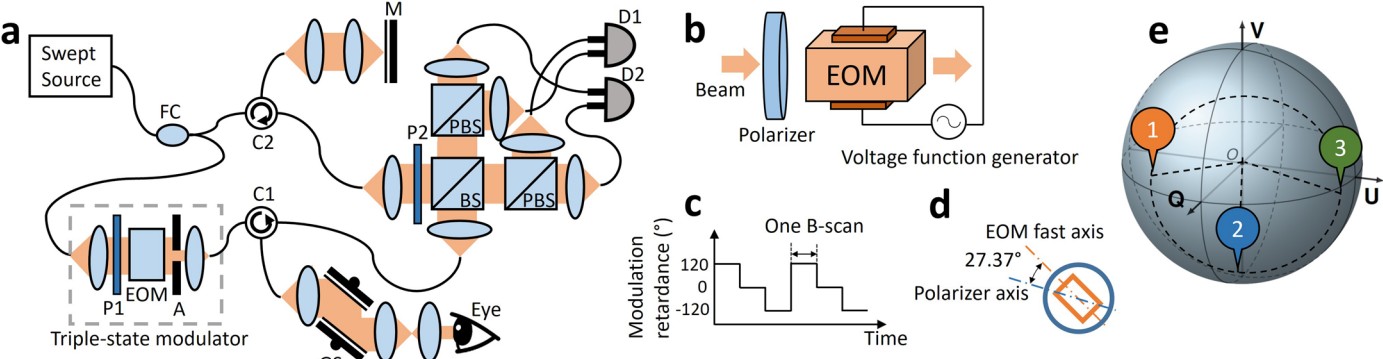

**Extended Data Fig. 1 | Triple-input PS (TRIPS) OCT. a**, TRIPS-OCT system schematic. Briefly, compared to a conventional OCT system that records only the intensity of the light, the polarization states of light are additionally recorded by a polarization diversity detection circuit (BS, PBSs, D1, D2). The triple-state modulator was inserted into the sample arm. **b**, Triple-state modulator. The modulator comprises a polarizer and an EOM. **c**, EOM modulation scheme. The EOM is driven by a triple-step voltage to produce three retardance values,

−120°, 0°, and 120°. **d**, Polarizer axis configuration. The angle between the linear polarizer preceding the EOM and the optic axis of the EOM is set as 27.37°. **e**, Mutually orthogonal polarization states on the Poincaré sphere resulting from triple-state modulation. FC, Fibre coupler. P1-2, Polarizer. A, Motorized aperture. C1-2, Circulator. PBS, Polarizing beam splitter. BS, Nonpolarizing beam splitter. SG, Scanning galvo mirrors. EOM, Electro-optic modulator. M, Mirror. D1-2, Photodetector.

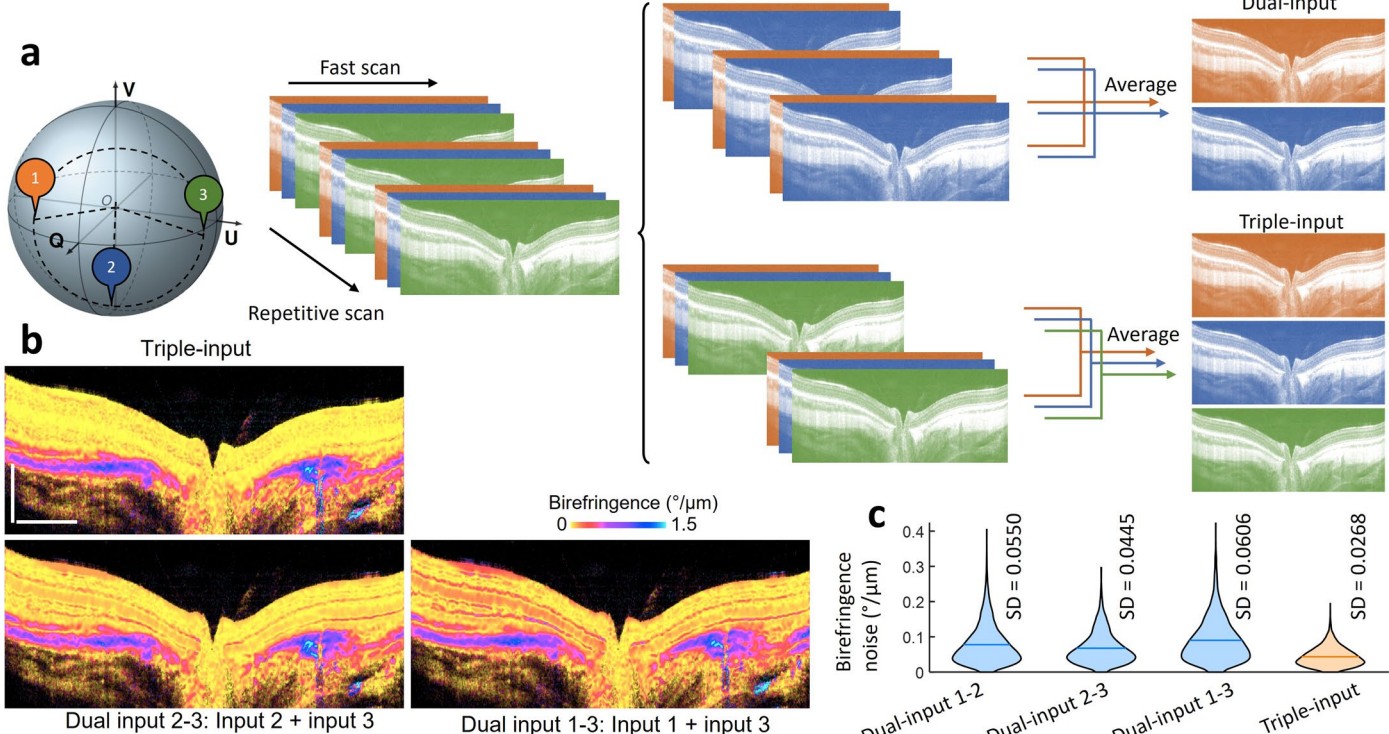

**Extended Data Fig. 2 | Comparison between dual-input and triple-input PS-OCT (TRIPS-OCT) methods under the condition of the same acquisition time.** **a**, Nine repetitive scans modulated by triple polarization states of the guinea pig retina were acquired *in-vivo*. Different sets of 6 of the 9 repetitive scans were used to reconstruct the birefringence images with the dual-input method and TRIPS-OCT, respectively. For the dual-input method, the three scans modulated by the same input polarization state were averaged before birefringence reconstruction. For the triple input method, the two scans modulated by the same polarization state were averaged before birefringence reconstruction. The averaging was performed on the intensity images without considering the phase. This averaging process confirms that the acquisition time of the data used by the dual-input and triple-input methods for the birefringence reconstruction are identical. **b**, Birefringence images reconstructed from triple-input and dual-input methods using different combinations of the input polarization states. **c**, Birefringence noise characterization for the different combinations using the inner retina in Fig. **a** indicated by the orange area (pixel number n = 5117 from 1 cross-sectional image). Central lines of violin plots indicate mean. The improvement in noise performance using TRIPS-OCT is quite consistent between different combinations of Stokes vectors. The slight difference in the noise level of dual-input combinations is due to the dependency between the edge-artifacts and the absolute polarization states (Supplementary Discussion 1). SD: Standard deviation.

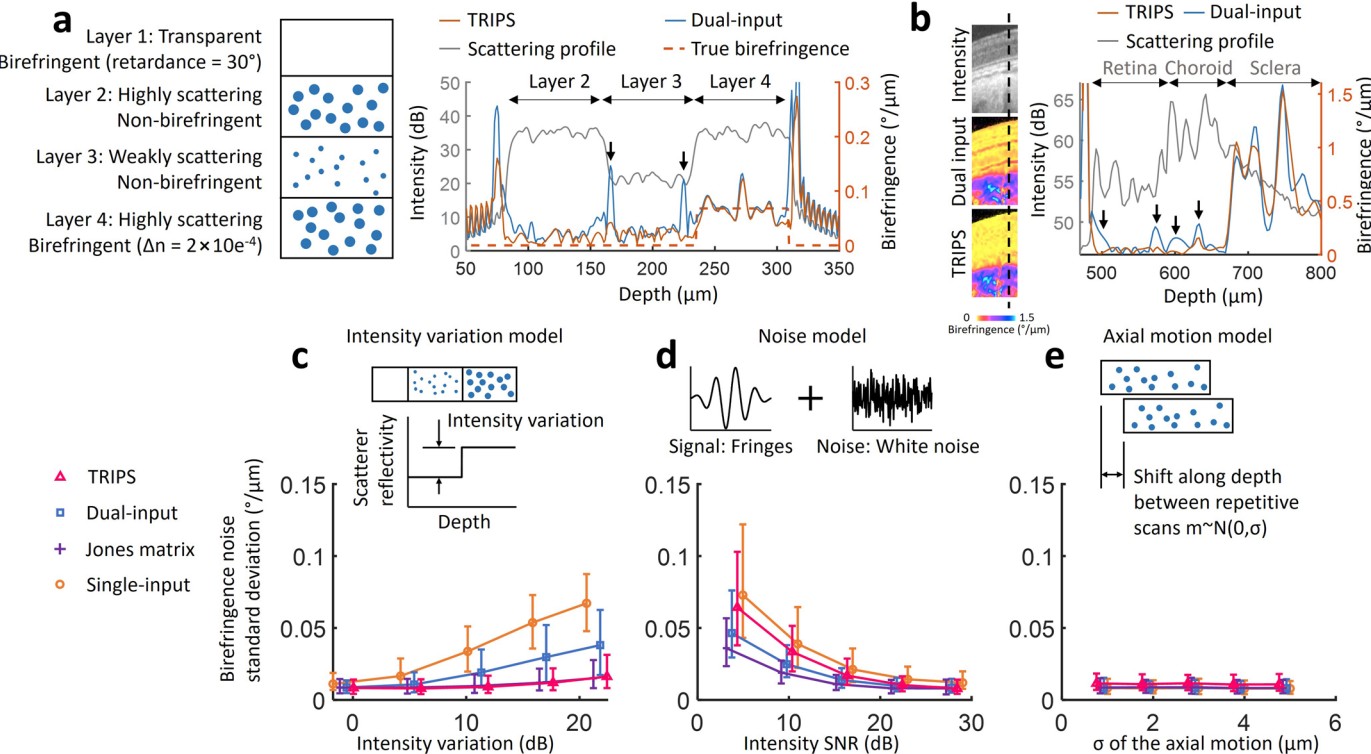

**Extended Data Fig. 3 | Noise analysis for PS-OCT. a**, Simulation comparing TRIPS and dual-input methods on a four-layer sample. Black arrows indicate edge-artifacts in the dual-input reconstruction method, absent in TRIPS-OCT. The artifacts are pronounced at brick-wall-jumps in the scattering profile. **b**, TRIPS *vs*. dual-input reconstruction of an A-scan of a guinea pig retina from the B-scan in Fig. 1a. Black arrows indicate the observed edge-artifacts. **c**, Numerical model studying edge-artifacts and intensity profile variations. The sample is modelled by two nonbirefringent scattering layers under a transparent birefringent layer with random optic axis and a retardance of 20°. Intensity variation is created by changing the reflectivity of the scatterers within each layer. **d**, Numerical model studying birefringence noise and intensity signal-to-noise ratio (SNR). The sample is modelled by a nonbirefringent scattering layer. White noise is added to the simulated fringes to create different SNRs. **e**, Numerical model studying birefringence noise and axial motion. Axial motion

is modelled by a random shift of the sample along depth with a zero mean and a standard deviation σ. Scattering layers are modelled by scatterers embedded within nonbirefringent or birefringent media, creating fully developed speckle patterns in OCT scans. OCT scans are simulated by generating the fringes in wavenumber domain of the individual scatterers and then transforming the summed fringes into the depth domain using Fourier transformation with a resolution of 6 micrometres. Sixteen scans with independent speckle patterns are averaged to suppress the speckle noise before proceeding to birefringence reconstruction. Birefringence reconstruction (Supplementary Method 2) is performed by TRIPS, dual-input geometric reasoning, Jones matrix and single-input geometric reasoning. Data are presented as mean values +/− 95% confidence intervals, which are created by bootstrapping with n = 500 repetitive simulations on random optic axis and random positioning of the scatterers.

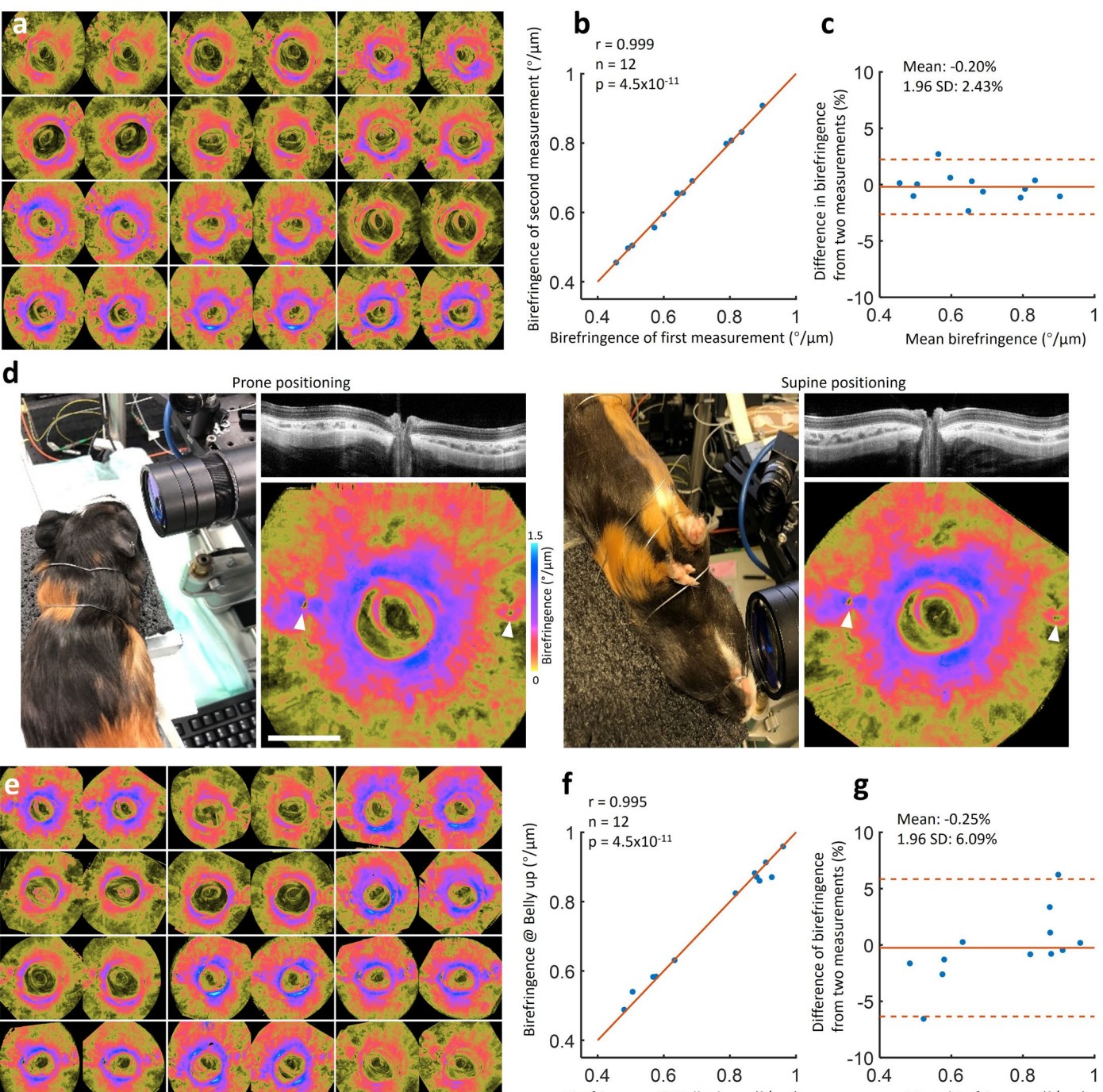

**Extended Data Fig. 4 | Repeatability of scleral birefringence measurements in guinea pigs using TRIPS-OCT. a**, Twelve guinea pig eyes were imaged repetitively with identical imaging angles. **b**, Linear correlation of mean values of repetitively measured birefringence. **c**, Bland–Altman plot of the repetitively measured birefringence. **d**, Repeatability test under varying imaging angles. To test the repeatability under slightly varying imaging angles, TRIPS-OCT imaging was performed repetitively on guinea pigs placed in both the prone and supine positions. Cross-sectional images indicate slightly varying retina tilt. Blood vessels indicated by white arrows are used to register the rotated birefringence images. **e**, Twelve eyes were imaged repetitively with prone and supine

positions. **f**, Linear correlation of the mean values of the repetitively measured birefringence. **g**, Bland–Altman plot of the repetitively measured birefringence. The $r$ values are calculated by Pearson correlation. The $p$ values are calculated by $F$-test against a constant model. Excellent repeatability was achieved under slightly varying imaging angles because the retardance and diattenuation of the cornea were correctly compensated for in the reconstruction. The 1.96 SD error was 6.1% under different imaging angles, higher than the value of 2.4% under identical imaging angles, perhaps due to the slight change in scleral thickness under different tilts. SD: Standard deviation.

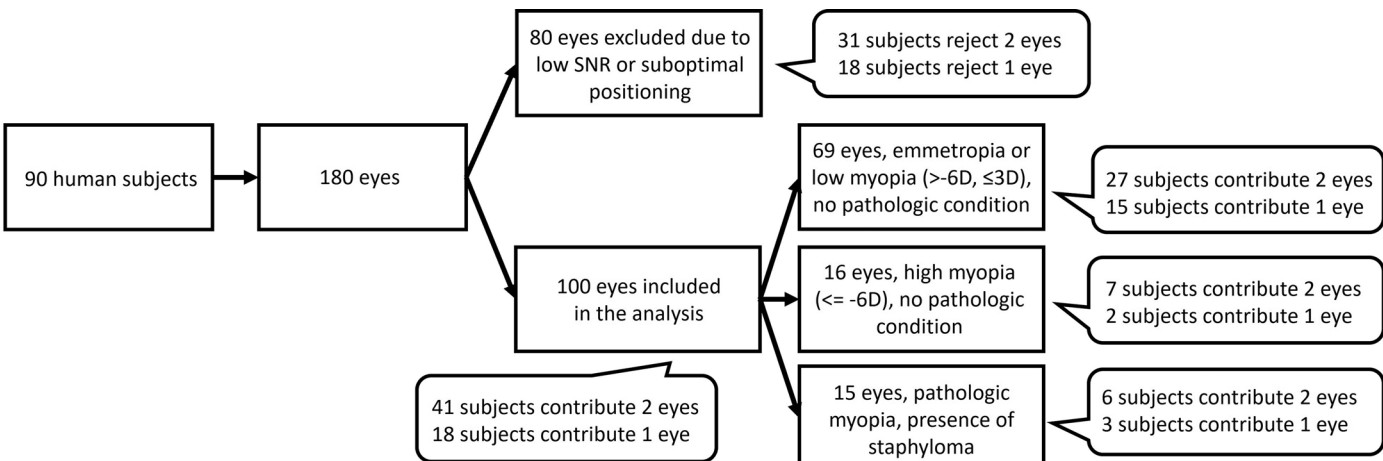

**Extended Data Fig. 5 | Data summary and analysis in the human study.** Flow diagram summarizing the human subjects and eyes included and excluded in the analysis. Eyes included in the analysis were grouped into three groups (more characteristics are provided in Supplementary Table 1). Group 1: Emmetropia or low myopia (3 D ≤ SE< −6 D) with no pathological conditions, 69 eyes from 42 subjects (23 female) with a mean age of 41.29 years, a mean SE of −1.74 D, and a mean AL of 24.44 mm. Group 2: High myopia (≤ −6D) with no pathological conditions, 16 eyes from 9 subjects (8 female) with a mean age of 39.20 years, a mean SE of −7.72 D, and a mean AL of 26.88 mm. Group 3: Pathologic myopia with staphyloma, 15 eyes from 9 subjects (6 female) with a mean age of 58.22 years and a mean AL of 29.05 mm. SE, Spherical equivalent refractive error. AL, Axial length.

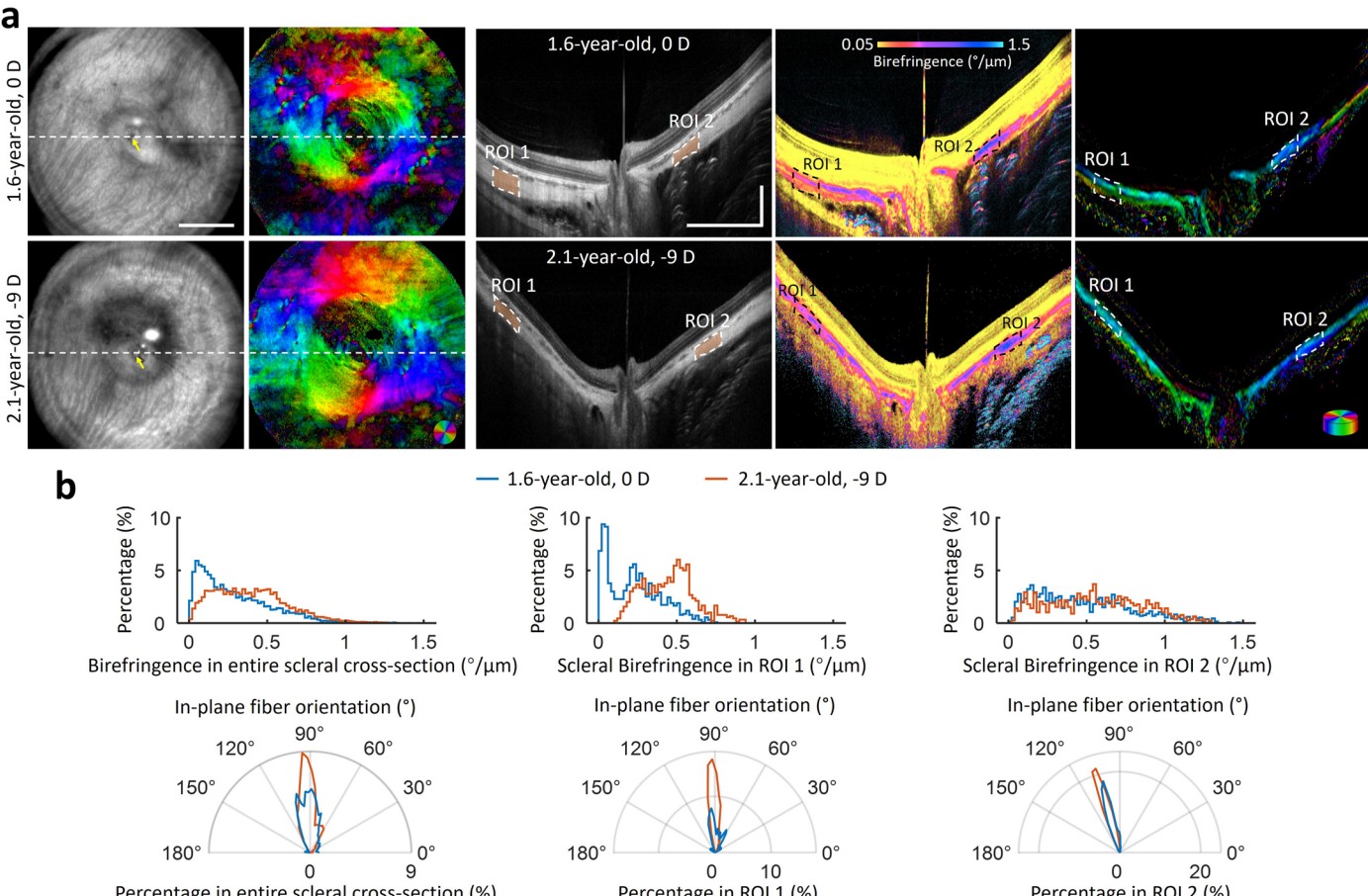

**Extended Data Fig. 6 | Scleral collagen fibre orientation and birefringence in adult guinea pigs without and with myopia, as assessed with TRIPS-OCT. a**, Images of two adult guinea pig eyes without and with myopia. En face intensity images are obtained from an average projection along the depth. En face optic axis images are obtained from the outer layer of the sclera after flattening the images using the surface of the retina. White dotted lines indicate the locations of the cross-sectional images. The sclera is manually segmented in the cross-sectional images, and two regions of interest (ROIs) are selected in each image for localized comparison. The locations of cross-sectional image are roughly matched by the same relative position according to the optical nerve head. From the cross-sectional images, we observe thinning of the sclera tissue and deformation of the eye shape towards axial elongation in the highly myopic eye

(−9 D). **b**, Histograms of measured birefringence and collagen fibre orientation in the entire sclera (left panel, pixel numbers n = 10399, 8477 for each region), ROI 1 (middle panel, pixel numbers n = 1269, 898 for each region) and ROI 2 (right panel, pixel numbers n = 1330, 784 for each region) from the cross-sectional images. Comparing the birefringence measurements, the average scleral birefringence increases in the myopic eye. The interweavement of collagen fibre, however, decreases as evidenced by that the local maxima in the angular histograms of the myopic eye are higher as compared to the emmetropic eye. As a result of scleral collagen remodelling, the increase in PSB in adult guinea pigs with myopia may be due to the augmented collagen fibre alignment and the reduction of interwoven fibres. Scale bars, **a**, vertical: 300 μm, horizontal: 1 mm.

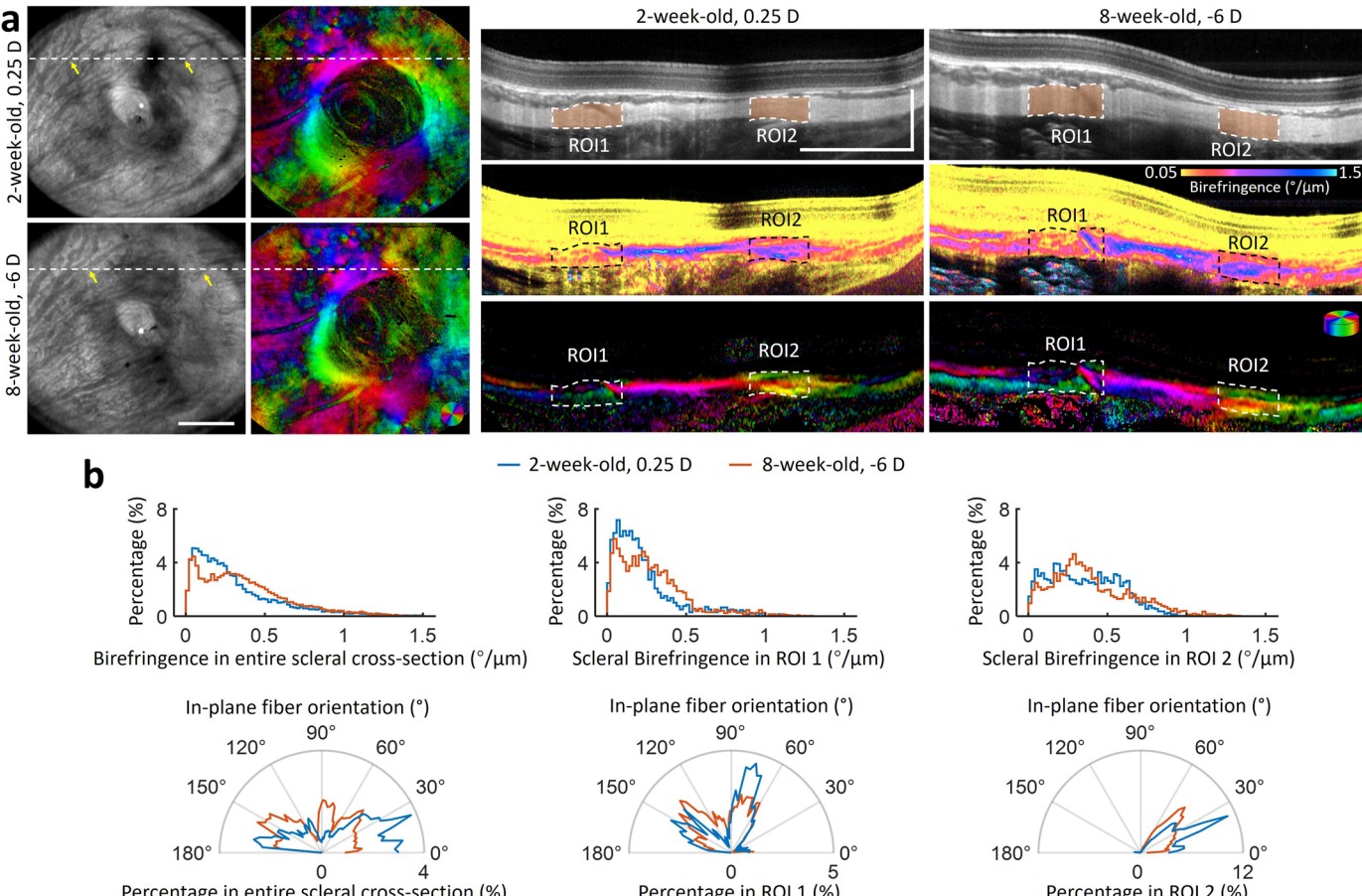

**Extended Data Fig. 7 | Scleral collagen fibre orientation and birefringence in a young guinea pig at ages of 2 and 8 weeks, as assessed with TRIPS-OCT.**
**a**, Images of a guinea pig eye at ages of 2 weeks and 8 weeks, respectively. En face intensity images are obtained from an average projection along the depth. En face optic axis images are obtained from the inner layer of the sclera after flattening the images using the surface of the retina. White dotted lines indicate the locations of the cross-sectional images. The sclera is manually segmented in the cross-sectional images, and two regions of interest (ROIs) are selected in each image for localized comparison. ROIs are registered by choroidal vessel patterns, indicated in a by yellow arrows. **b**, Histograms of measured birefringence and collagen fibre orientation in the entire sclera (left panel, pixel numbers n = 10717, 15096 for each region), region of interest 1 (ROI1, middle panel, pixel numbers n = 1981, 2615 for each region) and ROI 2 (right panel, pixel numbers n = 2772, 3732 for each region) from the cross-sectional images. Comparing the measurements at ages of 2 and 8 weeks, the average scleral birefringence increases with aging, whereas the distribution of collagen fibre orientation broadens as evidenced by a reduction of the local maxima and an increase of the local minima in the angular histograms. Broadening of the fibre orientation distribution may be due to an increase in fibre interweavement, the increase in the interwoven fibre diameters, or a combined effect of both. Scale bars, **a**, vertical: 300 μm, horizontal: 1 mm.

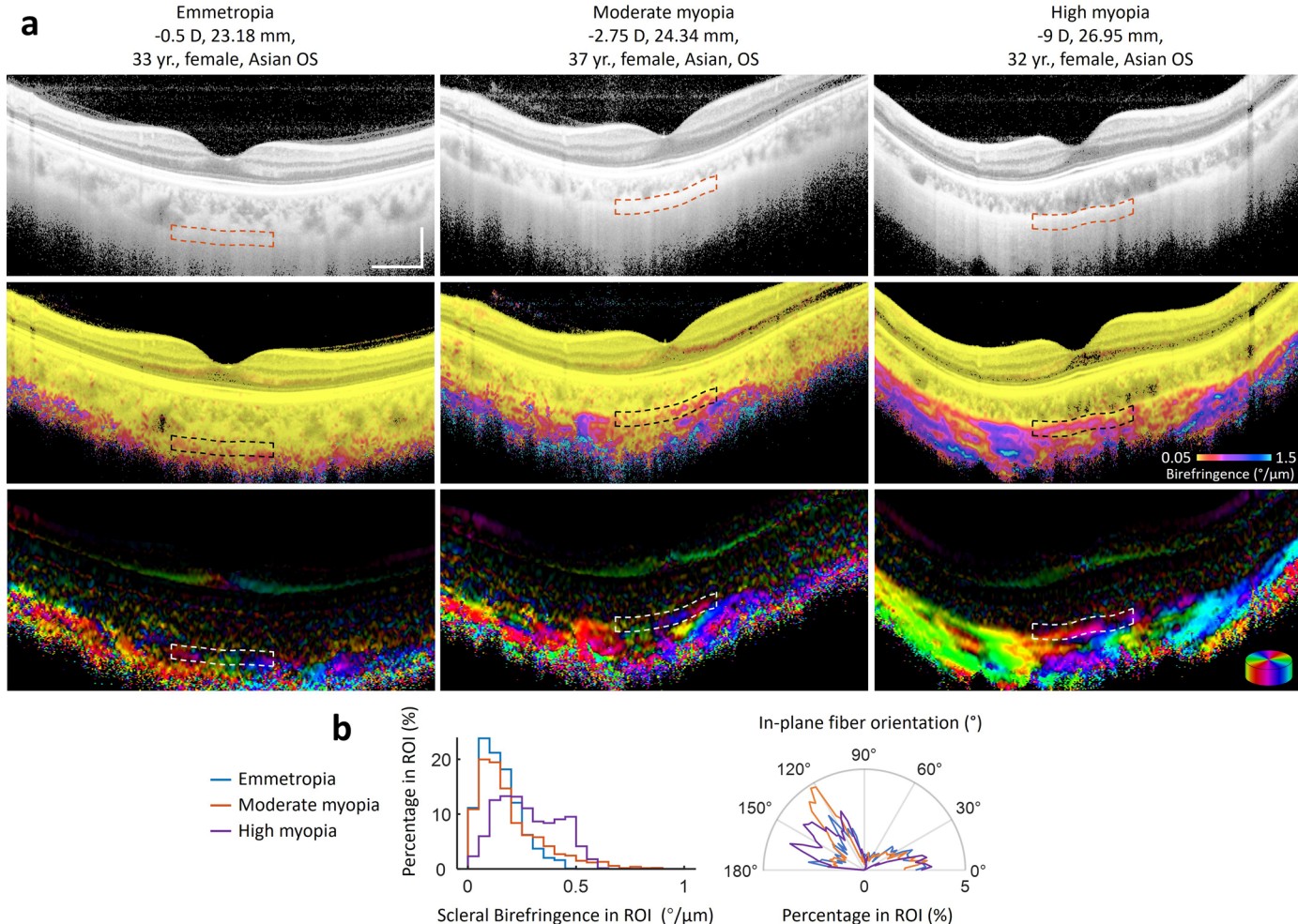

**a**

Emmetropia
-0.5 D, 23.18 mm,
33 yr., female, Asian OS

Moderate myopia
-2.75 D, 24.34 mm,
37 yr., female, Asian, OS

High myopia
-9 D, 26.95 mm,
32 yr., female, Asian OS

0.05 ■■■■■ 1.5
Birefringence (°/μm)

**b**

— Emmetropia
— Moderate myopia
— High myopia

In-plane fiber orientation (°)

**Extended Data Fig. 8 | Scleral collagen fibre orientation and birefringence of the macular region in humans, as assessed with TRIPS-OCT. a,** Cross-sectional scans of the macular region of human subjects with different degrees of myopia. **b,** Histograms of measured birefringence and collagen fibre orientation within the regions of interest (ROIs, pixel numbers n = 2032, 2554, 2079 for each region) from the cross-sectional images. The ROIs are selected as an area in the sclera below the fovea measuring 100 μm vertically and 2 mm laterally, extending from the choroidal-scleral interface. The boundaries of the ROIs are indicated by the dotted lines in the images. The choroidal-scleral interface is labelled manually.

We observe that the average scleral birefringence is higher in patients with a higher degree of myopia, as well as an increase in the local maxima and a decrease in the local minima in the angular histograms of collagen fibre orientation. Specifically, in the eye with moderate myopia, there is a reduction in fibres oriented at 150° compared to the emmetropic eye. In the eye with high myopia, the collagen fibres at 60° completely disappear in the ROI. From this observation, we suppose that the increased PSB in patients with myopia is due to a decrease of the interweavement of the collagen fibres. Histogram equalization is applied to **a** (upper panels). Scale bars, **a**, vertical: 300 μm, horizontal: 1 mm.

**Extended Data Table 1 | Univariate linear regression analysis of posterior pole scleral birefringence with multiple variables within the eyes in the emmetropia or low myopia (3 D ≤ SE< −6 D) group**

| Variable | Univariate analysis posterior pole scleral birefringence (°/μm) | | |
|---|---|---|---|
| | B (95% CI) | r (95% CI) | p |
| Age (year) | -0.0019 (-4.2x10⁻³,3x10⁻⁴) | -0.28 (-0.57,0.004) | 0.11 |
| Gender | -0.03 (-0.10,0.05) | -0.12 (-0.45, 0.24) | 0.38 |
| Refractive error (D) | -0.031 (-0.044,-0.020) | -0.60 (-0.76, -0.39) | **1.2x10⁻⁴** |
| Axial length (mm) | 0.049 (0.021, 0.076) | 0.54 (0.25, 0.75) | **0.001** |
| Choroidal thickness (μm) | -4.1x10⁻⁴ (-0.001, 3x10⁻⁴) | -0.24 (-0.54, 0.15) | 0.16 |
| Inner scleral thickness (μm) | -2.5x10⁻⁵ (-0.0018,0.0016) | -0.005 (-0.34, 0.35) | 0.49 |

Linear regression models are fitted using data from 69 eyes from 42 individuals. 95% confidence intervals of *B* and *r* are calculated by cluster bootstrapping. The *r* values are calculated by Pearson correlation. The *p* values are calculated by *F*-test against a constant model. *B*, Model gradient, *r*, Pearson's coefficient, CI, Confidence interval.

| | |
|---|---|

# Reporting Summary

## Statistics

For all statistical analyses, confirm that the following items are present in the figure legend, table legend, main text, or Methods section.

| n/a | Confirmed | |
|---|---|---|
| ☐ | ☒ | The exact sample size (*n*) for each experimental group/condition, given as a discrete number and unit of measurement |
| ☐ | ☒ | A statement on whether measurements were taken from distinct samples or whether the same sample was measured repeatedly |
| ☐ | ☒ | The statistical test(s) used AND whether they are one- or two-sided *Only common tests should be described solely by name; describe more complex techniques in the Methods section.* |
| ☒ | ☐ | A description of all covariates tested |
| ☐ | ☒ | A description of any assumptions or corrections, such as tests of normality and adjustment for multiple comparisons |
| ☐ | ☒ | A full description of the statistical parameters including central tendency (e.g. means) or other basic estimates (e.g. regression coefficient) AND variation (e.g. standard deviation) or associated estimates of uncertainty (e.g. confidence intervals) |
| ☐ | ☒ | For null hypothesis testing, the test statistic (e.g. *F*, *t*, *r*) with confidence intervals, effect sizes, degrees of freedom and *P* value noted *Give P values as exact values whenever suitable.* |
| ☒ | ☐ | For Bayesian analysis, information on the choice of priors and Markov chain Monte Carlo settings |
| ☒ | ☐ | For hierarchical and complex designs, identification of the appropriate level for tests and full reporting of outcomes |
| ☐ | ☒ | Estimates of effect sizes (e.g. Cohen's *d*, Pearson's *r*), indicating how they were calculated |

*Our web collection on statistics for biologists contains articles on many of the points above.*

## Software and code

Policy information about availability of computer code

| Data collection | NI LabVIEW 2020 was used to develope the data acquisition and control software. |
|---|---|
| Data analysis | We used MATLAB (R2019a, R2020b, R2021b) to reconstruct the TRIPS-OCT images from the photodetector readout. We used MATLAB R2021b to segment the TRIPS-OCT images, quantify the measurements, and conduct the statistical analysis. The central algorithm to reconstruct TRIPS-OCT images from triple measured Stokes vectors can be found at https://github.com/DrXinyu/TRIPS-OCT. |

For manuscripts utilizing custom algorithms or software that are central to the research but not yet described in published literature, software must be made available to editors and reviewers. We strongly encourage code deposition in a community repository (e.g. GitHub). See the Nature Portfolio guidelines for submitting code & software for further information.

## Data

Policy information about availability of data

All manuscripts must include a data availability statement. This statement should provide the following information, where applicable:

- Accession codes, unique identifiers, or web links for publicly available datasets
- A description of any restrictions on data availability
- For clinical datasets or third party data, please ensure that the statement adheres to our policy

Processed animal data (shown in Figs. 3 and 4), including en-face images and refractive errors, are available from figshare at https://doi.org/10.6084/m9.figshare.21300576. Additionally, one example guinea pig B-scan modulated by triple polarization states (shown in Fig. 1) is also available from figshare. The entire raw dataset of animal experiments is more than 25 TB in size and can be shared on request with appropriate data-transfer methods. For the clinical study, the raw data acquired during the study are available for at least 5 years from the corresponding author on reasonable request, subject to approval from the SingHealth Centralised Institutional Review Board. A request will be processed within 3 months.

# Field-specific reporting

Please select the one below that is the best fit for your research. If you are not sure, read the appropriate sections before making your selection.

☒ Life sciences ☐ Behavioural & social sciences ☐ Ecological, evolutionary & environmental sciences

For a reference copy of the document with all sections, see nature.com/documents/nr-reporting-summary-flat.pdf

# Life sciences study design

All studies must disclose on these points even when the disclosure is negative.

| | |
|---|---|
| Sample size | As this study of PSB was a pilot study, no sample-size calculation was performed for the animal experiments, owing to the lack of previous studies. Empirically we decided to use 21 guinea pigs to investigate the correlation between refractive error and scleral birefringence. In the clinical study (Figs. 5 and 6), the sample-size calculation was based on the evaluation of the correlation between refractive error and TRIPS-OCT measurements with a 90% statistical power using preliminary parameters from the longitudinal guinea-pig study. For other experiments aiming for validation of the technology of TRIPS-OCT, biological independence is not required. Therefore, one animal was used for each specific aim. |
| Data exclusions | In the longitudinal guinea-pig study (Figs. 3 and 4), no data were excluded. In the clinical study (Figs. 5 and 6), we excluded 75 eyes (47%) with suboptimal positioning (50 eyes, 31%) and insufficient signal-to-noise ratio (average scleral SNR < 4.6 dB) from the sclera (25 eyes, 16%). The exclusion of images with insufficient intensity signal was pre-established. The threshold 4.6 dB was determined after the study because no preliminary TRIPS-OCT data were available. Perfect positioning of the scan was not always guaranteed in the clinical study because the positioning of TRIPS-OCT imaging head was not always accurate, owing to the lack of pupil alignment camera, fundus camera, eye tracker, or automatic depth-positioning unit. A sufficient signal from the sclera was not always guaranteed in the clinical study because a thick choroid may limit the light penetration to the sclera. |
| Replication | For the TRIPS-OCT technology, at least 294 volume scans (42 eyes, 7 weeks) were performed in the guinea-pig eyes in the longitudinal animal-model study; 180 volume scans were performed in human eyes, from which 100 volume scans were rated as high-quality, defined by the quality criteria in the human study. For the analysis of the correlation between scleral birefringence and the degree of myopia, no replication was performed because all validated data are included to achieve maximum statistical power. |
| Randomization | No randomization was applied because the study was focused on the correlation between TRIPS-OCT measurements and the degree of myopia. |
| Blinding | For the animal study, blinding was not possible because there was no prior knowledge of the association or correlation between refractive error and scleral birefringence. In the human study, The investigators were blinded to the birefringence measurements when segmenting the images, and were blinded to myopia status of the subjects during the processing of the TRIPS-OCT measurements. |

# Reporting for specific materials, systems and methods

We require information from authors about some types of materials, experimental systems and methods used in many studies. Here, indicate whether each material, system or method listed is relevant to your study. If you are not sure if a list item applies to your research, read the appropriate section before selecting a response.

## Materials & experimental systems

| n/a | Involved in the study |
|---|---|
| ☒ | ☐ Antibodies |
| ☒ | ☐ Eukaryotic cell lines |
| ☒ | ☐ Palaeontology and archaeology |
| ☐ | ☒ Animals and other organisms |
| ☐ | ☒ Human research participants |
| ☐ | ☒ Clinical data |
| ☒ | ☐ Dual use research of concern |

## Methods

| n/a | Involved in the study |
|---|---|
| ☒ | ☐ ChIP-seq |
| ☒ | ☐ Flow cytometry |
| ☒ | ☐ MRI-based neuroimaging |

## Animals and other organisms

Policy information about studies involving animals; ARRIVE guidelines recommended for reporting animal research

| | |
|---|---|
| Laboratory animals | A 1-year-old pig (Yorkshire-Landrace cross, Male, National Large Animal Research Facility, Singapore) was euthanized for TRIPS-OCT and PLM imaging. Three guinea pigs (Elm Hill Labs, Male n = 3, Pigmented n = 1, Chelmsford, US) aged 1, 12, and 16 weeks were euthanized for TEM histology analysis. 21 guinea pigs (Elm Hill Labs, Albino n = 17, Female n = 13, Chelmsford, US) were bred on-site for refraction-development analysis. 2 guinea pigs (Elm Hill Labs, Albino n = 2, Female n = 1, Chelmsford, US) aged 1.6 and 2.1 years were selected for TRIPS-OCT imaging from a group of breeders in our animal facility. |

| Wild animals | The study did not involve wild animals. |
| --- | --- |
| Field-collected samples | The study did not involve samples collected from the field. |
| Ethics oversight | The use of animals for these studies was approved by the Institutional Animal Care and Use Committee of SingHealth (AAALAC Accredited; 2018/SHS/1441, IACUC 1290). |

Note that full information on the approval of the study protocol must also be provided in the manuscript.

# Human research participants

Policy information about studies involving human research participants

| Population characteristics | The population characteristics of each eye group used in the analysis are described in Supplementary Table 1. |
| --- | --- |
| Recruitment | 80 normal adults aged 21 years and above without any known ocular diseases were recruited. The inclusion criteria were as follows: age 21 years and above; no diabetes and free from clinically relevant eye disease that interferes with the aim of the study, including glaucoma, diabetic retinopathy, age-related macular degeneration, uveitis, or vascular occlusive diseases. 10 patients diagnosed with pathologic myopia with staphyloma were recruited. For patients with pathologic myopia, the inclusion criteria were as follows: age > 21 years; both eyes presented staphyloma under wide-field OCT imaging. The exclusion criteria were as follows: eye conditions that may potentially result in poor quality imaging scans (severe cataract, corneal haze/opacity).<br><br>Normal adults were recruited from patients, visitors and staff of the Singapore National Eye Center. Patients with pathologic myopia were referred by clinicians of the Singapore National Eye Center. As a pilot study conducted in Singapore, 91% of the participants were Asian. Participants were compensated with 30 Singapore dollars each. |
| Ethics oversight | All procedures performed were in adherence with the ethical standards of the SingHealth Centralized Institutional Review Board (CIRB Ref No. 2021/2592). Written informed consent was obtained from all participants in accordance with the Declaration of Helsinki. |

Note that full information on the approval of the study protocol must also be provided in the manuscript.

# Clinical data

Policy information about clinical studies

All manuscripts should comply with the ICMJE guidelines for publication of clinical research and a completed CONSORT checklist must be included with all submissions.

| Clinical trial registration | CIRB Ref No. 2021/2592. There is no public registration because the clinical study was not interventional. |
| --- | --- |
| Study protocol | Polarization Sensitive Optical Coherence Tomography — Phase II: A Pilot Study (R1819/61/2021). |
| Data collection | The data of patients and healthy individuals were collected between 15/02/2021 and 28/05/2021 from the research clinic in the Singapore Eye Research Institute. Healthy individuals were recruited from patients, hospital staff and visitors. Patients with pathologic myopia were recruited from general clinics in the Singapore National Eye Center and refereed by clinicians. Auto-refraction, axial length, age, sex and TRIPS-OCT data were extracted from the data-management unit of the institute. |
| Outcomes | Outcomes are not relevant because the study was not interventional. |

