## [Peer Review File · Nature Biomedical Engineering]

Posterior scleral birefringence measured by triple-input polarization-sensitive imaging as a biomarker of myopia progression

Corresponding author: Leopold Schmetterer

Editorial note

This document includes relevant written communications between the manuscript's corresponding author and the editor and reviewers of the manuscript during peer review. It includes decision letters relaying any editorial points and peer-review reports, and the authors' replies to these (under 'Rebuttal' headings). The editorial decisions are signed by the manuscript's handling editor, yet the editorial team and ultimately the journal's Chief Editor share responsibility for all decisions.

Any relevant documents attached to the decision letters are referred to as **Appendix #**, and can be found appended to this document. Any information deemed confidential has been redacted or removed. Earlier versions of the manuscript are not published, yet the originally submitted version may be available as a preprint. Because of editorial edits and changes during peer review, the published title of the paper and the title mentioned in below correspondence may differ.

Correspondence

Thu 26 May 2022

Decision on Article nBME-22-0626

Dear Prof Schmetterer,

Thank you again for submitting to *Nature Biomedical Engineering* your manuscript, "Myopia biomarker revealed by triple-input polarization-sensitive imaging of the posterior sclera". The manuscript has been seen by 4 experts, whose reports you will find at the end of this message.

You will see that the reviewers appreciate the work. However, they express concerns about the degree of support for the claims, and provide useful suggestions for improvement. We hope that with significant further work you can address the criticisms and convince the reviewers of the merits of the study. In particular, we would expect that a revised version of the manuscript provides:

- * Clarification of the contradictory differences in birefringence across the experiments with the human patients and the pigs, and of the influence of the diameter of the collagen fibres on birefringence.
- * Fairer discussion of the comparisons of TRIPS-OCT and PS-OCT, particularly with regards to sample motion, laser jitters, and the angles between the input polarization states, as requested by Reviewers #2 and #1.
- * Thorough methodological reporting, as per the pertinent comments of all reviewers.When you are ready to resubmit your manuscript, please upload the revised files, a point-by-point rebuttal to the comments from all reviewers, the reporting summary, and a cover letter that explains the main improvements included in the revision and responds to any points highlighted in this decision.

Please follow the following recommendations:

- * Clearly highlight any amendments to the text and figures to help the reviewers and editors find and understand the changes (yet keep in mind that excessive marking can hinder readability).
- * If you and your co-authors disagree with a criticism, provide the arguments to the reviewer (optionally, indicate the relevant points in the cover letter).
- * If a criticism or suggestion is not addressed, please indicate so in the rebuttal to the reviewer comments and explain the reason(s).
- * Consider including responses to any criticisms raised by more than one reviewer at the beginning of the rebuttal, in a section addressed to all reviewers.
- * The rebuttal should include the reviewer comments in point-by-point format (please note that we provide all reviewers with the reports as they appear at the end of this message).
- * Provide the rebuttal to the reviewer comments and the cover letter as separate files.

We hope that you will be able to resubmit the manuscript within 20 weeks from the receipt of this message. If this is the case, you will be protected against potential scooping. Otherwise, we will be happy to consider a revised manuscript as long as the significance of the work is not compromised by work published elsewhere or accepted for publication at *Nature Biomedical Engineering*.

We hope that you will find the referee reports helpful when revising the work. Please do not hesitate to contact me should you have any questions.

Best wishes,

Filipe

Dr Filipe Almeida
Associate Editor, Nature Biomedical Engineering

Reviewer #1 (Report for the authors (Required)):

The manuscript describes an interesting triple-input PSOCT system (TRIPS-OCT) and the corresponding reconstruction method to investigate the relationship between the myopia and the posterior scleral birefringence (PSB). The sensitivity, accuracy and robustness of the TRIPS-OCT are shown improved after the retardance, diattenuation and PMD compensation. Taking advantage of the improved performance of the TRIPS-OCT, the PSB is investigated as a potential biomarker for myopia and its pathological complications. Overall, the manuscript has merits and worth publishing. The authors may find the comments below beneficial in revising their paper.

Specific Comments:

Experiment:

1. The authors demonstrate the better birefringence sensitivity of the TRIPS-OCT system in Figs. 1g-1j. In the orange area in Fig. 1g, there are multiple dark stripes. These dark stripes contribute to the lower signals in histogram results in Fig. 1i and 1j, which are thought to be part of the noise floor in the results using the dual-input method in the manuscript. These stripes disappeared in Fig. 1h using the proposed TRIPS-OCT method, indicating that the tissues in the orange areas should be homogenous and the dark stripes in the results obtained from the dual-input method should be artifacts. The author can provide some references to show that the orange areas are the birefringent homogenous tissues and further demonstrates the superiority of the TRIPS-OCT method.

2. To demonstrate the compensation for corneal diattenuation, and retardance can improve the accuracy of the optic axis measurements, the authors compared the TRIPS-OCT results with the theoretical orientations in Fig. 1p. The results look good. However, as mentioned in the manuscript, "HFL is known distributed approximately radially around the fovea", it seems like there is not an absolute distribution of the orientations of the fibers around the fovea. So how were the theoretical orientations defined? Is there any reference to support this theoretical orientation plot? Due to the individual differences, it would be more convincing to compare the TRIPS-OCT results with the polarization microscopy results using a same sample, perhaps with ex vivo sample or phantoms.

3. It seems like there is a contradiction in conclusions between the guinea pig results and the human results. "We observed that guinea pig eyes in the myopia group showed significantly lower PSB than those in the emmetropia and hyperopia group (Lines 280-281)". In the human results, the authors found that "we observed that the myopic eye presented increased PSB, both in the outer peripapillary area and the posterior pole area. (Lines 315-316)". However, the authors didn't provide any discussions on this contradictory conclusion between the guinea pig results and the human results.

4. In Fig. 3a, with the same OD value: -2.75 D, there is a non-negligible difference in the PSB between the 2 eyes. The PSB in the eye of Guinea pig 2 is much higher than that in Guinea pig 1 even when they have the same OD value. This phenomenon is very interesting, but the authors didn't explain this phenomenon in the manuscript. The reviewer guess this may be because the myopia affected the guinea pig 1 earlier than guinea pig 2, and hence affect and hinder the growth of the scleral collagen fibers, leading to a less collagen content in guinea pig 1 compared with guinea pig 2. This results in lower PSB value in the guinea pig 1 compared with guinea pig 2. If this is true, the findings in the manuscript should be valuable for the studies of the early-stage myopia.

5. Are there any differences in the axis orientation values between the normal eyes and the myopia eyes? For example, can the authors find the remodeling of the fibers in the myopia eyes? The authors spend many efforts to provide a higher accuracy axis orientation measurement, however, there is very limited results and conclusions about the axis orientation of the eyes.

Discussion:

6. The method cited by the author (Ref. 65) is classified as the circularly polarized light-based PS-OCT. However, this method is not limited to the circularly polarized light-based PS-OCT system. As mentioned in this paper, "the requirement of known polarization states incident on the sample can be removed." This needs fixed.

Method:

7. The THIPS-OCT system requires 3 mutually orthogonal input polarization states to compute the results. In Supplementary Fig. 1g, the authors show that the angles between the 3 Stokes vectors are not exactly equal to 90 degrees due to the diattenuation of the cornea, polarization mode dispersion of the system, and measurement noise. Can the author explain the effect when the 3 input polarization states are not exactly equal to 90 degrees (For example, introducing more noises

and errors)? After the compensation of the diattenuation and the PMD of the system, what's angles between the 3 input polarization states?

8. The authors claim the TRIPS-OCT is superior to dual input OCT. However, TRIPS captures the three-input data consecutively whereas dual-input does it concurrently. This rings an issue of the system tolerability to the sample motion for TRIPS approach. This issue must be discussed in a good depth, particularly for in vivo imaging.

9. The author mentioned in the manuscript: "Favorably, the Mueller matrix reconstruction does not use the absolute phase of the OCT signal, thus TRIPS-OCT is resistant to phase noise induced by laser jitter micrometer-range sample vibration. (Lines 140-142)". It's true that the laser jitter would not affect the relative phase measurement of the OCT signals. However, the laser jitters can induce the phase shift between each scan. Can this jitter render the measurements incoherent among the 3 input polarization states?

10. In supplementary method 2, the author mentioned the effect of the depolarization. It's not clear whether the authors correct, ignore or partially correct the depolarization effect. Please explain this part clearly in the supplementary method 2. The definition of the intermediate matrix μ^{\wedge} is not clear. Does it have the same value as μ ?

Reviewer #2 (Report for the authors (Required)):

Review of Myopia biomarker revealed by triple-input polarization sensitive imaging of the posterior sclera by Xinyu Liu et al.

The authors measure scleral birefringence in a guinea pig model and in humans and correlate the birefringence with myopia.

This is a well written, interesting and extensive paper, that however has major issues.

1) The most prominent problem of the manuscript is that the pre-clinical and the clinical results are contradicting. The effect in guinea pigs and humans is opposite. In guinea pigs the birefringence decreases with increasing myopia (figure 3b, higher myopia correlates with lower birefringence), in humans the birefringence increases with increasing myopia (figure 5 i,j,k). This discrepancy is not addressed in the paper. Can this discrepancy between the animal and human model be explained? The predictive value of scleral birefringence is evaluated in guinea pigs from 2 to 8 weeks, however, at 9 months guinea pigs are full grown. How predictive is a correlation at week 8?

2) The paper is surprisingly, unnecessarily and incorrectly negative about other PS-OCT methods such as accuracy and Signal to Noise Ratio (especially depth encoded PS-OCT), and omits mentioning drawbacks of their method with respect to other PS-OCT processing methods.

3) At the same time, 75 out of 160 eyes were excluded from the study due to poor Signal to Noise Ratio.

4) Quantitative information on the instruments is lacking (like scan speed, depth resolution, and depth resolution of the birefringence information). The method to mitigate polarization mode dispersion, spectral binning, reduces the depth resolution significantly, but this is not mentioned.

5) The birefringence of the sclera is evaluated at the choroid-sclera interface, taking a 200 micron band centered on this interface. It is puzzling that to quantify the scleral birefringence, half of the signal is coming from the choroid, which is a different tissue structure. It is unclear why this choice was made, and why also incorporating the choroid characterizes the scleral birefringence better than calculating the birefringence over a band 200 micron below the choroid-sclera interface.

Page by page comments are provided below.

Page 1

'Here, we developed triple-input polarization-sensitive optical coherence tomography (TRIPS-OCT) with improved sensitivity, accuracy, and robustness compared to traditional PS-OCT'.

This claim is exaggerated. As pointed out below, triple input PS-OCT is likely better than dual input PS-OCT, at the expense of slower acquisition. With respect to depth encoded PS-OCT, the presented comparison and characterization of depth encoded PS-OCT is flawed.

Page 2, 1

'Currently, no clinical tool is available to inspect the posterior scleral collagen in-vivo.' This sentence suggests that there is no tool available at all.

However, Willemse et al. recently demonstrated for the first time that PS-OCT can characterize the scleral collagen fiber architecture. This should be changed into something like 'no commercially device is currently available to ...'

Willemse J, Gräfe MGO, Verbraak FD, de Boer JF. In Vivo 3D Determination of Peripapillary Scleral and Retinal Layer Architecture Using Polarization-Sensitive Optical Coherence Tomography. *Transl Vis Sci Technol.* 2020;9(11):21. Published 2020 Oct 19.

doi:10.1167/tvst.9.11.21

Page 2, 2

The authors state that it is difficult to image the sclera because 'the probing light is attenuated and the input polarization state is scrambled when passing through the eye³⁸'. This statement is incorrect and doesn't follow from the work presented in ref. 38. The polarization state changes due to birefringence when passing through the eye, but doesn't necessarily get scrambled.

Page 2, 3

'... the architecture of scleral collagen fibers in normal eyes was revealed³⁹, but the low scleral signal limited a further quantitative analysis of the scleral birefringence'.

This statement unfairly characterizes prior results. No quantitative analysis was performed in ref. 39, but not because of low scleral signal. With TRIPS-OCT, the scleral signal was too low for quantitative analysis in 75 out of 160 eyes and no further than 100 um under the scleral-choroidal interface was used for analysis. One could consider this a low scleral signal as well.

Page 2, 4

'The limitations in detection sensitivity and high system complexity of current PS-OCT have hampered its translation from the optical bench to the clinic.'

This is an unfair and incorrect characterization of prior work, there are no limitations in detection sensitivity and the high system complexity is exaggerated. These 'highly complex' systems have made it to the clinic. The most complex part about the system is the detection unit, and the complexity of the detection unit is equal for triple, dual and depth encoded PS-OCT. The polarization delay unit required for depth encoded PS-OCT is fairly simple compared to the detection unit that all systems have in common.

'In particular, the clinical study of using PS-OCT to image posterior scleral collagen in patients with myopia and pathologic myopia has not been achieved.'

Please rephrase. This is not related to the prior unfair and incorrect characterization, it has not been achieved because it has not been tried before.

Page 2, 5

'Current PS-OCT instruments using sequential dual-input polarization states^{40,41} assume to measure pure retardance, ignoring the negative impact of diattenuation.' This statement is not true. Diattenuation was characterized with dual-input polarization states by Park et al. (2004).

B. Hyle Park, Mark C. Pierce, Barry Cense, and Johannes F. de Boer, "Jones matrix analysis for a

polarization-sensitive optical coherence tomography system using fiber-optic components," Opt. Lett. 29, 2512-2514 (2004)

Page 2, 6

Depth-encoded PS-OCT is characterized incorrectly on multiple occasions in the manuscript. Statements such as '... [depth encoded PS-OCT] uses complex high-maintenance systems that sacrifice sensitivity and imaging range' are exaggerated and incorrect.

Page 3, 1

'As far as we know, there has been no previous clinical studies focusing on PSB in patients with myopia' should be 'As far as we know, there have been no previous clinical studies focusing on PSB in patients with myopia'.

Page 3, 2

It should be specified that polarization modulation configuration was already demonstrated by Saxer et al. (2000). Therefore, the approach is not novel. The authors build on prior work. The novelty is creating 3 orthogonal states in the Poincare sphere.

Saxer, C. E., de Boer, J. F., Park, B. H., Zhao, Y., Chen, Z., & Nelson, J. S. (2000). High-speed fiber-based polarization-sensitive optical coherence tomography of in vivo human skin. Optics letters, 25(18), 1355-1357.

Page 3, 3

'... reconstruction does not use the absolute phase of the OCT signal, thus TRIPS-OCT is resistant to phase noise induced by laser jitter and micrometer-range sample vibration'. This sentence implies that this is a problem in other PS-OCT systems, which is not the case. No prior PS-OCT system uses the absolute phase of the OCT signal. As pointed out in the earliest PS-OCT dual detector systems, the relative phase between two orthogonal polarization state interference signals is relevant. Even with depth encoded PS-OCT the absolute phase is not used. This is an inaccurate characterization of prior work.

Page 3, 4

Can the authors provide more evidence for their claim that the whole inner retina of the guinea pig exhibits the same value of low birefringence throughout all the layers? Nerves are present in the inner retina which exhibit birefringence.

Page 4

By comparing Fig. 1g (dual state PS-OCT) and Fig. 1h (TRIPS-OCT), two layers seem to appear with higher birefringence in Fig. 1g which are not present in Fig. 1h. These two reddish bands do not correspond to low intensity. Why does the dual state PS-OCT show additional layer contrast which disappears in the TRIPS-OCT image? One expects that noise does not have structure, is this then truly noise?

Page 4, 2

The approach to compare birefringence noise seems unfair if the triple-state measurement takes 1.5 times longer. This time could be used for averaging in dual-state measurements.

Page 4, 3

There are three possible combinations for the dual state measurements, (1,2), (1,3) and (2,3). Are the presented results for dual state equivalent for these three pairs? Please compare noise levels, which could depend on the chosen combination.

Page 6:

The authors claim a close agreement between TRIPS-OCT in Fig. 2d,f and PLM in Fig 2e,g. This statement seems valid enough comparing Fig 2f and 2g. However, the orientations depicted on the right side of Fig 2d and 2e seem completely different. Can the authors comment on this?

Page 7

A colorbar is missing in Figure 3a, and the SE labels are hard to read.

Page 7, 2

The effects in guinea pigs and humans is opposite. In guinea pigs the birefringence decreases with increasing myopia, in humans the birefringence increases with increasing myopia. Can this discrepancy between the animal and human model be explained?

Page 9

'Due to the requirement for a sufficient signal from the sclera, we excluded 75 eyes based on the average en-face intensity signal by a threshold of 30 dB. ' How is the dB signal from an en face image defined? It should be specified how large a fraction 75 eyes is of the total amount of eyes scanned, and that the author's criterium is rejecting a lot of data.

Page 10

Optic axis cross-sectional images seem to be displayed with a different colorwheel than en face optic axis images. While the en face images are clear and seem to correspond the provided colorwheel, the cross sectional images are/do not.

Page 10,2

It is unclear what is shown and pointed out in figure 5h. The 'choroidal-scleral interface structures resulting in the petaloid birefringence pattern'? I don't see what the black arrows are pointing at. An explanation is missing in the text. Without the supplementary material, it is unclear to the reader at this point that the en face images are based on the choroidal-scleral interface in general.

Page 12

Figure 6d is very hard to interpret. What is meant with 'the two orthogonal scans'? Isn't a scan a 3D image in itself? What is orthogonal to that? Videos are a better tool for visualizing 3D structures.

Page 13

'To control early-stage myopia, increasing outdoor activities that can be unequivocally encouraged for all children'. This sentence is grammatically incorrect.

Page 13, 2

The authors state that an increase in PSB can be explained by "a change in collagen fiber direction from circumferential to radial in the peripapillary sclera³⁰". In ref. 30, WAXS was used to measure the collagen fiber direction. However, this technique lacks depth imaging, and therefore misses the information about the thickness of the layers. If the inner layer becomes thinner for example (which is likely in the case of an elongation of the eye), the signal from the outer layer becomes dominant. Do the authors find this change in direction with their own optic axis images in the longitudinal guinea pig study? Can they confirm the results from Markov et al.?

Page 14

Whenever the round-trip retardance reaches a half wave, the local probing light is linearly polarized and can coincide with the sample's optic axis, frustrating the reconstruction of depth-resolved metrics⁶⁷. The work referenced in 67 is not appropriate for this sentence. This has been pointed out much earlier than 2012.

Page 14

'... the optimal sensitivity window along the imaging range and requires frequent expert maintenance when the images encoded at the two depths drift in position or amplitude due to environmental variation.'

This is not an issue. The delay can be determined through autocorrelation. This is done in many

depth encoded PS-OCT systems. Therefore, frequent calibration to maintain accuracy is not needed in the PDU. The authors raise a moot problem. The same inaccurate statement is repeated on page 15 'Slight alignment drift undermines the PSNR in polarization depth-encoding PS-OCT', and should be removed there as well. Maintenance/alignment of the detection unit is far more of an issue, but this is the same in both TRIPS-OCT and depth-encoded PS-OCT.

With respect to the detection unit, the authors do not consider wavelength dependent splitting in the PBS of the detection unit in their TRIPS-OCT system.

Page 14, 2

'... readily corrupted by the presence of diattenuation induced by both the system components and the sample.' In tissue, the axis of the diattenuation and birefringence often coincide, and therefore no 'corruption' occurs. This comment is questionable and lacks nuance.

Page 15

Ext. data Fig 1c shows a 10 dB better SNR for triple-input PS-OCT compared to depth-encoding. However, this 10 dB increase is not visible at all in ext. data Fig 1b. The depth-penetration in the images is not significantly different. How is this possible?

Page 15, 2

'the images at different depths are incoherently combined in polarization depth-encoding PS-OCT but coherently combined in triple input PS-OCT'. This statement is incorrect, and should be the other way around. Images at different depths are coherently combined in depth-encoding PS-OCT but incoherently combined in triple input PS-OCT.

Page 15, 3

The comparison of TRIPS-OCT and depth encoded PS-OCT is flawed and should be removed from the manuscript. The images do not show a 10 dB difference in SNR. It is unclear what leads to the 10dB SNR difference, but there is no argument rooted in physics that can explain this. The statements in this paragraph are confusing and incorrect ('the images at different depths are incoherently combined in polarization depth-encoding PS-OCT but coherently combined in triple input PS-OCT'), which makes it difficult to pinpoint the problem. It is unclear how averaging of different images was handled and how it might have affected SNR. It might be that the background level in the second depth encoded PS-state suffers from a significantly higher background due to cross talk from the first depth encoded PS-state. But then the difference is due to a flawed implementation of depth encoded PS-OCT.

Page 16

Authors should repeat the comparison between dual-input and triple-input for all three combinations of two input states, i.e., (1,2) (2,3) and (1,3) and show that this produces the same results for Fig 1. The effect of this calculation should propagate to Figure 1F to calculate the noise background.

Page 19

The authors should specify which wavelength (range) and repetition rate was used in this setup. They should also specify the depth range and axial and radial resolutions.

Page 21,1

For guinea pig a maximum birefringence projection is used. Maximum projections are often unreliable and highly influenced by noise. An average or median projection should be used.

Page 21,2

Why are human en face birefringence images from only 200 um around the choroidal-scleral interface, instead of the sclera? Half of the 'scleral birefringence' is therefore taken from the choroid? No myopia-specific changes are expected in the choroid. If the authors can argue why the

choroidal-scleral interface should be used, it should be specified throughout the manuscript that this is the actual biomarker. I.e., 'scleral birefringence' should be replaced with 'scleral-choroidal interface birefringence'. Is this small slab that is used for the en face images also used for the comparisons?

Page 26

In Supplementary Fig. 1g, the authors show a variation of 10 degrees for the orthogonality of the input states, which is surprisingly large. "The reasons for the errors are probably the diattenuation of the cornea, polarization mode dispersion of the system, and measurement noise." Can the authors also present this graph after compensation for diattenuation and PMD? The polarizing beam splitter in the detection unit also affects the spectral dependence of the orthogonality, which is not taken into account.

Page 27

It is unclear what is meant with the 'deterministic component' μ' in supplementary method 2. 'Eq. 6 imposes that the G-product of any Stokes vector pair is unitary'. This sentence is unclear, and H is not unitary? After this sentence, the rest of the section is even more difficult to follow and the notations are unclear (e.g. what is meant by $c-mcn\ \mu nT\ G\ \mu m?$). Please clarify.

Page 30

"... and the depth-resolved physical optic axis can be solved." Do the authors mean the absolute orientation of the optical axis? This is impossible, since the circular birefringence of the system, the cornea and the vitreous remains unknown. Please change into '... and the depth-resolved relative optic axis can be solved'.

Page 30, 2

The authors claim that the original measured Mueller matrix, L , does not fulfil the symmetry constraint due to the circulator only. This is also due to non-reciprocally traversed optical fiber. Isn't the constraint that is utilized here that the measured circular birefringence of the sample should be zero, a constraint also pointed out by Park et al. (2005).

Park, B. H., Pierce, M. C., Cense, B., & De Boer, J. F. (2005). Optic axis determination accuracy for fiber-based polarization-sensitive optical coherence tomography. *Optics letters*, 30(19), 2587-2589.

Page 32

'Spectral binning to correct PMD was proposed ...' should be changed into 'Spectral binning to mitigate PMD was proposed ...'.

Spectral binning does not correct for PMD. As Zhang et al have pointed out, to correct for PMD more than one measurement should be used.

Ellen Ziyi Zhang, Wang-Yuhl Oh, Martin L. Villiger, Liang Chen, Brett E. Bouma, and Benjamin J. Vakoc, "Numerical compensation of system polarization mode dispersion in polarization-sensitive optical coherence tomography," *Opt. Express* 21, 1163-1180 (2013)

It should be pointed out that with spectral binning the depth resolution is significantly reduced.

Page 32,2

The second equation of the Supplementary Method 2 indicates that Mueller matrices of each bin (with a depth resolution x times poorer than the original OCT image) are averaged into a general Mueller matrix. What is the depth-resolution of the birefringence? It should be mentioned in the main text what it is and that it is x times poorer than the original OCT resolution.

Page 34

The approach presented in Supplementary Method 5 seems very interesting. Corneal birefringence is corrected in a frame-based manner. Would it also be possible to correct in 2D? x -direction as well as y -direction? Would you be able to create an image of the corneal phase retardation?

Page 34, 2

The authors should make it clearer that circular birefringence was not corrected. ‘... the single pass cornea birefringence ...’ should be ‘the single pass cornea linear birefringence’. Circular birefringence of the cornea cannot be measured.

Page 34,3

‘The Mueller matrices on the surface were extracted, denoted as $S(x)$, which is a function of the lateral position. To reconstruct the single pass cornea birefringence, $C(x)$, the differential Mueller matrix $s(x)$ was calculated by taking the matrix logarithm of $S(x)$.’

What are the input Stokes vectors that evolve to the Stokes vectors from the surface to calculate the Mueller matrix or differential Mueller matrix? Is an identity matrix assumed here (after corrections) as input?

No page associated

The link provided for the central algorithm to reconstruct TRIPS-OCT images is invalid.

The authors flip en face images left-right, without providing a reason. It is more intuitive to display the en face images in the same orientation as regular fundus photos, and have the B-scans match this orientation.

Figures are cluttered with at least 15 subpanels. If the only purpose of Fig. 1d is to illustrate an angle of 27.37° , it can be left out. Fig. 1b is the same as Supplementary Fig. 1a, and Fig. 1e is similar to Supplementary Fig. 1d.

Could the authors also add a line to the en face images in the supplementary material video 2 to indicate which B-scan is shown? It looks as if the B-scan is taken vertically. En face images should

Reviewer #3 (Report for the authors (Required)):

This is a very interesting study on a promising new methodology in ophthalmology with special reference to myopia.

1. Line 29: “Evaluating the risk of progression of myopia is critical for optimizing the administration and timing of interventions.”: There are no interventions available to prevent the progression of high myopia?

2. Line 56: “For end-stage myopia, the posterior scleral reinforcement (PSR) surgery, including macular buckling, is a clinically approved therapy to strengthen the posterior sclera and arrest the continued elongation of the eye^{14,15}”: These procedures are not generally approved.

3. Line 70: “and is the main” may be re-worded to “and may be one of the main”.

4. Line 115: Typo: “there has been” may be “there have been”.

5. Line 133: The abbreviation of SNR should be explained

6. Line 127ff: The article may be written either past tense or in present tense

7. Line 157: “known distributed“ may be better “known to be distributed”?

8. Did the possibility to visualize the sclera differ between parapapillary regions (with beta, gamma and delta zone and other fundus regions with an intact retinal pigment epithelium)?

9. Was the technique dependent on axial length and ethnic background (i.e., pigmentation of the fundus)?
10. Line 211: “animal euthanasia“ may be better „after sacrificing the animal“?
11. Line 223: physiological eye growth may be different of myopic axial elongation?
12. Line 251: “refraction development“ may be better “development of refractive error“?
13. Line 413: The term “alarming“ may be dropped

Reviewer #4 (Report for the authors (Required)):

Line 98, The reviewer suggests using “three-phase” to replace the word “triple-input” and “triple state” throughout the manuscript. Please consider abbreviating the technology as “3P-PS-OCT”.

Line 100, Change to “...., assume that the measurements contain only the pure retardance.....”

Line 133, Change to “..longer penetration depth..”

Line 134-143, Please remove the descriptions as they are not results.

Line 145-152, The noise floor is not equal to sensitivity. Please show that the signal doesn't attenuate in these two systems in this situation.

Line 154, Please call the method on the compensation of corneal diattenuation and retardance.

Line 160, The orientation of HFL changes in 3-D near the fovea. The reviewer doesn't understand how the authors calculated the theoretical orientation of HFL.

Line 230, Could the authors explain why larger collagen fiber diameter corresponds to increased birefringence?

Fig. 3, How was the metric of posterior scleral birefringence in panel b calculated from the en face images in panel a?

Line 152, As the authors indicated that the collagen fiber diameter can affect the scleral birefringence and it changes significantly with age. To verify that PBS is correlated with myopia development, please provide results on a control group to exclude the effect of age in the correlation.

Line 318, Please provide the distribution of PSB with age in the groups.
Please provide histology validations of scleral remodeling for myopia development in guinea pigs.

Please rewrite the Discussion to remove any redundant descriptions. The authors should focus on the explanation of larger PBS in myopia, the new insight of in vivo data added to previous ex vivo and histology studies, the limitation, and the prospect of this new method for better diagnosis of myopia.

Line 527-531, Please move it into Discussion.

Line 514, why the lower boundary of scleral is not visible in the B-scan images of Ext.

Data Fig. 1 panel d.

The value of this work should stand in claiming that PBS is a better biomarker for myopia than the refractive error. However, the reviewer believes the authors failed on proving this. The reviewer suggests shrinking the description on correlation comparison but highlighting the prediction of myopia from earlier PBS data, especially in human eyes.

Tue 29 Nov 2022

Decision on Article nBME-22-0626A

Dear Prof Schmetterer,

Thank you for your revised manuscript, "Myopia biomarker revealed by triple-input polarization-sensitive imaging of the posterior sclera", which has been seen by the original reviewers. In their reports, which you will find at the end of this message, you will see that the reviewers acknowledge the improvements to the work and reviewer #2 raises a few additional technical criticisms that we hope you will be able to address.

As before, when you are ready to resubmit your manuscript, please upload the revised files, a point-by-point rebuttal to the comments from all reviewers, the reporting summary, and a cover letter that explains the main improvements included in the revision and responds to any points highlighted in this decision.

As a reminder, please follow the following recommendations:

- * Clearly highlight any amendments to the text and figures to help the reviewers and editors find and understand the changes (yet keep in mind that excessive marking can hinder readability).
- * If you and your co-authors disagree with a criticism, provide the arguments to the reviewer (optionally, indicate the relevant points in the cover letter).
- * If a criticism or suggestion is not addressed, please indicate so in the rebuttal to the reviewer comments and explain the reason(s).
- * Consider including responses to any criticisms raised by more than one reviewer at the beginning of the rebuttal, in a section addressed to all reviewers.
- * The rebuttal should include the reviewer comments in point-by-point format (please note that we provide all reviewers with the reports as they appear at the end of this message).
- * Provide the rebuttal to the reviewer comments and the cover letter as separate files.

We hope that you will be able to resubmit the manuscript within 12 weeks from the receipt of this message. If this is the case, you will be protected against potential scooping. Otherwise, we will be happy to consider a revised manuscript as long as the significance of the work is not compromised by work published elsewhere or accepted for publication at *Nature Biomedical Engineering*.

We look forward to receive a further revised version of the work. Please do not hesitate to contact me should you have any questions.

Best wishes,

Filipe

Dr Filipe Almeida
Associate Editor, Nature Biomedical Engineering

Reviewer #1 (Report for the authors (Required)):

The revision has addressed my concerns satisfactory. I have no further comments. The study is important which may have high clinical significance if proved useful in further clinical trials.

Reviewer #2 (Report for the authors (Required)):

Review of Myopia biomarker revealed by triple-input polarization sensitive imaging of the posterior sclera by Xinyu Liu et al.

The manuscript has improved considerably, but still contains a few inaccurate statements and raises a few other questions.

The authors have addressed the discrepancy between the guinea pig (2-8 weeks) and human measurements by adding one adult guinea pig (n=1) with myopia and one adult guinea pig (n=1) without myopia.

From the abstract:

“In a guinea pig study spanning 8 weeks, we found that scleral birefringence is significantly correlated with the development of refractive errors and predicts the onset of myopia. In a human cross-sectional study, we found that scleral birefringence is correlated with myopia status and is associated with scleral pathological changes.”

- Please clarify that an opposite correlation was found in humans and the guinea pig study spanning 8 weeks.

From Supplementary Discussion 3:

“In our study, for safety consideration, we did not allow for repetitive scans more than two times in one patient during one visit.”

- Was the reason the incident power on the retina, and what was the power used incident on the eye?

Page 2:

triple-input PS-OCT (TRIPS-OCT), a novel modulation and reconstruction strategy for PS-OCT that increases imaging sensitivity, accuracy, and system robustness.

- Sensitivity improvement by a factor of 1.33 (See below). Is this improvement significant? Otherwise remove claim.

Depth-encoding PS-OCT^{44–46} is able to measure diattenuation but compromises the optimal ranging window by encoding two orthogonal input states along depth. TRIPS-OCT rigorously measures diattenuation and corrects for depolarization while maintaining the simplicity of dual-input systems.

- Depth-encoding PS-OCT does not compromise the optical ranging window, but utilizes the available ranging window to multiplex polarization input states. On the other hand, TRIPS-OCT compromises speed (3 times slower than Depth-encoding PS-OCT).

TRIPS-OCT rigorously measures diattenuation.

- Suggest to remove rigorously, unnecessary hyperbole. Depth-encoding PS-OCT also “rigorously” measures diattenuation

Page 3:

We also demonstrate that TRIPS-OCT outperforms depth-encoding PS-OCT in terms of ranging depth and laser jitter tolerance.

- Laser jitter model is unrealistic. See below. Suggest to remove this statement

Page 3:

In this comparison, we ensured that the signal-to-noise ratios (SNRs) and the sampling time of the original signals used by the two methods were identical (Ext. Data Figs. 2a-c).

- But still the dual state is 1.5 times faster than the triple state measurement. Has this been taken into account in the signal-to-noise ratios (SNRs)? For a proper comparison, in dual state the sweep time of the laser needs to be 1.5 times slower than in triple state.

The standard deviation of the noise, or noise floor, of TRIPS-OCT was 48% lower than that of the conventional dual-input method (Fig. 1c, Ext. Data Figs. 2d, e), corresponding to a 2-fold improvement in the birefringence sensitivity.

- But the acquisition time was 1.5 times longer, so the net gain is $2/1.5 = 1.33$, not a factor of 2

The edge-artifact is a dominant source of birefringence noise that is induced by a shift of the point-spread-functions in the two eigen-polarization states of the preceding system components or tissue layers (Supplementary Discussion 1). Because this shift leads to apparent diattenuation, the resulting artifacts are markedly suppressed by correctly accounting for diattenuation in the Mueller matrices of the sample.

- The authors argue that a shift of the point-spread-functions occur due to the preceding system components or tissue layers. For the system components this would be Polarization Mode Dispersion, which they mitigate, and for the tissue layers the authors argued elsewhere that the preceding layers in the guinea pig are not birefringent.

More likely a polarization dependent scattering cross section combined with an overall jump in the scattering cross section of tissue structures plays a role in the edge-artifacts, which the authors did not model.

Correcting for retardance and diattenuation results in fiber orientations that closely follow the anticipated profile.

- Even triple state PS-OCT can only measure a relative angle, not an absolute angle, so the absolute angle is calibrated on henly 's fiber direction. Please state this more clearly.

Page 13

To enable clinical PSB measurements, we developed TRIPS-OCT and improved the sensitivity, accuracy, and robustness of birefringence imaging compared to traditional PS-OCT, thereby mitigating previous obstacles towards its clinical translation.

- Authors mean "compared to traditional dual input state PS-OCT"

Page 14:

Depth-encoding PS-OCT offers reliable depth-resolved measurements independent of the sample but compromises the optimal sensitivity window along the imaging range and requires autocorrelation calibration when the images encoded at the two depths drift in position or amplitude due to environmental variation.

- Depth-encoding PS-OCT does not compromise the optimal sensitivity window. Previously the authors stated that Depth-encoding PS-OCT compromises the optical ranging window. Depth-encoding PS-OCT does not compromise the optical ranging window, but utilizes the available ranging window to multiplex polarization input states. On the other hand, TRIPS-OCT compromises speed (3 times slower than Depth-encoding PS-OCT).

The dual-input EOM-based PS-OCT is robust to environmental variation, but it assumes measurements to represent pure retardance, hence, is readily impacted by the presence of diattenuation and edge-artifacts induced by both the system components and the sample. TRIPS-OCT does not have the aforementioned limitations but requires a longer acquisition time because of triple repetitive scans at the same location of the sample.

- Drift of the polarization state in the fibers before the polarizer in the EOM will also lead to environmental variations, which will require (auto) calibration.

Page 14:

TRIPS-OCT is not sensitive to sample motion.

- Axial sample motion on the order of tens of microns between three sequentially acquired depth profiles will affect TRIPS-OCT. Autocalibration methods need to be implemented to realign in depth the three sequentially acquired depth profiles to have the Stokes vectors of corresponding tissue structures overlap.

Page 14:

Overall, TRIPS-OCT mitigates some drawbacks of traditional PS-OCT implementations and makes it more suitable for clinical translation.

- Authors mean traditional dual state PS-OCT implementations

Page 18 figure 3:

e, Numerical model of the dependency of birefringence noise and laser jitter. Laser jitter is modelled by a random shift of the simulated fringes in the fringe domain. The random shift, modelled by a Gaussian distribution, induces a relative phase variation between the images encoded in shallow and deep channels of a depth-encoding PS-OCT system.

- The model the authors use for laser jitter consists of a catastrophic breakdown of the depth encoded method, i.e., significant shifts in the acquired data (several missed k-clocks at the start). This is an unrealistic model. This comparison should be removed from the paper.

Page 34:

These edge-artifacts cannot be removed by kernel averaging as they are associated with intensity variation, which is the information carrier of the sample structure.

- The artifact manifests itself on a depth scale of 10 micron (sup fig 1b), from 172 to 182 micron in depth. This creates a wiggle in the stokes vector evolution (fig 1c) over a depth trace of 10 micron in depth. However the depth resolution is 30 micron, as stated earlier. How come the full width at half max is about 6 micron in this figure?

Reviewer #3 (Report for the authors (Required)):

All comments have satisfactorily been addressed.

Reviewer #4 (Report for the authors (Required)):

The authors have addressed all of my comments very well.

Tue 03 Jan 2023

Decision on Article nBME-22-0626B

Dear Prof Schmetterer,

Thank you for your revised manuscript, "Myopia biomarker revealed by triple-input polarization-sensitive imaging of the posterior sclera", which has been seen by Reviewers #2. In the report, which you will find at the end of this message, you will see that the reviewer acknowledges the improvements to the work and raise a few additional technical criticisms that we hope you will be able to address.

As before, when you are ready to resubmit your manuscript, please upload the revised files, a point-by-point rebuttal to the comments from all reviewers, the reporting summary, and a cover letter that explains the main improvements included in the revision and responds to any points highlighted in this decision.

As a reminder, please follow the following recommendations:

- * Clearly highlight any amendments to the text and figures to help the reviewers and editors find and understand the changes (yet keep in mind that excessive marking can hinder readability).
- * If you and your co-authors disagree with a criticism, provide the arguments to the reviewer (optionally, indicate the relevant points in the cover letter).
- * If a criticism or suggestion is not addressed, please indicate so in the rebuttal to the reviewer comments and explain the reason(s).
- * Consider including responses to any criticisms raised by more than one reviewer at the beginning of the rebuttal, in a section addressed to all reviewers.
- * The rebuttal should include the reviewer comments in point-by-point format (please note that we provide all reviewers will the reports as they appear at the end of this message).
- * Provide the rebuttal to the reviewer comments and the cover letter as separate files.

We hope that you will be able to resubmit the manuscript within 12 weeks from the receipt of this message. If this is the case, you will be protected against potential scooping. Otherwise, we will be happy to consider a revised manuscript as long as the significance of the work is not compromised by work published elsewhere or accepted for publication at *Nature Biomedical Engineering*.

We look forward to receive a further revised version of the work. Please do not hesitate to contact me should you have any questions.

Best wishes,

Filipe

Dr Filipe Almeida
Associate Editor, Nature Biomedical Engineering

Reviewer #2 (Report for the authors (Required)):

Review of Myopia biomarker revealed by triple-input polarization sensitive imaging of the posterior sclera by Xinyu Liu et al.

The manuscript has improved considerably, but still contains a few inaccurate statements. To this reviewer, the arguments about superiority of triple input state PS-OCT over all other PS-OCT implementations are not convincing and distract from the main message of the paper, namely the myopia results and the suppression of edge artifacts by triple input state versus dual input state PS-OCT.

A) The noise analysis comparing dual input OCT with Triple input OCT is still incorrect. A triple input state measurement with the system of the authors takes 1.5 times longer than a dual input OCT measurement. A measurement that takes 1.5 times longer should have a SNR improvement by a factor of 1.5. Similarly, a measurement in the same amount of time with 1.5 times more sample arm power should have a SNR improvement by a factor of 1.5.

To address this difference in the acquisition time of the two methods, the authors propose to average three (3) dual state input OCT images and two (2) triple input state OCT images. The authors assume that through this averaging process the SNR of the dual state image is improved by a factor of 3 and the triple input state image is improved by a factor of 2. Thus, according to the authors, the birefringence results calculated after this averaging method can be compared, since the time difference for the two acquisition methods has been compensated for. However, in the averaging method chosen by the authors, the SNR of the images does not improve by a factor of 3 by averaging three dual state input images, but by a factor of square root of 3, nor does the SNR improve by a factor of 2 by averaging two triple state images, but by a factor of square root of 2.

In order to achieve the SNR improvement of a factor of 3 for the dual state images, the fringe data of three images need to be averaged (known as coherent averaging, averaging the complex depth profiles before taking the square to obtain an intensity image). The same argument holds for the triple state images.

It is clear to this reviewer that triple input state OCT reduces the edge artifacts as compared to dual input state, and thus is an important improvement. However, the presented quantification in terms of birefringence sensitivity improvement by a factor of 2 is incorrect. Based on the presented analysis, the improvement is a factor of 1.63

As a minor point, figure Ext Data 2 c also indicates that something is wrong. To the understanding of the reviewer, figure b ("Averaged intensity B-scans used for comparing the two methods") shows 2 and 3 fold averaging for triple and dual input state images, respectively. The authors argue that figure c (based on the images in b) proves that the SNR of the images are the same. However, the SNR of the 3 times averaged dual input state image should be 1.5 times better than the SNR of the 2 times averaged triple input state image according to the reasoning of the authors to compensate for the difference in required acquisition time of the two

methods for a fair comparison.

B) For the comparison with Depth Encoded PS-OCT the authors do not choose the best phase jitter performance reported in the literature (state of the art) but choose to restrict the comparison to a jitter model without a k-clock device. This results in a restricted comparison. For a previously reported phase jitter of 0.3 degrees (as acknowledged by the authors in the figure caption), both methods perform equally well. In addition, the manuscript does not provide enough information of the simulation to be able to reproduce the results of Ext data figure 3e. It is recommended to remove Ext data figure 3e

It is recommended to remove the following statement on line 615:

Laser jitter affects the Jones matrix-based method because the relative phase between the two columns of the Jones matrix is used in the reconstruction.

The depth range of swept sources has significantly improved over the last 10 years. Depth ranges of 12 millimeter with negligible sensitivity roll-off are currently available (e.g. Thorlabs SL100060).

It is recommended to modify the following statement on line 510 and further:

Depth-encoding PS-OCT offers reliable depth resolved measurements independent of the sample but compromises the optimal sensitivity window because the image encoded in the deeper channel suffers from a sensitivity loss (usually around 3 dB which depends on the encoding depth, digitization rate and laser linewidth) due to the sensitivity roll-off.

It is unclear to this reviewer why the authors are so adamant about stating that triple input OCT is superior to depth encoded PS-OCT. The authors do not provide convincing arguments and have to resort to restricted implementations (a jitter model without a k-clock device) and out of date depth range capabilities to make this point. These statements should be removed from the manuscript. The authors show that triple state input OCT is superior to dual state input OCT. This is a nice result.

Fri 20 Jan 2023

Decision on Article nBME-22-0626C

Dear Prof Schmetterer,

Thank you for your revised manuscript, "Myopia biomarker revealed by triple-input polarization-sensitive imaging of the posterior sclera". Having consulted with the Reviewers #2 (whose comments you will find at the end of this message), I am pleased to write that we shall be happy to publish the manuscript in *Nature Biomedical Engineering*.

We will be performing detailed checks on your manuscript, and in due course will send you a checklist detailing our editorial and formatting requirements. You will need to follow these instructions before you upload the final manuscript files.

Best wishes,

Filipe

Dr Filipe Almeida
Associate Editor, Nature Biomedical Engineering

Reviewer #2 (Report for the authors (Required)):

I am pleased with the latest revision. have no further comments.

The implementation of Triple State PS-OCT is clever (Polarization state incident at 27.37 degrees on optical axis of the modulator to generate three orthogonal states in the poincare sphere) and leads to a suppression of the edge artefacts, compared to dual state PS-OCT.

Rebuttal 1

We sincerely thank Dr. Almeida for his efforts in gathering these reviews and providing us the opportunity to improve this manuscript. We also greatly appreciate all the insightful comments and feedback from the reviewers. Below, we have summarized the major changes, provided a list of new experiments, simulations, and analyses, and included a point-by-point response to the reviewers with revisions to the manuscript colored as orange in the responses.

Summary of major changes for all reviewers

1. To understand the “contradictory” results between the animal model and humans, we have additionally imaged adult guinea pigs with high myopia, analyzed the birefringence changes related to fiber orientation in myopic eyes in guinea pigs and humans, and discussed the results in detail.

In the revised manuscript, a paragraph has been added to the Discussion section.

Comparing the correlations between PSB and the degree of myopia in the young guinea pigs and adult humans, the PSB decreased in animals but increased in adult humans with the severity of the disease. We additionally imaged two adult guinea pigs without and with high myopia (**Ext. Data Fig. 7**) and found the PSB was higher in the myopic eye, which is consistent with the data obtained in humans. We suppose that there is a fundamental difference between the scleral changes in the young guinea pig model and adult humans representing different stages of myopia development within the lifespan. Our data indicate that the early eye growth at younger age and the myopic elongation in adults lead to opposing effects on PSB related to scleral collagen fiber arrangement and diameter. To understand the difference between these two stages of myopia development, we analyzed the collagen fiber orientation and birefringence within the regions of interest in guinea pigs and humans. In the young guinea pig at the ages of 2 and 8 weeks (**Ext. Data Fig. 6**), the average scleral birefringence increased with age, in conjunction with an increase in the interweavement of the collagen fiber as evidenced by a reduction in the local maxima in the angular histograms of the fiber orientation. In the adult guinea pigs, the average birefringence was higher in the myopic eye, however, in conjunction with a reduction in the interweavement of the collagen fibers (**Ext. Data Fig. 7**). The latter phenomenon was also observed in the sub-macular sclera of adult humans (**Ext. Data Fig. 8**). As such increased PSB in young guinea pigs may indicate the enlargement of scleral fiber diameter, associated with increased scleral collagen synthesis to achieve the required scleral stiffness during the eye growth. In contrast, increased PSB in adult guinea pigs and humans with myopia may indicate alterations in the arrangement of scleral collagen and reductions in interwoven fibers, associated with an extended elongation of the eyeball. However, these suppositions are solely based on TRIPS-OCT measurements and still require further histological investigations.

In the revised manuscript, three extended data figures (**Ext. Data Figs. 6,7,8**) reporting the analysis of birefringence changes related to fiber orientation have been added.

Ext. Data Fig. 6 | Scleral collagen fiber orientation and birefringence in a young guinea pig at ages of 2 and 8 weeks as assessed with TRIPS-OCT. a, Images of a guinea pig eye at ages of 2 weeks and 8 weeks, respectively. **b,** Boxplots of measured birefringence and angular histograms of measured collagen fiber orientation in the entire sclera (left panel), region of interest 1 (ROI 1) (middle panel) and ROI 2 (right panel) from the cross-sectional images. *En-face* intensity images are obtained from an average projection along the depth. *En-face* optic axis images are obtained from the inner layer of the sclera after flattening the images using the surface of the retina. White dotted lines indicate the locations of the cross-sectional images. The sclera is manually segmented in the cross-sectional images, and two regions of interest (ROIs) are selected in each image for localized comparison. ROIs are registered by choroidal vessel patterns, indicated in **a** by yellow arrows. Comparing the measurements at ages of 2 and 8 weeks, in the entire sclera and ROI1, the average scleral birefringence increases with aging, whereas the distribution of collagen fiber orientation broadens as evidenced by a reduction of the local maxima and an increase of the local minima in the angular histograms. Broadening of the fiber orientation distribution may be due to an increase in fiber interweavement, the increase in the interwoven fiber diameters, or a combined effect of both. Locally in ROI 2, we observe that the maximum scleral birefringence increases while the median decreases with aging. Considering the collagen arrangement is highly variable and localized in the sclera, at locations where fibers are interwoven, the birefringence change with increased fiber diameter will be bidirectional and dependent on the collagen microstructures.

Ext. Data Fig. 7 | Scleral collagen fiber orientation and birefringence in adult guinea pigs without and with myopia as assessed with TRIPS-OCT. a, Images of two adult guinea pig eyes without and with myopia. **b,** Boxplots of measured birefringence and angular histograms of measured collagen fiber orientation in the entire sclera (left panel), ROI 1 (middle panel) and ROI 2 (right panel) from the cross-sectional images. The cross-sectional image locations are roughly matched by the same relative position according to the optical nerve head. From the cross-sectional images, we observe thinning of the sclera tissue and deformation of the eye shape towards axial elongation in the highly myopic eye (-9D). Comparing the birefringence measurements, the average scleral birefringence increases in the myopic eye. The interweavement of collagen fiber, however, decreases as evidenced by that the local maxima in the angular histograms of the myopic eye are higher as compared to the emmetropic eye. As a result of scleral collagen remodeling, the increase in PSB in adult guinea pigs with myopia may be due augmented collagen fiber alignment and the reduction of interwoven fibers. Scale bars, a, vertical: 300 μm , horizontal: 1 mm.

Ext. Data Fig. 8 | Scleral collagen fiber orientation and birefringence of the macular region in humans as assessed with TRIPS-OCT. **a**, Cross-sectional scans of the macular region of human subjects with different degrees of myopia. **b**, Boxplots of measured birefringence and angular histograms of measured collagen fiber orientation within the regions of interest (ROIs) from the cross-sectional images. The ROIs are selected as an area in the sclera below the fovea measuring 100 μm vertically and 2 mm laterally, extending from the choroidal-scleral interface. The boundaries of the ROIs are indicated by the dotted lines in the images. The choroidal-scleral interface is labelled manually. We observe that the average scleral birefringence is higher in patients with a higher degree of myopia, as well as an increase in the local maxima and a decrease in the local minima in the angular histograms of collagen fiber orientation. Specifically, in the eye with moderate myopia, there is a reduction in fibers oriented at 150° compared to the emmetropic eye. In the eye with high myopia, the collagen fibers at 60° completely disappear in the ROI. From this observation, we suppose that the increased PSB in patients with myopia is due to a decrease of the interweavement of the collagen fibers. Scale bars, **a**, vertical: 300 μm , horizontal: 1 mm.

In the revised manuscript, a paragraph reporting the adult animal model has been added into the Methods section.

Adult guinea pig model. Two albino guinea pigs (Elm Hill Labs, Chelmsford, US) were selected for TRIPS-OCT imaging from a group of breeders in our animal facility. The selection criteria for these animals were as follows: 1) older than 1.5 years; 2) with clear and healthy eyes without any evidence of anterior segment or retinal changes; and 3) emmetropia ($\text{SE} = 0\text{ D}$) or high myopia ($\text{SE} \leq -6\text{ D}$). Retinoscopy was performed on cycloplegic eyes in alert animals. TRIPS-OCT imaging was performed on the eyes meeting these inclusion criteria.

In the revised manuscript, we edited the Introduction to mention the difference in the results of model animals and humans.

...Next, in our human cross-sectional study, we found a strong, albeit negative, correlation between PSB and myopia status within eyes with emmetropia (normal vision) and low myopia (69 eyes, $-6\text{ D} < \text{SE} \leq 3\text{ D}$)....

2. To compare different PS-OCT reconstruction methods, we have numerically simulated the reconstruction methods and compared the noise performance regarding the different noise sources. Specifically, we have

pointed out that the intensity SNR is a fundamental limitation of the sensitivity in birefringence imaging. Furthermore, if the intensity SNR is sufficiently high, in dual-input and single-input reconstruction methods where only retardance is assumed within the sample, the dominate noise source is the edge-artifacts.

In the revised manuscript, an extended figure (Ext. Data Fig. 3) has been added to the manuscript.

Ext. Data Fig. 3 | Birefringence noise sources in different PS-OCT methods. **a**, TRIPS and dual-input reconstruction methods on simulated OCT scans of a four-layer sample. Black arrows indicate the so-called edge-artifacts (**Supplementary Discussion 1**) that are present in the dual-input reconstruction method but are absent in TRIPS-OCT reconstruction. The artifacts are pronounced at locations where there are brick-wall-jumps in the scattering profile. **b**, TRIPS and dual-input reconstruction methods on an A-scan of a guinea pig retina *in-vivo*. The A-scan is taken from the B-scans in **Fig. 1a**. Black arrows indicate the observed edge-artifacts. **c**, Numerical model of the edge-artifacts and its dependency on variations in the intensity profile. The sample is modeled by two nonbirefringent scattering layers under a transparent birefringent layer with random optic axis and a retardance of 20° . Intensity variation is created by changing the reflectivity of the scatterers within each layer. **d**, Numerical model of the dependency of birefringence noise and the intensity signal-to-noise ratio. The sample is modeled by a nonbirefringent scattering layer. White noise is added to the simulated fringes to create different SNRs. **e**, Numerical model of the dependency of birefringence noise and laser jitter. Laser jitter is modelled by a random shift of the simulated fringes in the fringe domain. The random shift, modelled by a Gaussian distribution, induces a relative phase variation between the images encoded in shallow and deep channels of a depth-encoding PS-OCT system. **f**, Numerical model of the dependency of birefringence noise and the axial motion. Axial motion is modelled by a random shift of the sample in the depth. A Gaussian window centered at 1060 nm with a width of 80 nm at half maximum is used to simulate the spectral shape of the swept-source laser. The OCT scan is simulated by generating the fringes in wavenumber domain of the individual scatterers and then transforming the summed fringes into the depth domain using Fourier transformation. Scattering layers are modelled by scatterers embedded within nonbirefringent or birefringent media, creating fully developed speckle patterns in OCT scans. Sixteen scans with independent speckle patterns are averaged to suppress the speckle noise before proceeding to birefringence reconstruction. Birefringence reconstruction (**Supplementary Method 2**) is performed by TRIPS, dual-input geometric reasoning, Jones matrix and single-input geometric reasoning. Birefringence noise is characterized by the standard deviation within the scattering layer(s). These plots are created by 500 repetitive simulations with random positioning of the scatterers. The error bar is defined as the 95% confidence interval. SNR, signal-to-noise ratio. From these simulations, we find that birefringence sensitivity is fundamentally limited by the intensity SNR in all reconstruction methods. With a sufficient intensity signal ($\text{SNR} > 10$ dB), birefringence noise is dominated by the edge-artifacts if pure retardance is assumed in the reconstruction method. Laser jitter affects the Jones matrix-based method because the relative phase between the two columns of the Jones matrix is used in the reconstruction. Sample motion does not affect any of the reconstruction methods including TRIPS-OCT because the Stokes vectors rather than the phases of repetitive scans are used.

In the revised supplementary information, we added a Supplementary Discussion section to further clarify the origin of edge-artifacts.

Supplementary Discussion 1: Edge-artifacts in local retardance reconstruction

The edge-artifact is a fundamental noise source of polarization sensitive detection in PS-OCT. The polarization states of light backscattered from the sample are recorded by the relative amplitude and phase between two detection channels principally configured at the two orthogonal outputs of a polarized beam splitter. Considering the axial scan of a particle with the size smaller than the axial resolution, ideally, the point-spread functions (PSFs) of the particle in the two detection channels should perfectly align. However, due to the existence of birefringence in the superficial layers of the sample (including the cornea and optical components in the instrument), the optical path lengths are slightly different because of the difference in refractive indices along the two optical paths. The shift in PSFs creates a relative amplitude difference which not truly reflects the polarization states but is an artifact.

Supplementary Discussion Fig. 1 | Simulated edge-dependent artifacts in local retardance reconstruction. **a**, Sample structure used in the PS-OCT simulation. **b**, Reconstructed sample profiles of intensity, cumulative retardance and local retardance. **c**, The trace of polarization state evolution of light backscattered from the sample.

To illustrate the edge-artifacts, we simulated the axial scan of a particle embedded in a weakly-scattering, birefringent medium and reconstructed the cumulative and local retardance of the sample. Specifically, we simulated the axial scans of a 300-μm-thick sample made of a weakly scattering birefringent medium ($\Delta n = 0.0015$) with one highly scattering particle embedded in the middle of the medium (**Supplementary Discussion Fig. 1a**). In this simulation, circularly polarized light was used as the input state. The optic axis of the sample was set as 0° (aligned with the horizontal channel of the detection system). OCT scans were simulated by generating the fringes in the wavenumber domain of the individual scatterers and then transforming the summed fringes into the depth domain using Fourier transformation. To suppress the speckle noise, we averaged 1000 simulated A-lines with random speckle patterns and obtained the depth profile and the cumulative and local retardance of the sample (**Supplementary Discussion Fig 1b**). In the profile of the highly scattering particle, a shift in PSFs between the two polarization sensitive detection channels can be seen. In this simulation, the length of shift was determined by the cumulative retardance of the superficial layer ($\Delta n \times 150 \mu\text{m}$). Cumulative retardance was calculated as the rotation angle from the input state to the measured polarization states on the Poincaré sphere and local retardance was calculated as the local rotation angle of the Stokes vectors within 1-μm section along the depth. The theoretical evolution trace of the polarization state should locate in the UV-plane on the Poincaré sphere; but due to the PSF shift, measured polarization states deviate from the theoretical trace, creating artifacts in reconstructed retardance.

The derivative nature of the local retardance profile makes it more sensitive to these artifacts, and thus, the edge-artifacts are more obvious in the local retardance profile.

These edge-artifacts are also related to the absolute polarization state of the light, the eigenstates of the superficial sample, and the direction of the detection channels. These artifacts are more obvious when the PSFs of the two channels are comparable in amplitude. In particular, for a specific polarization state where the backscattered light is directed to only one detection channel, no edge-artifacts would exist in the reconstruction.

These edge-artifacts cannot be removed by kernel averaging as they are associated with intensity variation, which is the information carrier of the sample structure. Interestingly, these artifacts, induced by unwanted amplitude differences of the shifted PSFs in two polarization channels, are physically behaving like sample diattenuation, and therefore, the artifacts can be suppressed by removing the sample diattenuation component from the Mueller matrix using polar decomposition when reconstructing the local birefringence properties.

In the revised manuscript, the details about the numerical simulations are reported in Supplementary Method 2.

Supplementary Method 2: Dual-input, Jones matrix, and single-input birefringence reconstruction methods

We numerically simulate different PS-OCT reconstruction methods that are adopted from previous reports¹²⁻¹⁴. The procedures are summarised in **Supplementary Method Fig. 1**.

Supplementary Method Fig. 1 | Diagram of local retardance reconstruction methods including dual-input geometric reasoning, single-input geometric reasoning, TRIPS proposed in this study, and Jones matrix method used in a depth-encoding system.

$F_{Hn}(x, k)$ and $F_{Vn}(x, k)$ represent the fringes detected from two polarization diverse channels in a B-scan. x is the lateral position, k is wavenumber, and d is depth position. n represents the index of repetitive scans in a sample location, where $n = 1$ in a single-input PS-OCT, $n = 1, 2$ in a dual-input PS-OCT, and $n = 1, 2, 3$ in a TRIPS-OCT. In a depth-encoding PS-OCT, $n = 1$ and D represents the delay length between the two orthogonal polarization imaging ranges. FFT represents Fourier transform. $*$ represents the conjugate. \otimes represents the Kronecker

product. $\mathcal{L} = \begin{bmatrix} 1 & 0 & 0 & 1 \\ 1 & 0 & 0 & -1 \\ 0 & 1 & 1 & 0 \\ 0 & i & -i & 0 \end{bmatrix}$. Ω is the constraint of the polarization state evolution¹³. In the geometric reasoning

methods, the local polarization states and apparent optic axis are projected to the surface of a Poincare sphere. Over a local depth dz , the polarization state is rotating about the apparent optic axis, and the local retardance is

the rotation angle within the dz range. In the TRIPS and Jones matrix methods, the differential Mueller matrix $\mathbf{a}(x, z)$ can be calculated by matrix logarithm of the local Mueller matrix $\mathbf{m}(x, z)$, where $\mathbf{a}(x, z) = \ln[\mathbf{m}(x, z)]$. The non-depolarization component is evaluated by calculating the G-antisymmetric part of $\mathbf{a}(x, z)$, where $\mathbf{b}(x, z) = [\mathbf{a}(x, z) - \mathbf{G}\mathbf{a}^T(x, z)\mathbf{G}]/2$, and $\mathbf{G} = \text{diag}(1, -1, -1, -1)$. Local retardance is retrieved from the norm of the vector $[b_{3,4} \ b_{4,2} \ b_{2,3}]$, where $b_{i,j}$ is i, j entry of \mathbf{b} .

In the revised manuscript, we have discussed the limitations of TRIPS-OCT regarding longer acquisition duration and motion sensitivity.

TRIPS-OCT does not have the aforementioned limitations but requires a longer acquisition time because of triple repetitive scans at the same location of the sample. However, due to the development of faster lasers and OCT-angiography techniques⁶⁸, repetitive scanning has become a standard in current ophthalmic imaging. TRIPS-OCT is not sensitive to sample motion, as the measurements of modulated frames are based on Stokes vectors and the phase variation between repetitive scans does not influence the reconstruction. Overall, TRIPS-OCT mitigates some drawbacks of traditional PS-OCT implementations and makes it more suitable for clinical translation.

In the revised manuscript, we have removed the previous comparison on SNR between depth-encoding PS-OCT and TRIPS. Although in our hands, TRIPS-OCT performed significantly better in terms of sensitivity, we agree with the comments that the performance of a depth-encoding PS-OCT may involve various engineering parameters including the digitization rate, the laser scanning speed, the laser line width and the polarization encoding depth. We clarify in our revised manuscript that as the intensity SNR is fundamentally limiting the sensitivity of birefringence imaging, the deep channel of a depth-encoding system is reducing the intensity SNR and thereby compromising the birefringence sensitivity. Furthermore, TRIPS-OCT provides twice the ranging depth as compared to the depth encoding system.

3. We have added the reporting of detailed parameters of the system in the Methods Section. We have rewritten the Supplementary Methods 1 and 2 to clarify the details of Mueller matrix reconstruction and the numerical simulations.

In the revised manuscript, the following sentences have been added to the Methods section.

A swept-source optical coherence tomography (SS-OCT) system previously developed by our group⁶⁹ was modified to achieve TRIPS-OCT. The OCT system employed a swept-source laser (1060 nm, sweep rate 200 kHz, tuning range 100 nm, Axsun Technologies Inc, Billerica). The measured axial full-width-at-half-maximum of the intensity point-spread-function (PSF) in air was 6.1 μm . The digitizing rate was 1 GHz, and the measured 3 dB roll-off ranging depth was 3.5 mm. The beam size entering the pupil was 1.2 mm, corresponding to an optical lateral resolution of 25 μm and a depth of focus of 4.6 mm in a normal human eye. The spatial averaging in the birefringence reconstruction, including filtering of Stokes vectors and spectral binning, reduced the resolution to 150 μm in the lateral and 30 μm in the axial directions for birefringence imaging.

The Mueller matrix reconstruction algorithm has been described in the revised Supplementary Method 1. In addition, the Matlab implementation of the algorithm has been uploaded to <https://github.com/DrXinyu/TRIPS-OCT>.

Supplementary Method 1: Reconstruction of a pure Mueller matrix in TRIPS-OCT

In our TRIPS-OCT system, the detected interferometric fringes undergo 3 steps to reconstruct the cumulative Mueller matrix, including (1) filtering in Stokes domain, (2) minimizing the depolarization effect, and last, (3) solving a pure Mueller matrix.

1. Filtering in Stokes domain

Considering the fringes $F_{Hn}(x, k)$ and $F_{Vn}(x, k)$ of one Bscan from two polarization diverse detection channels, complex tomograms $E_{Hn}(x, z)$ and $E_{Vn}(x, z)$ can be obtained by Fourier transformation of the fringes from k -domain to z -domain, where x represent the lateral scanning position and z represents the depth. n represents the index of repetitive scans in one location and $n = 1, 2, 3$. Stokes vectors of the image pixels can be reconstructed by

$$\mathbf{S}_n(x, z) = \begin{bmatrix} |E_{Hn}(x, z)|^2 + |E_{Vn}(x, z)|^2 \\ |E_{Hn}(x, z)|^2 - |E_{Vn}(x, z)|^2 \\ 2\text{Re}(E_{Hn}(x, z)E_{Vn}^*(x, z)) \\ -2\text{Im}(E_{Hn}(x, z)E_{Vn}^*(x, z)) \end{bmatrix} \dots (1)$$

* represents the complex conjugate. To suppress noise, spatial filtering was performed on Stokes vectors by convolving the components of Stokes vectors with a filter kernel \mathbf{k} , where $\tilde{\mathbf{S}}_n(x, z) = \mathbf{S}_n(x, z) \otimes \mathbf{k}$.

In the TRIPS-OCT system, three sets of Stokes vectors are acquired, reconstructed, and filtered, noted as $\tilde{\mathbf{S}}_1(x, z), \tilde{\mathbf{S}}_2(x, z), \tilde{\mathbf{S}}_3(x, z)$. Without loss of generality, the input probing light with three orthogonal input states before going through the system and sample can be assembled into a probing matrix \mathbf{s} .

$$\mathbf{s} = \begin{bmatrix} 1 & 1 & 1 \\ 1 & 0 & 0 \\ 0 & 1 & 0 \\ 0 & 0 & 1 \end{bmatrix} \dots (2)$$

The reconstructed Stokes vectors corresponding to the three input states can be assembled into a 4x3 detection matrix $\boldsymbol{\mu}(x, z) = [\tilde{\mathbf{S}}_1(x, z) \quad \tilde{\mathbf{S}}_2(x, z) \quad \tilde{\mathbf{S}}_3(x, z)]$. Using a Mueller matrix to describe the entire optical system and sample from the probing matrix to the detection,

$$\boldsymbol{\mu}(x, z) = \mathbf{M}(x, z)\mathbf{s} \dots (3)$$

$\mathbf{M}(x, z)$ is the unknown Mueller matrix. To simplify the notation, we omit the pixel coordinate (x, z) in the following steps, as the operations are applied indiscriminately to every pixel in the image.

2. Minimizing the depolarization effect

A general full Mueller matrix can be decomposed as

$$\mathbf{M} = \mathbf{D}\mathbf{L} \dots (4)$$

where \mathbf{L} is a pure Mueller matrix, \mathbf{D} is a general depolarization matrix. A pure Mueller matrix, or Jones-derived Mueller matrix, belongs to the subset of physically admissible Mueller matrices describing a non-depolarizing medium.

In an OCT system, the detected interferometric signal arises only from the component of the backscattered light that is coherently mixed with the reference light. An individual measurement is by definition fully polarized and non-depolarized; therefore, the entire optical system and sample can be described by a pure Mueller matrix. In this ideal case, \mathbf{D} should be an identity matrix and \mathbf{M} should be in the form of \mathbf{L} . However, spatial filtering to the Stokes vectors is equivalent to introducing depolarization^{9,10}. The induced depolarization, either from noise or truly depolarizing structures, undermines the pure Mueller form of \mathbf{M} .

Full determination of the underlying Muller matrix \mathbf{M} would require the measurement its 16 (constrained) parameters. In TRIPS-OCT, an approximate solution is found by fitting a pure Mueller matrix \mathbf{L} to the available measurements $\boldsymbol{\mu}$. Specifically, we attempt to find the \mathbf{L} that results in observed polarization states $\boldsymbol{\mu}' = \mathbf{L}\mathbf{s}$, which align with the polarized component of the actually measured states.

To this end, we first polarize the observed states $\boldsymbol{\mu}$ by replacing their filtered i components with the Euclidean norms of the q, u, v components by $\tilde{t}_n = \sqrt{\tilde{q}_n^2 + \tilde{u}_n^2 + \tilde{v}_n^2}$, where $\tilde{\mathbf{S}}_n = [\tilde{t}_n \quad \tilde{q}_n \quad \tilde{u}_n \quad \tilde{v}_n]^T$ to obtain $\tilde{\boldsymbol{\mu}}$ by

$$\tilde{\boldsymbol{\mu}} = \begin{bmatrix} \tilde{t}_1 & \tilde{t}_2 & \tilde{t}_3 \\ \tilde{q}_1 & \tilde{q}_2 & \tilde{q}_3 \\ \tilde{u}_1 & \tilde{u}_2 & \tilde{u}_3 \\ \tilde{v}_1 & \tilde{v}_2 & \tilde{v}_3 \end{bmatrix} \dots (5)$$

Second, a necessary condition of pure Mueller matrix¹¹ is that $\mathbf{L}^T\mathbf{G}\mathbf{L} = c^2\mathbf{G}$, where \mathbf{G} is a diagonal matrix defined as $\text{diag}(1, -1, -1, -1)$. c is a scalar. Therefore, $\boldsymbol{\mu}'$ should satisfy that

$$\boldsymbol{\mu}'^T\mathbf{G}\boldsymbol{\mu}' = \mathbf{s}^T\mathbf{L}^T\mathbf{G}\mathbf{L}\mathbf{s} = c^2\mathbf{s}^T\mathbf{G}\mathbf{s} = c^2\mathbf{H} \dots (6)$$

where $\mathbf{H} = \begin{bmatrix} 0 & 1 & 1 \\ 1 & 0 & 1 \\ 1 & 1 & 0 \end{bmatrix}$. For the observed and polarized polarization states $\tilde{\boldsymbol{\mu}}$ to meet this condition, they have to be rebalanced with a 3x3 diagonal matrix $\mathbf{A} = \text{diag}(a_1, a_2, a_3)$ to obtain $\boldsymbol{\mu}' = \tilde{\boldsymbol{\mu}}\mathbf{A}$. Plugging this into eq. (6) gives

$$\mathbf{A}^T\tilde{\boldsymbol{\mu}}^T\mathbf{G}\tilde{\boldsymbol{\mu}}\mathbf{A} = c^2\mathbf{H} \dots (7)$$

Although this matrix equation contains 9 sub-equations, we find that the number of constraints is 4, which allows for exact solutions of c, a_1, a_2, a_3 :

$$c = \frac{1}{3}(\tilde{i}_1\tilde{i}_2 + \tilde{i}_1\tilde{i}_3 + \tilde{i}_2\tilde{i}_3 - \tilde{q}_1\tilde{q}_2 - \tilde{q}_1\tilde{q}_3 - \tilde{q}_2\tilde{q}_3 - \tilde{u}_1\tilde{u}_2 - \tilde{u}_1\tilde{u}_3 - \tilde{u}_2\tilde{u}_3 - \tilde{v}_1\tilde{v}_2 - \tilde{v}_1\tilde{v}_3 - \tilde{v}_2\tilde{v}_3) \dots (8)$$

$$a_1 = c \frac{(\tilde{i}_1\tilde{i}_2 - \tilde{q}_1\tilde{q}_2 - \tilde{u}_1\tilde{u}_2 - \tilde{v}_1\tilde{v}_2)(\tilde{i}_1\tilde{i}_3 - \tilde{q}_1\tilde{q}_3 - \tilde{u}_1\tilde{u}_3 - \tilde{v}_1\tilde{v}_3)}{\tilde{i}_2\tilde{i}_3 - \tilde{q}_2\tilde{q}_3 - \tilde{u}_2\tilde{u}_3 - \tilde{v}_2\tilde{v}_3} \dots (9)$$

$$a_2 = c \frac{(\tilde{i}_1\tilde{i}_2 - \tilde{q}_1\tilde{q}_2 - \tilde{u}_1\tilde{u}_2 - \tilde{v}_1\tilde{v}_2)(\tilde{i}_2\tilde{i}_3 - \tilde{q}_2\tilde{q}_3 - \tilde{u}_2\tilde{u}_3 - \tilde{v}_2\tilde{v}_3)}{\tilde{i}_1\tilde{i}_3 - \tilde{q}_1\tilde{q}_3 - \tilde{u}_1\tilde{u}_3 - \tilde{v}_1\tilde{v}_3} \dots (10)$$

$$a_3 = c \frac{(\tilde{i}_2\tilde{i}_3 - \tilde{q}_2\tilde{q}_3 - \tilde{u}_2\tilde{u}_3 - \tilde{v}_2\tilde{v}_3)(\tilde{i}_1\tilde{i}_3 - \tilde{q}_1\tilde{q}_3 - \tilde{u}_1\tilde{u}_3 - \tilde{v}_1\tilde{v}_3)}{\tilde{i}_1\tilde{i}_2 - \tilde{q}_1\tilde{q}_2 - \tilde{u}_1\tilde{u}_2 - \tilde{v}_1\tilde{v}_2} \dots (11)$$

$\tilde{i}_n, \tilde{q}_n, \tilde{u}_n, \tilde{v}_n$ are the elements of $\tilde{\boldsymbol{\mu}}$. Thereby, $\boldsymbol{\mu}'$ is reconstructed and the scaling factor c is solved.

3. Solving the pure Mueller matrix

A pure Mueller matrix \mathbf{L} can be modelled as a combination of a retarder \mathbf{L}_R and a diattenuator \mathbf{L}_D , both with unitary determinant.

$$\mathbf{L} = c\mathbf{L}_R\mathbf{L}_D \dots (12)$$

\mathbf{L}_R is a general retarder in the form of $\mathbf{L}_R = \begin{bmatrix} 1 & 0 \\ 0 & \mathbf{m}_R \end{bmatrix}$, constrained by conditions of $\mathbf{m}_R \in SO(3)$, and $\det(\mathbf{L}_R) = 1$;

\mathbf{L}_D is a general diattenuation in the form of $\mathbf{L}_D = \begin{bmatrix} \cosh(a) & \sinh(a) \cdot \mathbf{a}^T \\ \sinh(a) \cdot \mathbf{a} & \mathbf{a} \cdot \mathbf{a}^T (\cosh(a) - 1) + \mathbf{I} \end{bmatrix}$, equally with $\det(\mathbf{L}_D) = 1$ and $|\mathbf{a}|_2 = 1$, where \mathbf{I} represents the 3-by-3 identity matrix, and a is a scalar. c is the overall scaler solved in eq (8).

To solve \mathbf{L} from $\boldsymbol{\mu}'$, we have that

$$\boldsymbol{\mu}' = \mathbf{L}\mathbf{s} = c \cdot \begin{bmatrix} \cosh(a) + \sinh(a) \cdot \mathbf{a}^T \\ \mathbf{m}_R \cdot [\sinh(a) \cdot \mathbf{a} + \mathbf{a} \cdot \mathbf{a}^T (\cosh(a) - 1) + \mathbf{I}] \end{bmatrix} \dots (13)$$

To simplify the expression, we denote $\boldsymbol{\mu}' = \begin{bmatrix} \mathbf{i}' \\ \mathbf{m} \end{bmatrix}$, which is the solved fitted detection matrix. By observing eq. (13), we find that

$$\cosh(a) = \frac{\sum \mathbf{i}' - \det(\mathbf{m})}{2} \dots (14)$$

Hence, the vector \mathbf{a} is solved by

$$\mathbf{a}^T = \frac{\mathbf{i}' - \cosh(a)}{\sinh(a)} \dots (15)$$

\mathbf{m}_R is then solved by

$$\mathbf{m}_R = \mathbf{m}\mathbf{m}_D^{-1} \dots (16)$$

where \mathbf{m}_D is defined as

$$\mathbf{m}_D = [\sinh(a) \mathbf{a} \quad \sinh(a) \mathbf{a} \quad \sinh(a) \mathbf{a}] + \mathbf{a}\mathbf{a}^T (\cosh(a) - 1) + \mathbf{I} \dots (17)$$

To conclude, the 4x4 pure Mueller matrix \mathbf{L} is reconstructed by

$$\mathbf{L} = \mathbf{L}_R \cdot \mathbf{L}_D = c \begin{bmatrix} 1 & 0 \\ 0 & \mathbf{m}_R \end{bmatrix} \begin{bmatrix} \cosh(a) & \sinh(a) \cdot \mathbf{a}^T \\ \sinh(a) \cdot \mathbf{a} & \mathbf{a} \cdot \mathbf{a}^T (\cosh(a) - 1) + \mathbf{I} \end{bmatrix} \dots (18)$$

Response for individual reviewers

Reviewer #1 (Report for the authors (Required)):

The manuscript describes an interesting triple-input PSOCT system (TRIPS-OCT) and the corresponding reconstruction method to investigate the relationship between the myopia and the posterior scleral birefringence (PSB). The sensitivity, accuracy and robustness of the TRIPS-OCT are shown improved after the retardance, diattenuation and PMD compensation. Taking advantage of the improved performance of the TRIPS-OCT, the PSB is investigated as a potential biomarker for myopia and its pathological complications. Overall, the manuscript has merits and worth publishing. The authors may find the comments below beneficial in revising their paper.

We thank the reviewer for the valuable comments and feedback. Please evaluate our improvements to the manuscript in the following point-to-point response.

Specific Comments:

Experiment:

1. The authors demonstrate the better birefringence sensitivity of the TRIPS-OCT system in Figs. 1g-1j. In the orange area in Fig. 1g, there are multiple dark stripes. These dark stripes contribute to the lower signals in histogram results in Fig. 1i and 1j, which are thought to be part of the noise floor in the results using the dual-input method in the manuscript. These stripes disappeared in Fig. 1h using the proposed TRIPS-OCT method, indicating that the tissues in the orange areas should be homogenous and the dark stripes in the results obtained from the dual-input method should be artifacts. The author can provide some references to show that the orange areas are the birefringent homogenous tissues and further demonstrates the superiority of the TRIPS-OCT method.

We understand the concern of the reviewer that the stripes in Fig. 1i may be real structure, and thus undermine the characterization of noise between dual-input and TRIPS. With a significant improvement by further analysis, simulation and discussion, we have shown strong evidence that the “stripes” are caused by the edge-artifacts. In the revised manuscript, we have clarified the origin of the artifacts in revised Ext. Data Figs. 2a-c and Supplementary Discussion 1. Please find more information in the list of major changes for all reviewers.

We have added a paragraph in the Results section to clarify the dominant noise source in the dual-input reconstruction.

We observed that the dual-input reconstruction method suffers from edge-artifacts that are associated with variations in the sample scattering profile (Ext. Data Figs. 3a, b). The edge-artifact is a dominant source of birefringence noise that is induced by a shift of the point-spread-functions in the two eigen-polarization states of the preceding system components or tissue layers (Supplementary Discussion 1). Because this shift leads to apparent diattenuation, the resulting artifacts are markedly suppressed by correctly accounting for diattenuation in the Mueller matrices of the sample. TRIPS-OCT isolates sample birefringence properties from the Mueller matrices, and thus is almost free from edge-artifacts (Ext. Data Fig. 3c).

In addition, we have added a sentence to the caption of Fig. 1f to clarify.

... The reddish stripes in the inner retina that are present in the dual-input reconstruction are induced by edge-artifacts (Ext. Data Figs. 3a-b). Of note the artifacts disappear in the TRIPS reconstruction....

2. To demonstrate the compensation for corneal diattenuation, and retardance can improve the accuracy of the optic axis measurements, the authors compared the TRIPS-OCT results with the theoretical orientations in Fig. 1p. The results look good. However, as mentioned in the manuscript, “HFL is known distributed approximately radially around the fovea”, it seems like there is not an absolute distribution of the orientations of the fibers around the fovea. So how were the theoretical orientations defined? Is there any reference to support this theoretical orientation plot? Due to the individual differences, it would be more convincing to compare the TRIPS-OCT results with the polarization microscopy results using a same sample, perhaps with ex vivo sample or phantoms.

To address this comment, first, we have added the images of an acrylonitrile butadiene styrene (ABS) phantom to validate the depth resolved optic axis reconstruction. Second, we analyzed the error pattern induced by uncompensated corneal retardance and diattenuation. Third, we added reference of previous reports of HFL measurement and changed the “theoretical orientation” to “angular location”. Considering all these results, we pointed out that the measurement difference between dual-input and TRIPS is supposed to be induced by corneal retardance. Accurate measurements can be obtained by a proper correction for the cornea birefringence.

In the revised supplementary information, we have added a section reporting the validation of TRIPS-OCT using a phantom.

Supplementary Method 7: Validation of depth-resolved optic axis measurement using a phantom

Supplementary Method Fig. 6 | Validation of depth-resolved optic axis measurement using a phantom. a, TRIPS-OCT depth-resolved optic axis imaging on a phantom. **b,** True phantom architecture. **c,** Angular histograms of measured optic axis orientation in multiple regions of interest (ROIs) from the cross-sectional images indicated in **a**. Scale bars, **a**, vertical: 250 μm , horizontal: 1 mm.

We used a phantom to validate the depth-resolved optic axis reconstruction using TRIPS-OCT (**Supplementary Method Fig. 6a**). The phantom contains three layers of 250- μm acrylonitrile butadiene styrene (ABS) sheet (8586K171, Beidge, McMASTER-CARR, CA, US). Homogeneous birefringence with a uniform optic axis was present in the material due to the extrusion manufacture process. The anisotropic orientation of the polymer structure, noted as α , can be measured under TRIPS-OCT as the optic axis orientation. A phantom²³ with a designed architecture (**Supplementary Method Fig. 6b**) can be assembled by cutting the ABS sheet into small pieces. From the angular histograms (**Supplementary Method Fig. 6c**), in the first layer, the standard deviation of orientation measurement is 4.6 $^\circ$; in the third layer, the standard deviation of orientation measurement is 11.5 $^\circ$. Overall, from the measurement of the third layer of the phantom, TRIPS-OCT demonstrates a good performance of depth-resolved optic axis measurement independent of superficial sample structures.

In the revised Results section, we separately evaluated the correction for corneal retardance and corneal diattenuation and concluded that the measurement error of optic axis was induced mainly by corneal retardance.

To test whether compensation for corneal retardance and diattenuation improves the accuracy of optic axis measurements, we scanned Henle’s fiber layer (HFL) in the retina of a healthy volunteer (32 years, male, OD) (**Fig. 1d**). The in-plane orientation of the HFL is approximately radially distributed around the fovea⁵². We used the

orientation of the HFL to assess the accuracy of the optic axis measurements. We extracted the 2-dimensional corneal retardance and diattenuation maps from the surface of retina (**Supplementary Method 6**). In the optic axis images of the HFL, the measured orientations were plotted against the angular location on a circle centered on the foveal pit (**Fig. 1e**). Without corneal correction, the recovered fiber orientation deviates significantly from the assumed radial profile. Correcting for retardance and diattenuation results in fiber orientations that closely follow the anticipated profile. After applying the correction for corneal retardance and diattenuation, the mean error of the measurement, characterized by the difference between the measured optic axis orientation and the assumed orientation, was 35% lower than that of the uncorrected results (**Fig. 1f**).

d. Two-dimensional correction of corneal retardance and diattenuation for a volume scan. The *en-face* intensity image (upper left) from a healthy human subject (32 yr., male, OD, Asian) was rendered from a volume scan of the posterior eye. The corresponding corneal retardance (upper middle) and diattenuation (upper right) maps were extracted from the retinal surface, and the optic axis images were reconstructed without (lower left) and with the correction for corneal retardance (lower middle) and for both retardance and diattenuation (lower right). The position of the fovea is indicated by a white arrow. The magnitude of diattenuation D is defined as the relative difference between the maximum p_1^2 and minimum p_2^2 attenuation coefficients, where $D = (p_1^2 - p_2^2)/(p_1^2 + p_2^2)$. Zoomed-in images indicated by white boxes highlight the Henle fiber layer (HFL). **e.** Measured in-plane HFL fiber orientation against angular location on a circle (indicated by white dotted circles in **d**) centered on the fovea with an eccentricity of 2° . **f.** Optic axis measurement error without/with corneal correction. Error bars indicate the standard deviation.

52. Cense, B. et al. Henle fiber layer phase retardation measured with polarization-sensitive optical coherence tomography. *Biomed. Opt. Express* 4, 2296–306 (2013).

3. It seems like there is a contradiction in conclusions between the guinea pig results and the human results. “We observed that guinea pig eyes in the myopia group showed significantly lower PSB than those in the emmetropia and hyperopia group (Lines 280-281)”. In the human results, the authors found that “we observed that the myopic eye presented increased PSB, both in the outer peripapillary area and the posterior pole area. (Lines 315-316)”. However, the authors didn’t provide any discussions on this contradictory conclusion between the guinea pig results and the human results.

This comment has been discussed in the major change list for all reviewers. Briefly, the contradictory results represent different stages of myopia development in a lifespan (the physiological eye growth versus the myopic elongation) and the mechanisms underlying PSB increase are different. We have added new experiments, analysis (Ext. Data Fig. 6-8), and discussion to clarify this issue. Please refer to the major change list for all reviewers for comprehensive information.

4. In Fig. 3a, with the same OD value: -2.75 D, there is a non-negligible difference in the PSB between the 2 eyes. The PSB in the eye of Guinea pig 2 is much higher than that in Guinea pig 1 even when they have the same OD value. This phenomenon is very interesting, but the authors didn’t explain this phenomenon in the manuscript. The reviewer guess this may be because the myopia affected the guinea pig 1 earlier than guinea pig 2, and hence affect and hinder the growth of the scleral collagen fibers, leading to a less collagen content in

guinea pig 1 compared with guinea pig 2. This results in lower PSB value in the guinea pig 1 compared with guinea pig 2. If this is true, the findings in the manuscript should be valuable for the studies of the early-stage myopia.

We agree with the comment that this observation is extremely interesting. Combined with the results in Fig. 4, our results may be explained by the following cause-and-consequence relationship. With the same SE values, the PSB is different among the animals due to the different levels of collagen synthesis, which are regulated by the vision-to-sclera signals controlling the ocular growth. Higher PSB indicates higher level of collagen synthesis and a more effective regulation of vision to scleral growth, thereby reducing the susceptibility to myopia. In this way, we can explain that the lower PSB value in guinea pig 1 predicts future myopia development in guinea pig 1, whereas in guinea pig 2, no myopia develops.

We have revised the discussion section to clarify this hypothesis.

In our guinea pig model, we showed that the development of refractive errors from the ages of 2 to 8 weeks was correlated with PSB. Myopia onset at the ages of 4 and 8 weeks can be predicted by PSB measured at 2 weeks. Various studies have demonstrated that the establishment of refraction is controlled through the modulation of scleral extracellular matrix (ECM) growth and remodeling⁵⁷. Specifically, in response to myopiagenic visual signals (i.e., defocus), the activity of scleral fibroblasts, chondrocytes, and myofibroblasts, regulated by gene expression or biochemical factors, is altered, sequentially altering ECM synthesis and organization^{19,58}. The PSB measured at the age of 2 weeks is related to the average collagen fiber diameter in the posterior sclera. As a possible explanation of our results, PSB is indicative of the level of scleral collagen synthesis¹⁹, and relatedly, the proliferation of myofibroblasts⁵⁸. Therefore, the rapid increase in PSB in young animals during the eye growth may indicate the establishment of an effective regulation from vision to scleral development¹⁸, which reduces the animals' susceptibility to myopia.

In addition, we have revised the figure to remove the SE values in Fig. 3 to avoid confusion. The specific animal data can be found in Supplementary Data Figure 2.

5. Are there any differences in the axis orientation values between the normal eyes and the myopia eyes? For example, can the authors find the remodeling of the fibers in the myopia eyes? The authors spend many efforts to provide a higher accuracy axis orientation measurement, however, there is very limited results and conclusions about the axis orientation of the eyes.

According to the reviewer's suggestion, we have added the analysis of scleral fiber orientation change in the guinea pig model and humans. The results are presented in Ext. Data. Figs 6,7,8. These analyses bring more interesting insights to the myopic remodeling in the sclera and provide clues to address comment 3. Please refer to the major change list for all reviewers for more information.

Discussion:

6. The method cited by the author (Ref. 65) is classified as the circularly polarized light-based PS-OCT. However, this method is not limited to the circularly polarized light-based PS-OCT system. As mentioned in this paper, "the requirement of known polarization states incident on the sample can be removed." This needs fixed.

We have replaced all "circular polarized light-based PS-OCT" to "Single input PS-OCT" in the manuscript.

Method:

7. The THIPS-OCT system requires 3 mutually orthogonal input polarization states to compute the results. In Supplementary Fig. 1g, the authors show that the angles between the 3 Stokes vectors are not exactly equal to 90 degrees due to the diattenuation of the cornea, polarization mode dispersion of the system, and measurement noise. Can the author explain the effect when the 3 input polarization states are not exactly equal to 90 degrees (For example, introducing more noises and errors)? After the compensation of the diattenuation and the PMD of the system, what's angles between the 3 input polarization states?

We have carefully evaluated the reason for the changed angles among 3 Stokes vectors and concluded that errors were induced by the diattenuation of cornea and system optics. After compensation of corneal diattenuation (together with the system diattenuation), the error was reduced from 10° to 3°. We realize that

this information is important for readers to evaluate the diattenuation compensation. Therefore, we have added the data, which is suggested by the reviewer, to the revised Supplementary Method Fig 2.

...h, Theoretical bonds of the angle between two orthogonal Stokes vectors after travelling through a diattenuating medium. i, Orthogonality of the Stokes vectors of the backscattered light from the retinal images in e indicated by the orange area before and after corneal diattenuation correction....

In addition, the sentences in Supplementary Method 3 have been revised as follows.

Shown in **Supplementary Method Fig. 2g**, we find that the maximum error is approximately $\pm 10^\circ$ to the set value of 90° . Judging from the theoretical bonds of the angle between two orthogonal Stokes vectors after travelling through a diattenuating medium (**Supplementary Method Fig. 2h**), the magnitude of system and sample attenuation, D , should be around 0.1, defined as the relative difference between the maximum (p_1^2) and minimum (p_2^2) attenuation coefficients, where $D = (p_1^2 - p_2^2)/(p_1^2 + p_2^2)$. The variation between bins is due to the polarization mode dispersion and the wavelength dependence of the polarizing beam splitter. After applying the proposed method to correct for diattenuation using the signals of the surface of the retina, the orthogonality of the Stokes vectors is restored with an error of less than 3° (**Supplementary Method Fig. 2i**).

8. The authors claim the TRIPS-OCT is superior to dual input OCT. However, TRIPS captures the three-input data consecutively whereas dual-input does it concurrently. This rings an issue of the system tolerability to the sample motion for TRIPS approach. This issue must be discussed in a good depth, particularly for in vivo imaging.

The eye motion is much slower than the frame rate in a normal intensity OCT. One piece of evidence is that averaging more than three adjacent B-scans in a normal clinical retinal OCT is widely used and is not degrading the image quality. In TRIPS-OCT, the resolution of birefringence reconstruction is $150 \mu\text{m} \times 30 \mu\text{m}$ (lateral x depth), which is much larger than a normal OCT so that the eye motion represents less of a problem in the intensity-based images.

The concern of sample motion within three repetitive B-scans is related to the phase rather than the intensity. In TRIPS-OCT, Stokes vectors of the three repetitive frames are used. Stoke vectors are naturally intensity signals and the relative phase among those frames is not used in the reconstruction.

To clarify how sample motion impacts TRIPS-OCT and PS-OCT, we have clarified in the revised Discussion.

...TRIPS-OCT does not have the aforementioned limitations but requires a longer acquisition time because of triple repetitive scans at the same location of the sample. However, due to the development of faster lasers and OCT-angiography techniques⁶⁸, repetitive scanning has become a standard in current ophthalmic imaging. TRIPS-OCT is not sensitive to sample motion, as the measurements of modulated frames are based on Stokes vectors and the phase variation between repetitive scans does not influence the reconstruction....

We have numerically simulated the axial motion and have presented the result in Ext. Data Fig. 3f.

...f, Numerical model of the dependency of birefringence noise and the axial motion. Axial motion is modelled by a random shift of the sample in the depth.... Sample motion does not affect any of the reconstruction methods including TRIPS-OCT because the Stokes vectors rather than the phases of repetitive scans are used.

9. The author mentioned in the manuscript: “Favorably, the Mueller matrix reconstruction does not use the absolute phase of the OCT signal, thus TRIPS-OCT is resistant to phase noise induced by laser jitter micrometer-range sample vibration. (Lines 140-142)”. It’s true that the laser jitter would not affect the relative phase measurement of the OCT signals. However, the laser jitters can induce the phase shift between each scan. Can this jitter render the measurements incoherent among the 3 input polarization states?

We agree with the comments that the sentences in Lines 140-142 were not clear. We have removed them from the results section. In TRIPS-OCT, Stokes vectors of the three repetitive frames are used. Stoke vectors are naturally intensity signals and the relative phase among those frames is not used in the reconstruction. To clarify this point, we have conducted a numerical simulation on laser jitter and presented the results in Ext. Data Fig. 3e.

...e, Numerical model of the dependency of birefringence noise and laser jitter. Laser jitter is modelled by a random shift of the simulated fringes in the fringe domain. The random shift, modelled by a Gaussian distribution, induces a relative phase variation between the images encoded in shallow and deep channels of a depth-encoding PS-OCT system... Laser jitter affects the Jones matrix-based method because the relative phase between the two columns of the Jones matrix is used in the reconstruction....

10. In supplementary method 2, the author mentioned the effect of the depolarization. It’s not clear whether the authors correct, ignore or partially correct the depolarization effect. Please explain this part clearly in the supplementary method 2. The definition of the intermediate matrix μ^\wedge is not clear. Does it have the same value as μ ?

We realize that the reconstruction method is a core element of TRIPS-OCT and have moved it to Supplementary Method 1. We have carefully rewritten this section. Briefly, the detected interferometric fringes undergo 3 steps to reconstruct the cumulative Mueller matrix, including (1) filtering in Stokes domain, (2) minimizing the depolarization effect, and last, (3) solving a pure Mueller matrix. Please refer to the major change list for all reviewers for a comprehensive report. In addition, the code of the algorithm has been published in <https://github.com/DrXinyu/TRIPS-OCT>.

Reviewer #2 (Report for the authors (Required)):

Review of Myopia biomarker revealed by triple-input polarization sensitive imaging of the posterior sclera by Xinyu Liu et al.

The authors measure scleral birefringence in a guinea pig model and in humans and correlate the birefringence with myopia.

This is a well written, interesting and extensive paper, that however has major issues.

We thank the reviewer for the valuable comments and feedback. Please evaluate our improvements to the manuscript in the following point-to-point response.

1) The most prominent problem of the manuscript is that the pre-clinical and the clinical results are contradicting. The effect in guinea pigs and humans is opposite. In guinea pigs the birefringence decreases with increasing myopia (figure 3b, higher myopia correlates with lower birefringence), in humans the birefringence increases with increasing myopia (figure 5 i,j,k). This discrepancy is not addressed in the paper. Can this discrepancy between the animal and human model be explained? The predictive value of scleral birefringence is evaluated in guinea pigs from 2 to 8 weeks, however, at 9 months guinea pigs are full grown. How predictive is a correlation at week 8?

This comment has been discussed in the major change list for all reviewers. Briefly, the contradictory results represent different stages of myopia development in a lifespan (the physiological eye growth versus the myopic elongation) and the mechanisms underlying PSB increase are different. We have added new experiments, analysis (Ext. Data Fig. 6-8), and discussion to clarify this issue. Please refer to the major change list for all reviewers for comprehensive information.

2) The paper is surprisingly, unnecessarily and incorrectly negative about other PS-OCT methods such as accuracy and Signal to Noise Ratio (especially depth encoded PS-OCT), and omits mentioning drawbacks of their method with respect to other PS-OCT processing methods.

We apologize for leaving this impression on the reviewer. We have revised the manuscript carefully to provide justified reviews of previous works. Previous development on PS-OCT has greatly expanded its technology and application and has been valuable guidance for our research.

In our previous publications, we have been working on EOM-based dual-input PS-OCT, and have developed the mirror constrained for solving single-input PS-OCT and built a depth-encoding PS-OCT. Our original research proposal was to translate the depth-encoding PS-OCT to pre-clinical and clinical studies. Later, upon the invention of TRIPS-OCT, we found that TRIPS-OCT was advantageous in several aspects compared to previous implementations and may be the current most feasible method for clinical birefringence imaging. The advantages include that TRIPS (1) does not sacrifice the ranging depth (and thus has the optimal SNR by inversely placing the sclera close to DC, the so-called depth enhancing imaging technique), (2) is not sensitive to laser jitter (and thus has no requirement of k-clock hardware), (3) is easy to calibrate and maintain (does not need frequent auto-correlation), (4) can rigorously model diattenuation (does not need to calibrate for the signal roll-off). TRIPS has no obvious drawbacks over previous approaches. Readers may wonder if the triple-repetitive scan that increases the acquisition time may limit clinical applicability. However, in commercial systems such as ZEISS Cirrus, ZEISS Plex-Elite and Heidelberg Spectralis, it is standard to use an average of 4-20 adjacent B-scans to achieve a good image quality. TRIPS can modulate the polarization states of these adjacent B-scans and requires no longer acquisition time than the commercially available devices.

To justify the comparison between different PS-OCT methods, our major changes to the manuscript are listed here

- 1) We have removed the direct comparison of intensity SNR between TRIPS-OCT and depth-encoding PS-OCT. We admit it was incorrect to simultaneously claim the 10 dB SNR improvement and doubled ranging depth. After a careful evaluation, we decided to remove the comparison as the reviewer

- suggested. Depth-encoding PS-OCT sacrifices the optimal SNR window, which is related to the encoding depth, laser linewidth, and digitization rate; after all, these are not related to the core concept of TRIPS.
- 2) We have discussed in detail why the noise is higher in the dual-input and single-input birefringence reconstruction in Ext. Data Fig 3 and Supplementary Discussion 1. Specifically, we pointed out that a dominant noise source in PS-OCT is the edge-artifacts that can be effectively suppressed by removing diattenuation in local retardance reconstruction using Mueller matrix method. Please refer to the major change list for all reviewers for comprehensive information.
 - 3) To compare different PS-OCT reconstruction methods, we have applied the reconstruction methods to numerically simulated phantoms and compared the noise performance regarding different noise sources. We have presented the results in Ext. Data Fig. 3 and Supplementary Method 2. Please refer to the major change list for comprehensive information.
 - 4) We have carefully revised the reviews of previous PS-OCT methods to avoid inappropriate exaggeration and implications. Here is a list of revised sentences.

In Introduction:

... Depth-encoding PS-OCT⁴⁴⁻⁴⁶ is able to measure diattenuation but compromises the optimal ranging window by encoding two orthogonal input states along depth. TRIPS-OCT rigorously measures diattenuation and corrects for depolarization while maintaining the simplicity of dual-input systems. We demonstrate that TRIPS-OCT provides a 2-fold improvement in birefringence sensitivity and an improvement in accuracy of optic axis measurement compared to dual-input PS-OCT. We also demonstrate that TRIPS-OCT outperforms depth-encoding PS-OCT in terms of imaging depth and laser jitter tolerance....

In Discussion:

... Depth-encoding PS-OCT offers reliable depth-resolved measurements independent of the sample but compromises the optimal sensitivity window along the imaging range and requires autocorrelation calibration when the images encoded at the two depths drift in position or amplitude due to environmental variation. The dual-input EOM-based PS-OCT is robust to environmental variation, but it assumes measurements to represent pure retardance, hence, is readily impacted by the presence of diattenuation and edge-artifacts induced by both the system components and the sample. TRIPS-OCT does not have the aforementioned limitations but requires a longer acquisition time because of triple repetitive scans at the same location of the sample. However, due to the development of faster lasers and OCT-angiography techniques⁶⁸, repetitive scanning has become a standard in current ophthalmic imaging. TRIPS-OCT is not sensitive to sample motion, as the measurements of modulated frames are based on Stokes vectors and the phase variation between repetitive scans does not influence the reconstruction. Overall, TRIPS-OCT mitigates some drawbacks of traditional PS-OCT implementations and makes it more suitable for clinical translation.

- 3) At the same time, 75 out of 160 eyes were excluded from the study due to poor Signal to Noise Ratio.

We have carefully reviewed the raw data and found that a large portion of data was rejected by sub-optimal positioning. Accordingly, we have revised the Result section as follows.

Due to the requirement for a sufficient signal from the sclera, we excluded 75 eyes (47%) based on quality criteria (**Supplementary Discussion 3**), composed of images with suboptimal positioning (50 eyes, 31%) and insufficient signal-to-noise ratio (average scleral SNR < 4.6 dB) from the sclera (25 eyes, 16%).

In the discussion section, we have further clarified the problem of intensity SNR limitation.

...First, TRIPS-OCT measurement is fundamentally limited by the intensity SNR. As the sclera is a dense and highly scattering structure, only a 100- μm band from the upper sclera can be reliably measured. In this study, we estimated the intensity SNR of the 100- μm scleral band and excluded approximately 16% of all imaged eyes due to insufficient intensity SNR. We found that a thick choroid (roughly > 450 μm) was one of the reasons limiting light penetration to the sclera....

We added Supplementary Discussion 3 to further clarify the engineering requirements of the alignment aiding devices and the fundamental limitation of the penetration.

Supplementary Discussion 3: Image quality criteria and penetration limitation

The workflow of image quality check for TRIPS-OCT human data included two steps. First, by inspecting the volume scan of the retina, we excluded 50 datasets with suboptimal depth or lateral positioning, or inaccurate pupil alignment (**Supplementary Discussion Fig. 3a**). Second, on the remaining datasets, the choroidal-scleral interface was manually labelled, and the average scleral intensity signal-to-noise ratio (SNR) was estimated from a 100- μm slab below the manually labelled choroidal-sclera interface. Datasets with an average scleral intensity SNR lower than 4.6 dB were excluded from the analysis.

Supplementary Discussion Fig. 3 | Example datasets excluded in the analysis. **a**, Cross-sectional images in datasets excluded by the reason of suboptimal positioning. **b**, Cross-sectional images in datasets excluded by the reason of insufficient intensity signal from the sclera. The yellow arrow in **b** indicate the choroidal-scleral interface. Scale bars, vertical: 300 μm , horizontal: 1 mm.

A relatively large portion (50 of 180) of the data was rejected by inappropriate alignment of the scanning head. As a first-in-human study, comparing to a commercial device, the TRIPS-OCT lacks a fundus preview, multiple location B-scan previews, a motor-driven reference mirror, and a retina tracker. Without these alignment-aiding techniques, it is relatively difficult to align the scanning head to the right position. In our study, for safety consideration, we did not allow for repetitive scans more than two times in one patient during one visit. Repetitive imaging was limited because TRIPS-OCT is classified as a dose-limited Group I system according to the American National Standard for Ophthalmic⁸, meaning that no potential light hazard exists in the procedure under the condition of a limited exposure time. For the abovementioned reasons, the rejection rate was high in this study due to inappropriate positioning. However, we believe that this issue can be minimized by the further engineering development of the alignment system.

TRIPS-OCT penetration is fundamentally limited by the intensity SNR. In this study, we excluded 25 (of 180) datasets due to insufficient intensity signal from the sclera. We found that thick choroid (roughly $> 450 \mu\text{m}$) (**Supplementary Discussion Fig. 3b**) was one of the main reasons blocking the visibility of the sclera. To overcome this limitation, techniques to further enhance the imaging sensitivity of TRIPS-OCT are still needed.

4) Quantitative information on the instruments is lacking (like scan speed, depth resolution, and depth resolution of the birefringence information). The method to mitigate polarization mode dispersion, spectral binning, reduces the depth resolution significantly, but this is not mentioned.

In the revised manuscript, we have added the required information to the first paragraph of the Methods section. Please refer to the major change list for all reviewers for comprehensive information.

5) The birefringence of the sclera is evaluated at the choroid-sclera interface, taking a 200 micron band centered on this interface. It is puzzling that to quantify the scleral birefringence, half of the signal is coming from the choroid, which is a different tissue structure. It is unclear why this choice was made, and why also incorporating the choroid characterizes the scleral birefringence better than calculating the birefringence over a band 200 micron below the choroid-sclera interface.

To minimize the truncation of inner parts of the sclera a 200- μm band centered on the choroidal-sclera interface is used to produce the en-face images and quantify the birefringence measurement. We added a few sentences to the Discussion section in the revised manuscript to clarify this problem.

...First, TRIPS-OCT measurement is fundamentally limited by the intensity SNR. As the sclera is a dense and highly scattering structure, only a 100- μm band from the upper sclera can be reliably measured.... Lastly, the scleral birefringence measurement highly depends on the segmentation of the choroidal-scleral interface, which is performed manually at present. The issue with manual segmentation is that the interface between the choroid and the sclera is not well defined. There are fine petaloid scleral structures unevenly connecting to the above choroid tissues and large blood vessels going through the sclera from the choroid. To minimize the truncation of the inner parts of the sclera, we used a 200- μm band centered on the choroidal-scleral interface to produce the *en-face*

images and quantify the birefringence measurement. Incorporating an area including the choroid does not affect the PSB measurement because no structure is observed to be birefringent in the choroid....

Page by page comments are provided below....

Page 1

'Here, we developed triple-input polarization-sensitive optical coherence tomography (TRIPS-OCT) with improved sensitivity, accuracy, and robustness compared to traditional PS-OCT'.

This claim is exaggerated. As pointed out below, triple input PS-OCT is likely better than dual input PS-OCT, at the expense of slower acquisition. With respect to depth encoded PS-OCT, the presented comparison and characterization of depth encoded PS-OCT is flawed.

We agree with the reviewer that the claim of robustness may be not easy to quantify. From our view, TRIPS-OCT is easier to implement and maintain. We do, however, agree that depth-encoding PS-OCT may be equally robust. Our concern is that depth-encoding PS-OCT compromises the optimal SNR window which affects the birefringence sensitivity and requires sophisticated compensation of the signal roll-off across the two encoding channels to measure the diattenuation. After all, our direct evidence in the results is based on the comparison of TRIPS and dual-input PS-OCT. In the revised manuscript, we have modified this sentence in the Abstract.

...Here, we developed triple-input polarization-sensitive optical coherence tomography (TRIPS-OCT), which has improved sensitivity and accuracy compared to traditional dual-input PS-OCT, to investigate posterior scleral birefringence as a myopia biomarker in animal models and humans...

Page 2, 1

'Currently, no clinical tool is available to inspect the posterior scleral collagen in-vivo.' This sentence suggests that there is no tool available at all.

However, Willemse et al. recently demonstrated for the first time that PS-OCT can characterize the scleral collagen fiber architecture. This should be changed into something like 'no commercially device is currently available to ...'

Willemse J, Gräfe MGO, Verbraak FD, de Boer JF. In Vivo 3D Determination of Peripapillary Scleral and Retinal Layer Architecture Using Polarization-Sensitive Optical Coherence Tomography. *Transl Vis Sci Technol.* 2020;9(11):21. Published 2020 Oct 19. doi:10.1167/tvst.9.11.21

We have revised the text as

...Currently, no tool is commercially available to inspect posterior scleral collagen *in-vivo*....

Page 2, 2

The authors state that it is difficult to image the sclera because 'the probing light is attenuated and the input polarization state is scrambled when passing through the eye³⁸'. This statement is incorrect and doesn't follow from the work presented in ref. 38. The polarization state changes due to birefringence when passing through the eye, but doesn't necessarily get scrambled.

Ref 38 supports the claim that light is attenuated and therefore depth enhanced imaging or swept source imaging is required for sclera imaging. TRIPS-OCT naturally is a combination of depth enhanced imaging and swept source imaging.

Regarding the polarization state of the probing light, we thought to highlight that the polarization state is scrambled by the RPE and pigmentation in the choroid. We agree with the reviewer that our manuscript focuses on the compensation of corneal birefringence, so the text has been revised as

...Unlike the anterior sclera that can be directly accessed³⁷, imaging the posterior sclera in humans is far more challenging and requires high detection sensitivity and accuracy, because the probing light is attenuated³⁸ and the input polarization state is altered when passing through the eye³⁹....

39 Pircher, M. *et al.* Corneal birefringence compensation for polarization sensitive optical coherence tomography of the human retina. *J. Biomed. Opt.* **12**, 041210 (2007).

Page 2, 3

'... the architecture of scleral collagen fibers in normal eyes was revealed³⁹, but the low scleral signal limited a further quantitative analysis of the scleral birefringence'.

This statement unfairly characterizes prior results. No quantitative analysis was performed in ref. 39, but not because of low scleral signal. With TRIPS-OCT, the scleral signal was too low for quantitative analysis in 75 out of 160 eyes and no further than 100 μm under the scleral-choroidal interface was used for analysis. One could consider this a low scleral signal as well.

We agree with the reviewer that intensity SNR or the light sensitivity is a fundamental limitation for the penetration of TRIPS-OCT. In the revised text, we have discussed this issue with a numerical simulation in Ext. Data Fig. 2.

Regarding the sentences in the introduction, we have revised it as

...Recently, posterior sclera imaging using PS-OCT has been demonstrated in seven healthy volunteers and the architecture of scleral collagen fibers in normal eyes has been revealed⁴⁰. However, investigating the clinical value of scleral collagen imaging in pre-clinical and clinical settings would benefit from further improvements in detection sensitivity and system robustness.

Page 2, 4

'The limitations in detection sensitivity and high system complexity of current PS-OCT have hampered its translation from the optical bench to the clinic.'

This is an unfair and incorrect characterization of prior work, there are no limitations in detection sensitivity and the high system complexity is exaggerated. These 'highly complex' systems have made it to the clinic. The most complex part about the system is the detection unit, and the complexity of the detection unit is equal for triple, dual and depth encoded PS-OCT. The polarization delay unit required for depth encoded PS-OCT is fairly simple compared to the detection unit that all systems have in common.

'In particular, the clinical study of using PS-OCT to image posterior scleral collagen in patients with myopia and pathologic myopia has not been achieved.'

Please rephrase. This is not related to the prior unfair and incorrect characterization, it has not been achieved because it has not been tried before.

We agree with the reviewer that the complexity of detection unit is equal, and the integrated polarization diverse detection module has been commercially available (PDR-106021103, G&H, Dowlsh Ford, UK). With our own experience, the depth encoding unit requires to collimate two free-space beams into a fiber with a good coupling efficiency with the encoding depth to be adjustable. Additionally, this unit needs to be minimized in size and stable for a long-term use in the clinical environment. We found that these features were challenging when we were translating our depth-encoding system to the clinic. In contrast, the TRIPS-OCT system was easier to build (especially with the commercial detection module) and was more robust in the clinical setting.

In the revised Introduction, we have been very careful with our statements about the complexity of depth-encoding PS-OCT to avoid unjustified characterization.

Page 2, 5

'Current PS-OCT instruments using sequential dual-input polarization states^{40,41} assume to measure pure retardance, ignoring the negative impact of diattenuation.' This statement is not true. Diattenuation was characterized with dual-input polarization states by Park et al. (2004)

B. Hyle Park, Mark C. Pierce, Barry Cense, and Johannes F. de Boer, "Jones matrix analysis for a polarization-sensitive optical coherence tomography system using fiber-optic components," Opt. Lett. 29, 2512-2514 (2004)

We appreciate the paper recommended here and re-read it carefully. We agree that the diattenuation was correctly characterized in that paper using an ex-vivo system. This is one of the papers claiming that diattenuation was neglectable and encouraged later PS-OCT to use a dual-input (orthogonal states on a Poincare sphere) method. Although the true diattenuation in the sample is neglectable, we demonstrated that a rigorous

model of the diattenuation can suppress the edge-artifacts and correct for the corneal diattenuation, thereby improving the birefringence sensitivity.

To clarify, we have revised the text as

...Current electro-optic modulator (EOM)-based PS-OCT instruments using sequential dual-input polarization states^{41,42} assume that the measurements contain only pure retardance and the impact of sample diattenuation is neglectable⁴³....

43. Hyle Park, B., Pierce, M. C., Cense, B. & de Boer, J. F. Jones matrix analysis for a polarization-sensitive optical coherence tomography system using fiber-optic components. *Opt. Lett.* **29**, 2512 (2004).

Page 2, 6

Depth-encoded PS-OCT is characterized incorrectly on multiple occasions in the manuscript. Statements such as ‘... [depth encoded PS-OCT] uses complex high-maintenance systems that sacrifice sensitivity and imaging range’ are exaggerated and incorrect.

We have revised the manuscript on all occasions where depth-encoding PS-OCT is mentioned. In the revised manuscript, we tried to be fair and objective.

In the Introduction section,

...Depth-encoding PS-OCT^{44–46} is able to measure diattenuation but compromises the optimal ranging window by encoding two orthogonal input states along depth. TRIPS-OCT rigorously measures diattenuation and corrects for depolarization while maintaining the simplicity of dual-input systems.... We also demonstrate that TRIPS-OCT outperforms depth-encoding PS-OCT in terms of ranging depth and laser jitter tolerance....

In the Discussion section,

...Today, there are three major types of PS-OCT instruments, including single-input PS-OCT^{67,68}, depth-encoding PS-OCT^{45,46,69}, and dual-input EOM-based PS-OCT^{41,42}.... Depth-encoding PS-OCT offers reliable depth-resolved measurements independent of the sample but compromises the optimal sensitivity window along the imaging range and requires autocorrelation calibration when the images encoded at the two depths drift in position or amplitude due to environmental variation.

In the Ext. Data Fig. 2 caption,

...Laser jitter is modelled by a random shift of the simulated fringes in the fringe domain. The random shift, modelled by a Gaussian distribution, induces a relative phase variation between the images encoded in shallow and deep channels of a depth-encoding PS-OCT system.... Laser jitter affects the Jones matrix-based method because the relative phase between the two columns of the Jones matrix is used in the reconstruction....

Page 3, 1

‘As far as we know, there has been no previous clinical studies focusing on PSB in patients with myopia’ should be ‘As far as we know, there have been no previous clinical studies focusing on PSB in patients with myopia’.

We have corrected the error in the revised manuscript.

Page 3, 2

It should be specified that polarization modulation configuration was already demonstrated by Saxer et al. (2000). Therefore, the approach is not novel. The authors build on prior work. The novelty is creating 3 orthogonal states in the Poincare sphere.

Saxer, C. E., de Boer, J. F., Park, B. H., Zhao, Y., Chen, Z., & Nelson, J. S. (2000). High-speed fiber-based polarization-sensitive optical coherence tomography of in vivo human skin. *Optics letters*, 25(18), 1355-1357.

We have added this to the Methods section of the revised manuscript.

... We replaced the polarization-depth-encoding unit with a novel triple-input modulator (**Ext. Data Fig. 1b**), consisting of a polarizer and an electro-optic modulator (EOM) (4104NF, Newport, Irvine, US), inspired by the previous dual-input configuration⁷⁰....

70. Saxer, C. E. et al. High-speed fiber-based polarization-sensitive optical coherence tomography of in vivo human skin. *Opt. Lett.* **25**, 1355 (2000).

'... reconstruction does not use the absolute phase of the OCT signal, thus TRIPS-OCT is resistant to phase noise induced by laser jitter and micrometer-range sample vibration'. This sentence implies that this is a problem in other PS-OCT systems, which is not the case. No prior PS-OCT system uses the absolute phase of the OCT signal. As pointed out in the earliest PS-OCT dual detector systems, the relative phase between two orthogonal polarization state interference signals is relevant. Even with depth encoded PS-OCT the absolute phase is not used. This is an inaccurate characterization of prior work.

We agree with the reviewer and have removed this sentence. Furthermore, we conducted a numerical simulation to validate the reviewer's point and include the result and conclusion in the revised manuscript. The simulation is presented in Ext. Data Fig. 2f.

...f, Numerical model of the dependency of birefringence noise and axial motion. Axial motion is modelled by a random shift of the sample in the depth.... Sample motion does not affect any of the reconstruction methods including TRIPS-OCT because the Stokes vectors rather than the phases of repetitive scans are used.

Can the authors provide more evidence for their claim that the whole inner retina of the guinea pig exhibits the same value of low birefringence throughout all the layers? Nerves are present in the inner retina which exhibit birefringence.

We searched for literature but found no reports of birefringence of the inner retina of a guinea pig. In a few PS-OCT studies on mice and rats, the nerve fiber is very thin and birefringent structures in the inner retina of small animals have never been reported.

We understand the concern of the reviewer that the stripes in Fig. 1a may be real structures, and thus, undermine the characterization of noise between dual-input and TRIPS. With a significant improvement by further analysis, simulation and discussion, we have shown strong evidence that the "stripes" are caused by the edge-artifacts. In the revised manuscript, we have clarified the origin of the artifacts in revised Ext. Data Figs. 3a-b and Supplementary Discussion 1. Please find more information in the list of major changes for all reviewers.

We have added a paragraph in the Results section to clarify the dominant noise source in the dual-input reconstruction.

We observed that the dual-input reconstruction method suffers from edge-artifacts that are associated with variations in the sample scattering profile (Ext. Data Figs. 3a, b). The edge-artifact is a dominant source of birefringence noise that is induced by a shift of the point-spread-functions in the two eigen-polarization states of the preceding system components or tissue layers (Supplementary Discussion 1). Because this shift leads to apparent diattenuation, the resulting artifacts are markedly suppressed by correctly accounting for diattenuation in the Mueller matrices of the sample. TRIPS-OCT isolates sample birefringence properties from the Mueller matrices, and thus is almost free from edge-artifacts (Ext. Data Fig. 3c).

In addition, we have added a sentence to the caption of Fig. 1a to clarify.

... The reddish stripes in the inner retina that are present in the dual-input reconstruction are induced by edge-artifacts (Ext. Data Figs. 3a, b). Of note the artifacts disappear in the TRIPS reconstruction....

Page 4

By comparing Fig. 1g (dual state PS-OCT) and Fig. 1h (TRIPS-OCT), two layers seem to appear with higher birefringence in Fig. 1g which are not present in Fig. 1h. These two reddish bands do not correspond to low intensity. Why does the dual state PS-OCT show additional layer contrast which disappears in the TRIPS-OCT image? One expects that noise does not have structure, is this then truly noise?

We have conducted further simulation and analysis and obtained strong evidence that the “reddish bands” are caused by edge-artifacts. In the revised manuscript, we added Ext. Figs. 3a-c and Supplementary Discussion 1 to clarify the origin of the noise. Please find more information in the list of major changes for all reviewers.

Page 4, 2

The approach to compare birefringence noise seems unfair if the triple-state measurement takes 1.5 times longer. This time could be used for averaging in dual-state measurements.

We have ensured that the SNR and acquisition duration of the comparison between TRIPS and dual-input are the same. The method has been presented in Ext. Data Fig. 1. To clarify this, we have revised a sentence in the Results section.

...In this comparison, we ensured that the signal-to-noise ratios (SNRs) and the sampling time of the original signals used by the two methods were identical (Ext. Data Figs. 2a-c)....

Page 4, 3

There are three possible combinations for the dual state measurements, (1,2), (1,3) and (2,3). Are the presented results for dual state equivalent for these three pairs? Please compare noise levels, which could depend on the chosen combination.

According to the reviewer’s suggestion, we analyzed the data of two more possible combinations in the dual-input reconstruction. We have included these results in Ext. Data Figs. 2d, e. The noise level of dual-input reconstruction is quite consistent. Slight differences in the noise level of different combinations are due to the dependency of the edge-artifacts and the absolute polarization states, as discussed in revised Supplementary Discussion 1.

...d, Birefringence images reconstructed from triple-input and dual-input methods using different combinations of the Stokes vectors. e, Birefringence noise characterization for the difference combinations using the inner retina in b indicated by the orange area. The improvement in noise performance using TRIPS-OCT is quite consistent between different combinations of Stokes vectors. The slight difference in the noise level of dual-input combinations is due to the dependency between the edge-artifacts and the absolute polarization states (Supplementary Discussion 1)....

Page 6:

The authors claim a close agreement between TRIPS-OCT in Fig. 2d,f and PLM in Fig 2e,g. This statement seems valid enough comparing Fig 2f and 2g. However, the orientations depicted on the right side of Fig 2d and 2e seem completely different. Can the authors comment on this?

We have added notations to the images and explanation in the caption of Fig. 2

...Due to the eyeball deformation during the tissue fixation, the cryosectioning plane intersected with the inner eye and the outer sclera, creating discrepancy areas in **d** and **e**, enclosed by white dotted lines....

Page 7

A colorbar is missing in Figure 3a, and the SE labels are hard to read.

We have added the colorbar and removed the SE labels.

Page 7, 2

The effects in guinea pigs and humans is opposite. In guinea pigs the birefringence decreases with increasing myopia, in humans the birefringence increases with increasing myopia. Can this discrepancy between the animal and human model be explained?

This comment has been discussed in the major change list for all reviewers. Briefly, the contradictory results represent different stages of myopia development in a lifespan (the physiological eye growth versus the myopic elongation) and the mechanisms underlying PSB increase are different. We have added new experiments, analysis (Ext. Data Figs. 6-8), and discussion to clarify this issue. Please refer to the major change list for all reviewers for comprehensive information.

Page 9

'Due to the requirement for a sufficient signal from the sclera, we excluded 75 eyes based on the average en-face intensity signal by a threshold of 30 dB.' How is the dB signal from an en face image defined? It should be specified how large a fraction 75 eyes is of the total amount of eyes scanned, and that the author's criterium is rejecting a lot of data.

We have carefully reviewed the raw data and found that a large portion of data was rejected due to sub-optimal positioning. Accordingly, we revised the Result section as follows.

...Due to the requirement for a sufficient signal from the sclera, we excluded 75 eyes (47%) based on quality criteria (**Supplementary Discussion 3**), composed of images with suboptimal positioning (50 eyes, 31%) and insufficient signal-to-noise ratio (average scleral SNR < 4.6 dB) from the sclera (25 eyes, 16%)....

In the discussion section, we have further clarified the problem of intensity SNR limitation.

...In this study, we estimated the intensity SNR of the 100- μ m scleral band and excluded approximately 16% of all imaged eyes due to insufficient intensity SNR. We found that a thick choroid (roughly > 450 μ m) was one of the reasons limiting light penetration to the sclera....

Page 10

Optic axis cross-sectional images seem to be displayed with a different colorwheel than en face optic axis images. While the en face images are clear and seem to correspond the provided colorwheel, the cross sectional images are/do not.

To help understand the in-plane orientation, we have made different colorbars (color disk and color cylinder) for the en-face and cross-sectional images, respectively. All figures and videos have been updated.

Page 10,2

It is unclear what is shown and pointed out in figure 5h. The ‘choroidal-scleral interface structures resulting in the petaloid birefringence pattern’? I don’t see what the black arrows are pointing at. An explanation is missing in the text. Without the supplementary material, it is unclear to the reader at this point that the en face images are based on the choroidal-scleral interface in general.

To clarify the choroidal-scleral interface structures, we have added notations in Fig. 5h and revised the text in the caption of Fig. 5.

... **h**, Cross-sectional images indicated by the white dashed line in **f**’ showing the uneven choroidal-scleral interface resulting in the petaloid birefringence pattern in **f**’. The blue dashed line indicates the manual segmentation. The orange dashed line indicates uneven structures that are difficult to segment manually...

Page 12

Figure 6d is very hard to interpret. What is meant with ‘the two orthogonal scans’? Isn’t a scan a 3D image in itself? What is orthogonal to that? Videos are a better tool for visualizing 3D structures.

We have revised the text and provided a supplementary video to present the 3D shape of the eye.

...3D reconstruction of the eyeball shape (**d**)....

...we further reconstructed the 3-dimensional eye shape (**Fig. 6d, Supplementary Video 3**) from the volume scan...

Page 13

‘To control early-stage myopia, increasing outdoor activities that can be unequivocally encouraged for all children’. This sentence is grammatically incorrect.

We have revised the sentence.

To control early-stage myopia, increasing outdoor activities can be unequivocally encouraged for all children....

Page 13, 2

The authors state that an increase in PSB can be explained by “a change in collagen fiber direction from circumferential to radial in the peripapillary sclera³⁰”. In ref. 30, WAXS was used to measure the collagen fiber direction. However, this technique lacks depth imaging, and therefore misses the information about the thickness of the layers. If the inner layer becomes thinner for example (which is likely in the case of an

elongation of the eye), the signal from the outer layer becomes dominant. Do the authors find this change in direction with their own optic axis images in the longitudinal guinea pig study? Can they confirm the results from Markov et al.?

According to the reviewer's suggestion, we have conducted analysis on scleral fiber orientation change in the guinea pig model and humans. The results are presented in Ext. Data. Figs. 6,7,8. These analyses bring more interesting insights into the myopic remodeling in the sclera and provide clues to address the "contradictory" results between animals and humans.

Briefly, in young guinea pigs, we observe that the scleral birefringence increases as well as the fibers becomes more interwoven. However, in adult guinea pigs, the myopic scleral birefringence increases, while collagen fibers become more aligned. In adult humans, we observed a reduction of interwoven fibers in myopic eyes. The reasons for the reduction of interwoven fibers are not clear yet but based on some previous reports we discuss possible reasons for this observation. Our data indicate that the sclera becomes thinner as myopia progresses, but the correlation between inner sclera thickness and myopia degree is not significant.

We guess that the development of myopia may be divided into two stages. Childhood myopia is related to the retarded development of the sclera fibers. Adult myopia progression is caused by eye elongation with the reduction of interweaving fibers. However, these speculations are not confirmed by direct evidence, so we choose not to include them in this manuscript. We are planning to conduct clinical trials on children in the future to investigate this issue.

To clarify the point, we revised the sentences in the Discussion section.

... The increase in PSB in myopic eyes can be explained by changes in the collagen fiber structure from interwoven to aligned, as observed in *ex-vivo* studies, including (1) unfolding of microstructural crimps^{22,29}; (2) a change in collagen fiber direction from circumferential to radial in the peripapillary sclera³⁰; (3) reorganization of the collagen fibers into a lamellar arrangement (rather than interwoven)⁶³; or (4) a combination of all aforementioned phenomena²¹.

Page 14

Whenever the round-trip retardance reaches a half wave, the local probing light is linearly polarized and can coincide with the sample's optic axis, frustrating the reconstruction of depth-resolved metrics⁶⁷. The work referenced in 67 is not appropriate for this sentence. This has been pointed out much earlier than 2012.

We have removed Ref 67.

Page 14

'... the optimal sensitivity window along the imaging range and requires frequent expert maintenance when the images encoded at the two depths drift in position or amplitude due to environmental variation.'

This is not an issue. The delay can be determined through autocorrelation. This is done in many depth encoded PS-OCT systems. Therefore, frequent calibration to maintain accuracy is not needed in the PDU. The authors raise a moot problem. The same inaccurate statement is repeated on page 15 'Slight alignment drift undermines the PSNR in polarization depth-encoding PS-OCT', and should be removed there as well. Maintenance/alignment of the detection unit is far more of an issue, but this is the same in both TRIPS-OCT and depth-encoded PS-OCT.

We have removed the statements about that depth-encoding PS-OCT requiring frequent maintenance. In addition, we have revised the sentences in the Discussion section.

...Depth-encoding PS-OCT offers reliable depth-resolving measurements independent of the sample but compromises the optimal sensitivity window along the imaging range and requires auto-correlation calibration when the images encoded at the two depths drift in position or amplitude due to environmental variation....

With respect to the detection unit, the authors do not consider wavelength dependent splitting in the PBS of the detection unit in their TRIPS-OCT system.

The wavelength dependent splitting in the PBS has been mitigated in the spectral binning averaging. We have added a sentence to point out this issue in the revised Supplementary Method 3 and 5.

...The variation between bins is due to the polarization mode dispersion and the wavelength dependence of the polarizing beam splitter...

...Using the spectral binning correction, the polarization mode dispersion of optical components and samples, as well as wavelength dependent splitting of the PBS in the detection unit can be mitigated.

Page 14, 2

'... readily corrupted by the presence of diattenuation induced by both the system components and the sample.' In tissue, the axis of the diattenuation and birefringence often coincide, and therefore no 'corruption' occurs. This comment is questionable and lacks nuance.

To clarify, we have revised the sentence in the Discussion section.

... The dual-input EOM-based PS-OCT is robust to environmental variation, but it assumes measurements to represent pure retardance, hence, is readily impacted by the presence of diattenuation and edge-artifacts induced by both the system components and the sample....

Page 15

Ext. data Fig 1c shows a 10 dB better SNR for triple-input PS-OCT compared to depth-encoding. However, this 10 dB increase is not visible at all in ext. data Fig 1b. The depth-penetration in the images is not significantly different. How is this possible?

We apologize for the unfair comparison between TRIPS and depth-encoding PS. We have completely removed this section and relating results in the revised manuscript and replaced the comparison with more informative numerical simulations.

Page 15, 2

'the images at different depths are incoherently combined in polarization depth-encoding PS-OCT but coherently combined in triple input PS-OCT'. This statement is incorrect, and should be the other way around. Images at different depths are coherently combined in depth-encoding PS-OCT but incoherently combined in triple input PS-OCT.

We have completely removed this section and results in the revised manuscript.

Page 15, 3

The comparison of TRIPS-OCT and depth encoded PS-OCT is flawed and should be removed from the manuscript. The images do not show a 10 dB difference in SNR. It is unclear what leads to the 10dB SNR difference, but there is no argument rooted in physics that can explain this. The statements in this paragraph are confusing and incorrect ('the images at different depths are incoherently combined in polarization depth-encoding PS-OCT but coherently combined in triple input PS-OCT'), which makes it difficult to pinpoint the problem. It is unclear how averaging of different images was handled and how it might have affected SNR. It might be that the background level in the second depth encoded PS-state suffers from a significantly higher background due to cross talk from the first depth encoded PS-state. But then the difference is due to a flawed implementation of depth encoded PS-OCT.

We have completely removed this section and relating results in the revised manuscript.

Page 16

Authors should repeat the comparison between dual-input and triple-input for all three combinations of two input states, i.e., (1,2) (2,3) and (1,3) and show that this produces the same results for Fig 1. The effect of this calculation should propagate to Figure 1F to calculate the noise background.

According to the reviewer's suggestion, we analyzed the data of two more possible combinations in the dual-input reconstruction. We have included these results in Ext. Data Figs. 2d, e. The noise level of dual-input reconstruction is quite consistent. Slight difference in the noise level of different combinations is due to the

dependency of the edge-artifacts and the absolute polarization states, as discussed in revised Supplementary Discussion 1.

...d, Birefringence images reconstructed from triple-input and dual-input methods using different combinations of the Stokes vectors. e, Birefringence noise characterization for the difference combinations using the inner retina in b indicated by the orange area. The improvement in noise performance using TRIPS-OCT is quite consistent between different combinations of Stokes vectors. The slight difference in the noise level of dual-input combinations is due to the dependency between the edge-artifacts and the absolute polarization states (Supplementary Discussion 1)....

Page 19

The authors should specify which wavelength (range) and repetition rate was used in this setup. They should also specify the depth range and axial and radial resolutions.

The information has been added to the revised Methods section.

Page 21,1

For guinea pig a maximum birefringence projection is used. Maximum projections are often unreliable and highly influenced by noise. An average or median projection should be used.

Before performing the maximum projection, we have done an average filtering to remove the noise, which is stated in the Method section.

...A 30- μm vertical-line kernel was used to filter each cross-sectional birefringence image, followed by a maximum projection along the depth....

From Ext. Data Figs. 6,7, considering the arrangement of collagen fibers is highly interweaving and random, the maximum project automatically selects the largest birefringence location along the depth where the fibers are most aligned. The maximum projection may be an effective way to reflect the largest fiber diameter and mask out the locations where fibers are interweaving. However, the results between different quantification methods are similar so we choose not to include this discussion in the current manuscript. We are investigating more advanced quantification and processing methods to better quantify the TRIPS-OCT measurements.

Page 21,2

Why are human en face birefringence images from only 200 μm around the choroidal-scleral interface, instead of the sclera? Half of the 'scleral birefringence' is therefore taken from the choroid? No myopia-specific changes are expected in the choroid. If the authors can argue why the choroidal-scleral interface should be used, it should be specified throughout the manuscript that this is the actual biomarker. I.e., 'scleral birefringence' should be replaced with 'scleral-choroidal interface birefringence'. Is this small slab that is used for the en face images also used for the comparisons?

We have added the reason for using a 200 μm slab in the Discussion section. As far as we know, no specific biological structures exist in the choroid-sclera interface.

...Lastly, the scleral birefringence measurement highly depends on the segmentation of the choroidal-scleral interface, which is performed manually at present. The issue with manual segmentation is that the interface between the choroid and the sclera is not well defined. There are fine petaloid scleral structures unevenly connecting to the above choroid tissues and large blood vessels going through the sclera from the choroid. To minimize the truncation of inner parts of the sclera, we used a 200- μm band centered on the choroidal-scleral interface to produce the *en-face* images and quantify the birefringence measurement. Incorporating an area including the choroid does not affect the PSB measurement because no structure is observed to be birefringent in the choroid. In addition, manual labeling is a time-consuming and subjective process, whereas an automated and reliable image segmentation algorithm will improve the accuracy and efficiency of TRIPS-OCT analysis.

Page 26

In Supplementary Fig. 1g, the authors show a variation of 10 degrees for the orthogonality of the input states, which is surprisingly large. "The reasons for the errors are probably the diattenuation of the cornea, polarization mode dispersion of the system, and measurement noise." Can the authors also present this graph after compensation for diattenuation and PMD? The polarizing beam splitter in the detection unit also affects the spectral dependence of the orthogonality, which is not taken into account.

We have carefully evaluated the reason for the changed angles among 3 Stokes vectors and concluded that errors were induced by the diattenuation of cornea and system optics. After compensation of corneal diattenuation (together with the system diattenuation), the error was reduced from 10° to 3°. We realize that this information is important for readers to evaluate the diattenuation compensation. Therefore, we have added the data, which is suggested by the reviewer, to the revised Supplementary Method Fig 2.

...h, Theoretical bonds of the angle between two orthogonal Stokes vectors after travelling through a diattenuating medium. i, Orthogonality of the Stokes vectors of the backscattered light from the retinal images in e indicated by the orange area before and after corneal diattenuation correction....

In addition, the sentences in Supplementary Method 3 have been revised as follows.

Shown in **Supplementary Method Fig. 2g**, we find that the maximum error is approximately $\pm 10^\circ$ to the set value of 90° . Judging from the theoretical bonds of the angle between two orthogonal Stokes vectors after travelling through a diattenuating medium (**Supplementary Method Fig. 2h**), the magnitude of system and sample attenuation, D , should be around 0.1, defined as the relative difference between the maximum (p_1^2) and minimum (p_2^2) attenuation coefficients, where $D = (p_1^2 - p_2^2) / (p_1^2 + p_2^2)$. The variation between bins is due to the polarization mode dispersion and the wavelength dependence of the polarizing beam splitter. After applying the proposed method to correct for diattenuation using the signals of the surface of the retina, the orthogonality of the Stokes vectors is restored with an error of less than 3° (**Supplementary Method Fig. 2i**).

Page 27

It is unclear what is meant with the 'deterministic component' μ in supplementary method 2. 'Eq. 6 imposes that the G-product of any Stokes vector pair is unitary'. This sentence is unclear, and H is not unitary? After this sentence, the rest of the section is even more difficult to follow and the notations are unclear (e.g. what is meant by $c\text{-}m\text{-}c\text{-}m\text{-}n\text{-}T\text{-}G\text{-}m\text{-}m\text{-}?$). Please clarify.

We realize that the reconstruction method is a core element of TRIPS-OCT and have moved it to Supplementary Method 1. We have carefully rewritten this section. Briefly, the detected interferometric fringes undergo 3 steps to reconstruct the cumulative Mueller matrix, including (1) filtering in Stokes domain, (2) minimizing the depolarization effect, and last, (3) solving a pure Mueller matrix. Please refer to the major change list for all

reviewers for a comprehensive report. In addition, the code of the algorithm has been published in <https://github.com/DrXinyu/TRIPS-OCT>.

Page 30

"... and the depth-resolved physical optic axis can be solved." Do the authors mean the absolute orientation of the optical axis? This is impossible, since the circular birefringence of the system, the cornea and the vitreous remains unknown. Please change into '... and the depth-resolved relative optic axis can be solved'.

We agree with the reviewer that the circular birefringence of the system is impossible to measure. In TRIPS-OCT, we used the Henle fiber layer (known to be radially around the foveal pit) to obtain the absolute orientation of the optic axis. We have revised the sentence in the text.

Taking advantage of the symmetry constraint, the free parameters in the Mueller matrix are further reduced, and the depth-resolved physical optic axis can be solved with a prior knowledge of the underlying structure (i.e., Henle fiber layer)...

Page 30, 2

The authors claim that the original measured Mueller matrix, L , does not fulfil the symmetry constraint due to the circulator only. This is also due to non-reciprocally traversed optical fiber. Isn't the constraint that is utilized here that the measured circular birefringence of the sample should be zero, a constraint also pointed out by Park et al. (2005).

Park, B. H., Pierce, M. C., Cense, B., & De Boer, J. F. (2005). Optic axis determination accuracy for fiber-based polarization-sensitive optical coherence tomography. *Optics letters*, 30(19), 2587-2589.

We have revised the sentence in the text.

...The reciprocity of light in our TRIPS-OCT system is disrupted by the non-reciprocally traversed optical circuit induced by a circulator¹⁶....

16. Park, B. H., Pierce, M. C., Cense, B. & de Boer, J. F. Optic axis determination accuracy for fiber-based polarization-sensitive optical coherence tomography. *Opt. Lett.* **30**, 2587 (2005).

Page 32

'Spectral binning to correct PMD was proposed ...' should be changed into 'Spectral binning to mitigate PMD was proposed ...'.

Spectral binning does not correct for PMD. As Zhang et al have pointed out, to correct for PMD more than one measurement should be used.

Ellen Ziyi Zhang, Wang-Yuhl Oh, Martin L. Villiger, Liang Chen, Brett E. Bouma, and Benjamin J. Vakoc, "Numerical compensation of system polarization mode dispersion in polarization-sensitive optical coherence tomography," *Opt. Express* 21, 1163-1180 (2013)

It should be pointed out that with spectral binning the depth resolution is significantly reduced.

We revised the sentence in the text.

Spectral binning to mitigate PMD was proposed²⁰ by dividing the fringe into several windows, in which the PMD is small and negligible....

Page 32,2

The second equation of the Supplementary Method 2 indicates that Mueller matrices of each bin (with a depth resolution x times poorer than the original OCT image) are averaged into a general Mueller matrix. What is the depth-resolution of the birefringence? It should be mentioned in the main text what it is and that it is x times poorer than the original OCT resolution.

We have added the information to the first paragraph of the Methods section. Please refer to the list of major changes for all reviewers for more information.

Page 34

The approach presented in Supplementary Method 5 seems very interesting. Corneal birefringence is

corrected in a frame-based manner. Would it also be possible to correct in 2D? x-direction as well as y-direction? Would you be able to create an image of the corneal phase retardation?

We did correct the corneal retardance and diattenuation in 2D for a volume scan of the posterior eye. We are able to create images of corneal phase retardance and diattenuation. In the revised Results section, we have further assessed the impact of corneal retardance and corneal diattenuation on the accuracy of optic axis measurement.

To test whether compensation for corneal retardance and diattenuation improves the accuracy of optic axis measurements, we scanned Henle’s fiber layer (HFL) in the retina of a healthy volunteer (32 years, male, OD) (Fig. 1d). The in-plane orientation of the HFL is approximately radially distributed around the fovea⁵². We used the orientation of the HFL to assess the accuracy of the optic axis measurements. We extracted the 2-dimensional corneal retardance and diattenuation maps from the surface of retina (Supplementary Method 6). In the optic axis images of the HFL, the measured orientations were plotted against the angular location on a circle centered on the foveal pit (Fig. 1e). Without corneal correction, the recovered fiber orientation deviates significantly from the assumed radial profile. Correcting for retardance and diattenuation results in fiber orientations that closely follow the anticipated profile. After applying the correction for corneal retardance and diattenuation, the mean error of the measurement, characterized by the difference between the measured optic axis orientation and the assumed orientation, was 35% lower than that of the uncorrected results (Fig. 1f).

d. Two-dimensional correction of corneal retardance and diattenuation for a volume scan. The *en-face* intensity image (upper left) from a healthy human subject (32 yr., male, OD, Asian) was rendered from a volume scan of the posterior eye. The corresponding corneal retardance (upper middle) and diattenuation (upper right) maps were extracted from the retinal surface, and the optic axis images were reconstructed without (lower left) and with the correction for corneal retardance (lower middle) and for both retardance and diattenuation (lower right). The position of the fovea is indicated by a white arrow. The magnitude of diattenuation D is defined as the relative difference between the maximum p_1^2 and minimum p_2^2 attenuation coefficients, where $D = (p_1^2 - p_2^2)/(p_1^2 + p_2^2)$. Zoomed-in images indicated by white boxes highlight the Henle fiber layer (HFL). **e.** Measured in-plane HFL fiber orientation against angular location on a circle (indicated by white dotted circles in **d**) centered on the fovea with an eccentricity of 2° . **f.** Optic axis measurement error without/with corneal correction. Error bars indicate the standard deviation.

52. Cense, B. et al. Henle fiber layer phase retardation measured with polarization-sensitive optical coherence tomography. *Biomed. Opt. Express* 4, 2296–306 (2013).

The authors should make it clearer that circular birefringence was not corrected. ‘... the single pass cornea birefringence ...’ should be ‘the single pass cornea linear birefringence’. Circular birefringence of the cornea cannot be measured.

We have revised the sentence in the Methods section to clarify that only linear birefringence can be corrected.

...Any single-pass circular retardance and diattenuation cancels on the round-trip, evading $S(x)$, and remains uncompensated.

Page 34,3

The Mueller matrices on the surface were extracted, denoted as $S(x)$, which is a function of the lateral position. To reconstruct the single pass cornea birefringence, $C(x)$, the differential Mueller matrix $s(x)$ was calculated by taking the matrix logarithm of $S(x)$.

What are the input Stokes vectors that evolve to the Stokes vectors from the surface to calculate the Mueller matrix or differential Mueller matrix? Is an identity matrix assumed here (after corrections) as input?

$S(x)$ and $C(x)$ are correction matrices respect to the cumulative Mueller matrices $M(x,z)$. The input is a probing 4×3 matrix. In the revised Supplementary Method 1, we have clarified this issue.

... Without loss of generality, the input probing light with three orthogonal input states before going through the system and sample can be assembled into a probing matrix s .

$$s = \begin{bmatrix} 1 & 1 & 1 \\ 1 & 0 & 0 \\ 0 & 1 & 0 \\ 0 & 0 & 1 \end{bmatrix} \dots (2)$$

The reconstructed Stokes vectors corresponding to the three input states can be assembled into a 4×3 detection matrix $\mu(x, z) = [\hat{S}_1(x, z) \quad \hat{S}_2(x, z) \quad \hat{S}_3(x, z)]$. Using a Mueller matrix to describe the entire optical system and sample from the probing matrix to the detection,

$$\mu(x, z) = M(x, z)s \dots (3) \dots$$

No page associated

The link provided for the central algorithm to reconstruct TRIPS-OCT images is invalid.

The authors flip en face images left-right, without providing a reason. It is more intuitive to display the en face images in the same orientation as regular fundus photos, and have the B-scans match this orientation.

According to the reviewer's suggestion, we have replaced the en-face images as regular fundus photo display. In our TRIPS-OCT, B-scans were taken vertically. Vertical presentation of a cross-sectional image of the retina is, however, not common. Therefore, we present horizontal images of retinal cross-sectional scans. Nevertheless, we have clearly labelled the locations of the B-scans in the en-face images and re-designed the color bar (and color cylinder) for a better orientation presentation.

Figures are cluttered with at least 15 subpanels. If the only purpose of Fig. 1d is to illustrate an angle of 27.37° , it can be left out. Fig. 1b is the same as Supplementary Fig. 1a, and Fig. 1e is similar to Supplementary Fig. 1d.

The original Ext. Data Fig. 1 has been removed. In order to limit the number of subpanels, we have moved a part of the subpanels of the original Fig. 1 to Ext. Data Fig. 1.

Could the authors also add a line to the en face images in the supplementary material video 2 to indicate which B-scan is shown? It looks as if the B-scan is taken vertically. En face images should (be rotated 90° clockwise from the true eye positioning).

We have reproduced the videos according to the reviewer's suggestion. In Supplementary Video 2, the B-scan was taken vertically, and we labelled the true fundus locations in terms of superior, nasal, inferior, and temporal in the video.

Reviewer #3 (Report for the authors (Required)):

This is a very interesting study on a promising new methodology in ophthalmology with special reference to myopia.

We thank the reviewer for the valuable comments and feedback. Please evaluate our improvements to the manuscript in the following point-to-point response.

1. Line 29: "Evaluating the risk of progression of myopia is critical for optimizing the administration and timing of interventions.": There are no interventions available to prevent the progression of high myopia?

We have revised the first sentence of the Abstract.

Predictive biomarkers for the progression of myopia and pathologic myopia are needed to optimize the potential treatment strategies at an early stage....

2. Line 56: "For end-stage myopia, the posterior scleral reinforcement (PSR) surgery, including macular buckling, is a clinically approved therapy to strengthen the posterior sclera and arrest the continued elongation of the eye^{14,15}": These procedures are not generally approved.

We have revised the sentence in the Introduction section.

...For end-stage myopia, posterior scleral reinforcement (PSR) surgery, including macular buckling, is a clinically available therapy to strengthen the posterior sclera and arrest the continued elongation of the eye^{14,15}....

3. Line 70: "and is the main" may be re-worded to "and may be one of the main".

We have revised the text as the reviewer's suggestion.

4. Line 115: Typo: "there has been" may be "there have been".

We have revised the text as the reviewer's suggestion.

5. Line 133: The abbreviation of SNR should be explained

We have spelled out the signal-to-noise ratio (SNR) in the text as the reviewer's suggestion.

6. Line 127ff: The article may be written either past tense or in present tense

We have carefully revised the Results section and consistently used the past tense to present the results.

7. Line 157: "known distributed" may be better "known to be distributed"?

The corresponding sentence has been revised.

...The in-plane orientation of the HFL is approximately radially distributed around the fovea⁵⁴....

8. Did the possibility to visualize the sclera differ between parapapillary regions (with beta, gamma and delta zone and other fundus regions with an intact retinal pigment epithelium)?

We agree with the reviewer that it would be extremely interesting to assess the sclera within the beta, gamma and delta zones. But the resolution of current TRIPS-OCT is 150 μm laterally and is hard to resolve the fine parapapillary structures. In the future, we plan to develop TRIPS-OCT with higher resolution by using a shorter wavelength and enlarging the diameter of the scanning beam. With the current system, only an area with features of several hundred microns can be evaluated.

9. Was the technique dependent on axial length and ethnic background (i.e., pigmentation of the fundus)?

The depth of focus of TRIPS-OCT is 4.6 mm in a normal eye, which is enough to image the posterior eye. Therefore, the TRIPS-OCT measurement principally will not be influenced by the axial length.

We have carefully reviewed our data and labeled the ethnic background of the presented subjects. We did not observe significant differences in scleral birefringence between different ethnic backgrounds. We did observe the penetration of sclera was slightly better in Caucasian eyes than Asian eyes. We have added a sentence to report this observation in the revised Discussion section.

...In addition, we observed that Caucasian eyes provide slightly better penetration of the sclera than Asian eyes, perhaps due to less scattering and absorption caused by lesser pigmentation....

10. Line 211: “animal euthanasia” may be better „after sacrificing the animal”?

We have edited the text as the reviewer’s suggestion.

11. Line 223: physiological eye growth may be different of myopic axial elongation?

We appreciate this comment. This difference helps us understand the results in animal models and humans. Comparing the images of guinea pigs aged at 1 and 12 weeks, we validated the relationship between collagen fiber diameter and scleral birefringence. The enlargement of collagen fibers is due to physiological eye growth. In the revised manuscript, we added Ext. Data Figs. 6,7. In Ext. Data Fig. 6, scleral changes in young guinea pigs were a result of physiological eye growth; in Ext. Data Fig. 7, scleral changes in adult guinea pigs with myopia were a result of myopic axial elongation. To clarify the point, we have revised the related sentences.

We have clarified this point in the Discussion section.

... We suppose that there is a fundamental difference between the scleral changes in the young guinea pig model and adult humans representing different stages of myopia development within the lifespan. Our data indicate that the early eye growth at younger age and the myopic elongation in adults lead to opposing effects on PSB related to scleral collagen fiber arrangement and diameter....

In the Results section:

...A significant increase in scleral birefringence was observed in the older animal due to physiological eye growth. We then sacrificed two guinea pigs at the ages of 1 week and 12 weeks for TEM analysis....

In the Discussion section:

... As such increased PSB in young guinea pigs may indicate the enlargement of scleral fiber diameter, associated with increased scleral collagen synthesis to achieve the required scleral stiffness during the eye growth. In contrast, increased PSB in adult guinea pigs and humans with myopia may indicate alterations in the arrangement of scleral collagen and reductions in interwoven fibers, associated with an extended elongation of the eyeball....

12. Line 251: “refraction development” may be better “development of refractive error”?

We have revised the text as the reviewer’s suggestion.

13. Line 413: The term “alarming” may be dropped

We have revised the text as the reviewer’s suggestion.

Reviewer #4 (Report for the authors (Required)):

Line 98, The reviewer suggests using “three-phase” to replace the word “triple-input” and “triple state” throughout the manuscript. Please consider abbreviating the technology as “3P-PS-OCT”.

For an overall response to the reviewer, we thank the reviewer for the valuable comments and feedback. Please evaluate our improvements to the manuscript in the following point-to-point response.

For this comment, the reason for the name of “triple-input” is that in the history of PS-OCT development, there have been proposed “single-input” and “dual-input” techniques. Triple-input describes the technique in that we use three input polarization states to probe the sample, as well as respecting the previous PS-OCT techniques. In the meanwhile, we agree that “three-phase” may be a possible name, but “2P”, “3P” are usually used to describe imaging techniques “two photon microscopy” and “three photon microscopy”. As such, we prefer to name the technology “Triple-input PS-OCT”.

Line 100, Change to “...., assume that the measurements contain only the pure retardance.....”

We have revised the text as the reviewer’s suggestion.

...Current electro-optic modulator (EOM)-based PS-OCT instruments using sequential dual-input polarization states^{41,42} assume that the measurements contain only pure retardance and the impact of sample diattenuation is neglectable⁴³....

Line 133, Change to “..longer penetration depth..”

The issue is resolved because the sentence is about a comparison between depth-encoding PS-OCT and TRIPS-OCT and has been removed in the revised manuscript. The new comparison is performed by a series of numerical experiments presented in Ext. Data Fig. 2.

Line 134-143, Please remove the descriptions as they are not results.

According to the reviewer’s suggestion, we have moved the paragraph describing the TRIPS-OCT method, together with some sub-panels of the original Fig. 1, to the Methods section and the Ext. Data Fig. 1.

Line 145-152, The noise floor is not equal to sensitivity. Please show that the signal doesn’t attenuate in these two systems in this situation.

The definitions of sensitivity of measurement and the noise floor are closely related. Sensitivity is the weakest signal that a system can detect and is limited by the noise floor. In our system, birefringence measurement is quantitative as the reconstruction method calculates the absolute retardance of the light. The noise floor, using a definite unit ($^{\circ}/\mu\text{m}$), is able to characterize the sensitivity of the birefringence imaging. In other words, there is no signal attenuation because the absolute retardance of the light is measured in both systems. To further clarify this, we have added a sentence to the caption of Fig. 1.

...Birefringence noise floor, characterized by the standard deviation,...

Another supporting evidence of no signal attenuation is in the revised Ext. Data Fig. 2, where an A-line of the retina scan reconstructed by the dual-input and TRIPS methods is shown.

...b, TRIPS and dual-input reconstruction methods on an A-scan of a guinea pig retina *in-vivo*....

Line 154, Please call the method on the compensation of corneal diattenuation and retardance.

We have revised the text to call the method.

...We extracted the 2-dimensional corneal retardance and diattenuation maps from the surface of retina (**Supplementary Method 6**)....

Line 160, The orientation of HFL changes in 3-D near the fovea. The reviewer doesn't understand how the authors calculated the theoretical orientation of HFL.

To clarify the orientation of HFL, we have revised the x label in Fig. 1e and explained that the measurements are "in-plane" orientation. To further clarify, we have added a reference to the previous work about measuring the in-plane orientation of HFL.

...The in-plane orientation of the HFL is approximately radially distributed around the fovea⁵⁴....

In the caption of Fig. 1, we have revised the sentences to clarify the in-plane orientation.

...Measured in-plane HFL fiber orientation against angular location on a circle (indicated by white dotted circles in i) centered on the fovea with an eccentricity of 2°....

Line 230, Could the authors explain why larger collagen fiber diameter corresponds to increased birefringence?

To clarify this issue, we have added a supplementary Discussion section to the supplementary information. Briefly, the reasons of scleral birefringence increasing with collagen fiber diameter may include (1) increased form birefringence because of a larger volume fraction of the fibrillar collagen; (2) increased intrinsic birefringence due to larger collagen content; and (3) thickening of collagen lamellas in the sclera.

Supplementary Discussion 2: Collagen fiber diameter and scleral birefringence

The reasons for scleral birefringence increasing with collagen fiber diameter may include (1) increased form birefringence because of a larger volume fraction of the fibrillar collagen; (2) increased intrinsic birefringence due to larger collagen content; and (3) thickening of collagen lamellas in the sclera.

Scleral form birefringence in a localized region can be crudely modelled by parallel collagen cylinders embedded in an extrafibrillar matrix. Revisiting Wiener's theory¹ for birefringence of a diluted solution of cylinders, for a dielectric solvent with a refractive index n_1 and volume fraction f_1 consisting well-separated, parallelly-aligned cylinders with a refractive index n_2 and volume fraction f_2 , the refractive indices parallel and perpendicular to the cylinder axes, n_e and n_o , are expressed by

$$n_e^2 = n_1^2 f_1 + n_2^2 f_2,$$
$$n_o^2 = n_1^2 + \frac{(n_2^2 - n_1^2) f_2}{1 + \frac{\gamma f_1}{2}},$$

where $\gamma = (n_2^2 - n_1^2)/n_1^2$. We used the parameters in a recent report² using the refractive index of fibrillar collagen $n_2 = 1.504$, and extrafibrillar matrix $n_1 = 1.345$ to model the sclera. From this model, scleral birefringence is increasing with the fibrillar collagen volume fraction f_2 in the range of 0 to 0.5 and reaches a maximum when $f_2 = 0.5$ (**Supplementary Discussion Fig. 2a**).

Supplementary Discussion Fig. 2 | Scleral birefringence increases with the enlargement of collagen fiber. **a**, Form birefringence model based on parallel collagen cylinders embedded in an extrafibrillar matrix. **b**, Geometric model of sclera under constant inter-fiber spacing. **c**, Fibrillar collagen volume fraction as a function of fiber diameter using the geometric model. **d**, Thickening of collagen lamellas during the development of sclera.

It has been reported that the human sclera contains approximately 50% collagen by weight³, with type I fibrillar collagen being the main sub-type. The diameter of collagen fibers varies between 20 to 230 nm and the spacing between collagen fibers is highly irregular and was reported to have an average length of 285 nm (fiber center-to-center, corresponding to a minimum of around 60 nm edge-to-edge) in human sclera⁴. Electron microscopy may not be a proper technique to estimate the volume fraction due to the tissue shrinkage in the preparation process⁵. Using a crude geometric model to study the relationship between the collagen fiber diameter and its volume fraction, we assume the average spacing between fibers (edge-to-edge) to be a constant. This assumption is based on the understanding of the necessary space for the microstructures of the matrix including decorin, biglycan, and glycosaminoglycan forming cross-linking bridges between adjacent fibers. In this geometric model, the fibers are naturally arranged in a hexagonal pattern (**Supplementary Discussion Fig. 2b**). From this model, with the edge-to-edge inter-fiber spacing set as 60 nm, as the fiber diameter increases from 20 to 230 nm, the fibrillar collagen volume fraction increases from 0.057 to 0.57 (**Supplementary Discussion Fig. 2c**).

We attempted to use this model to qualitatively explain the increased birefringence associated with the increased fiber diameter, specifically, the relationship between the collagen fiber diameter and its volume fraction. The limitations of this model include that, first, the collagen fiber diameter is highly variable even within a single lamella; second, the spacing between the fibers is highly irregular; third, the collagen fiber arrangement in a lamella is not parallel but curved and wavy; fourth, Wiener's theory is only accurate in diluted solution where $f_2 \ll 1$. Considering all of these, the form birefringence model of the sclera is crude and limited but provides an explanation of the increased form birefringence with larger collagen fiber diameter. Further investigations are needed to establish an accurate model for the form birefringence in sclera.

In addition to the form birefringence, it has been reported that collagen is highly intrinsically birefringent due to the bonds of protein molecules⁶. The intrinsic birefringence is proportional to the collagen content in the sclera and may also contribute to the increase of total birefringence with a larger collagen fiber diameter.

The collagen fibers are similarly arranged within each lamella, therefore, thickening of lamellas are correlated with the increase of observed birefringence due to more aligned fibers within a resolution volume. In the scleral tissue of guinea pigs from ages of 1 and 12 weeks, we observed that the thickness of lamella was increasing with age (**Supplementary Discussion Fig. 2d**). It has been reported that the growth of sclera in tree shrews is associated with a positive skewing of the distribution of the collagen fiber diameter from a relatively normal distribution profile at birth⁷, consistent with our observations in guinea pigs. The thickening of the lamella is highly likely a result of the

enlargement of collagen fibers in the lamella and is supposed to be one of the reasons of the increase of scleral birefringence.

Putting everything together, a larger average collagen fiber diameter in the sclera leads to increased PSB in TRIPS-OCT images.

Fig. 3, How was the metric of posterior scleral birefringence in panel b calculated from the en face images in panel a?

We have described the metric of PSB calculation in the Methods section.

...The overall PSB value was obtained by a maximum birefringence projection along the radial direction followed by averaging along the circumferential direction....

Furthermore, the detailed birefringence quantification method can be found in Supplementary Method 8.

Line 152, As the authors indicated that the collagen fiber diameter can affect the scleral birefringence and it changes significantly with age. To verify that PBS is correlated with myopia development, please provide results on a control group to exclude the effect of age in the correlation.

In the animal study, we used the data collected from the same age to conduct the correlation analysis. To clarify, we have added a label to Figs 3b, 4c.

In the human study, we recruited adults aged > 21 yr. We have verified that the PSB is not significantly correlated with age in adult humans. The result was mentioned in the correlation matrix of Fig. 5m. The original scatter plot can be found in Supplementary data 5, the plot of PSB vs age.

Line 318, Please provide the distribution of PSB with age in the groups.

Please provide histology validations of scleral remodeling for myopia development in guinea pigs.

We have provided the PSB distribution with animal age. The data can be found in Supplementary Data Figure 3. PSB vs human scattering plot can be found in Supplementary Data 5. We understand that the reviewer may be concerned about how to compare the animal and human results. This issue has been addressed in the list of major changes for all reviewers, where we have used additional experiments and analysis to prove that these are two stages of myopia development in a lifespan. Specifically, increased PSB in young guinea pigs reflects the

physiological increase of collagen fiber diameter, highlighting a high level of scleral collagen synthesis. In contrast, increased PSB in adult guinea pigs and humans with myopia is due to the reduction of interweaving fibers.

We did not provide histology validations of scleral remodeling for myopia development because there is no well-accepted standard to validate myopia histologically. Refraction error is the clinical standard. However, high myopia and pathologic myopia can be diagnosed by OCT scans in which the eye shape can be evaluated. We have added Ext. Data Fig. 7, a scan of high myopia guinea pig with SE of -9 D. We observed obvious thinning of the sclera and the eye shape change towards axial elongation compared to the normal eye.

In the revised manuscript, the following results have been added to Ext. Data Fig. 7.

...From the cross-sectional images, we observe thinning of the sclera tissue and deformation of the eye shape towards axial elongation in the highly myopic eye (-9 D)....

Please rewrite the Discussion to remove any redundant descriptions. The authors should focus on the explanation of larger PBS in myopia, the new insight of in vivo data added to previous ex vivo and histology studies, the limitation, and the prospect of this new method for better diagnosis of myopia.

We have significantly revised the Discussion section. In the revised Discussion section, we first stated our overall rationale for conducting this study. Second, we explained our speculations about the reasons for myopic changes as assessed with TRIPS-OCT in the animal model. Then we discussed the implications of the animal results in the clinical control of early-stage myopia. Third, we explained our speculations about the reasons for myopic changes in adult human eyes. Then we discussed the implications of the human results in the clinical treatment of high myopia and pathologic myopia. Fourth, we compared the animal and human results and clarified the “contradiction” in the results. Fifth, we discussed the technical advantages of TRIPS-OCT in the context of current PS-OCT. Lastly, we discussed the limitations of this study.

Line 527-531, Please move it into Discussion.

The issue has been resolved because the sentences about a comparison between depth-encoding PS-OCT and TRIPS-OCT have been removed in the revised manuscript. The new comparison is performed by a series of numerical experiments presented in the revised Ext. Data Fig. 3.

Line 514, why the lower boundary of sclera is not visible in the B-scan images of Ext. Data Fig. 1 panel d.

The image is about a comparison between depth-encoding PS-OCT and TRIPS-OCT and has been removed in the revised manuscript. Although this image is not a typical one, we can still discern the lower boundary of the sclera, episclera and the fat cells posterior to the eyeball.

The value of this work should stand in claiming that PBS is a better biomarker for myopia than the refractive error. However, the reviewer believes the authors failed on proving this. The reviewer suggests shrinking the description on correlation comparison but highlighting the prediction of myopia from earlier PBS data, especially in human eyes.

In our opinion, refractive error is a sensitive functional biomarker as well as a defining metric for myopia and cannot be clinically replaced by other biomarkers. But PSB may be the first biomarker to relate vision function to microscopic pathophysiological changes in early myopic eyes. Our vision is to establish PSB as a biomarker to aid clinical treatment decisions.

The value of this study is twofold; first, we introduce TRIPS-OCT, a new implementation of PS-OCT with improved sensitivity, accuracy, and robustness; second, we demonstrate that scleral birefringence is closely related to the future development and progression of myopia.

We agree with the reviewer that there is still a lot of work to be done to establish TRIPS-OCT as a new standard in myopia management. However, we believe that the potential of TRIPS-OCT has been fully demonstrated in this translational study. In our vision, further large-scale, longitudinal clinical trials are warranted for TRIPS-OCT.

This email has been sent through the Nature Biomedical Engineering Springer Nature Manuscript Tracking System.

Confidentiality Statement: This e-mail is confidential and subject to copyright. Any unauthorized use or disclosure of its contents is prohibited. If you have received this email in error please notify our helpdesk.

Confidentiality and pre-publicity policy | Privacy Policy | Update Profile

Rebuttal 2

Reviewer #1 (Remarks to the Author):

The revision has addressed my concerns satisfactory. I have no further comments. The study is important which may have high clinical significance if proved useful in further clinical trials.

We thank the reviewer for the supportive comments.

Reviewer #2 (Remarks to the Author):

Review of Myopia biomarker revealed by triple-input polarization sensitive imaging of the posterior sclera by Xinyu Liu et al.

The manuscript has improved considerably, but still contains a few inaccurate statements and raises a few other questions.

We thank the reviewer for the favourable and supportive comments. We have clarified the statements and answered the questions in the following point-by-point response. The sentences related to our revisions in the manuscript and supplementary information are highlighted in orange, and key revisions are underlined.

The authors have addressed the discrepancy between the guinea pig (2-8 weeks) and human measurements by adding one adult guinea pig (n=1) with myopia and one adult guinea pig (n=1) without myopia.

We thank the reviewer for the supportive comment.

From the abstract:

“In a guinea pig study spanning 8 weeks, we found that scleral birefringence is significantly correlated with the development of refractive errors and predicts the onset of myopia. In a human cross-sectional study, we found that scleral birefringence is correlated with myopia status and is associated with scleral pathological changes.”

- Please clarify that an opposite correlation was found in humans and the guinea pig study spanning 8 weeks.

We have revised the sentence in the abstract as follows.

In a study on young guinea pigs spanning 8 weeks, we found that scleral birefringence is positively correlated with refractive errors and predicts the onset of myopia. In an adult human cross-sectional study, we found that scleral birefringence is negatively correlated with refractive errors and is associated with scleral pathological changes.

From Supplementary Discussion 3:

“In our study, for safety consideration, we did not allow for repetitive scans more than two times in one patient during one visit.”

- Was the reason the incident power on the retina, and what was the power used incident on the eye?

We have added a sentence in the Methods section to clarify the laser power used in TRIPS-OCT.

...The laser power entering the eye, which was controlled by a motorized aperture placed in the free space before the triple-state modulator, was set to 1 mW for alignment and 4 mW for retinal volume scan.

In addition, the Ext. Data Fig. 1 has been revised to include the aperture.

... A, Motorized aperture....

We have added a sentence in the Supplementary Discussion 3 to clarify the reason of limited scan time.

...Repetitive imaging was limited because TRIPS-OCT is classified as a dose-limited Group I system according to the American National Standard for Ophthalmics⁸, meaning that no potential light hazard is allowed to arise from the procedure, which limits the total laser exposure time on patients within 24 hours. In our study, considering the patients may already undergo eye examinations in the same visit by other two OCT modalities in our research clinic, to minimize the laser dose exposed on the patients on a single day, our protocol did not allow for repetitive scans more than two times in one patient during one visit....

Page 2:

triple-input PS-OCT (TRIPS-OCT), a novel modulation and reconstruction strategy for PS-OCT that increases imaging sensitivity, accuracy, and system robustness.

- Sensitivity improvement by a factor of 1.33 (See below). Is this improvement significant? Otherwise remove claim.

We have double checked our analysis (described in Ext. Data Fig. 2) and maintain that the comparison between TRIPS and dual-input OCT is correctly considering the sampling time.

We acquired 9 repetitive scans in the same location, corresponding to three sequences of input state modulation. We used only 6 of these B-scans for both the dual-input and TRIPS reconstruction in the comparison. Hence, the effective time to acquire the raw data we used in the comparison was identical (time of 6 B-scans). Because the 6 B-scans for dual-input could indeed have been acquired continuously in a shorter time, we selected the first and the third B-scan sequences (including 6 B-scans) for TRIPS reconstruction. We also confirmed that the intensity SNRs of the averaged

sequences (resulted from the 6 B-scans) between dual input and TRIPS were identical (Ext. Data Fig. 2b).

Furthermore, as we have clarified in our manuscript, with sufficient intensity SNR, the dominant birefringence noise source is edge-artifacts. This conclusion is based on the analysis presented in Ext. Data Figs. 3 c, d. To further support this point, we have included some more specific evidence in this response letter, as shown in the figure below. Note that in both the dual-input and TRIPS reconstruction, the absolute birefringence noise level decreases when more than 1 frame is averaged. Averaging more than 3 B-scans for each input, however, does not reduce the birefringence noise anymore. (For both reconstruction methods, we performed identical spectral binning and spatial averaging, which also increase the intensity SNR. Without spectral binning and spatial averaging, birefringence measurement may be benefit from averaging more than 3 B-scans)

Depth-encoding PS-OCT44–46 is able to measure diattenuation but compromises the optimal ranging window by encoding two orthogonal input states along depth. TRIPS-OCT rigorously measures diattenuation and corrects for depolarization while maintaining the simplicity of dual-input systems.

- Depth-encoding PS-OCT does not compromise the optical ranging window, but utilizes the available ranging window to multiplex polarization input states. On the other hand, TRIPS-OCT compromises speed (3 times slower than Depth-encoding PS-OCT).

We have revised the statement and clarified the drawback of depth-encoding PS-OCT in the Introduction Section.

...Depth-encoding PS-OCT⁴⁴⁻⁴⁶ is able to measure diattenuation but reduces the imaging range due to multiplexing the images of two polarization input states along depth....

TRIPS-OCT rigorously measures diattenuation.

- Suggest to remove rigorously, unnecessary hyperbole. Depth-encoding PS-OCT also “rigorously” measures diattenuation

We have removed the word “rigorously” according to the reviewer’s suggestion.

Page 3:

We also demonstrate that TRIPS-OCT outperforms depth-encoding PS-OCT in terms of ranging depth and laser jitter tolerance.

- Laser jitter model is unrealistic. See below. Suggest to remove this statement

We have removed the sentence “We also demonstrate that TRIPS-OCT outperforms depth-encoding PS-OCT in terms of ranging depth and laser jitter tolerance” as the reviewer suggested.

Page 3:

In this comparison, we ensured that the signal-to-noise ratios (SNRs) and the sampling time of the original signals used by the two methods were identical (Ext. Data Figs. 2a-c).

- But still the dual state is 1.5 times faster than the triple state measurement. Has this been taken into account in the signal-to-noise ratios (SNRs)? For a proper comparison, in dual state the sweep time of the laser needs to be 1.5 times slower than in triple state.

As discussed above, we respectfully disagree and maintain that the claim of a factor 2 in increased sensitivity is supported. Please see response to comment “Page 2”.

The standard deviation of the noise, or noise floor, of TRIPS-OCT was 48% lower than that of the conventional dual-input method (Fig. 1c, Ext. Data Figs. 2d, e), corresponding to a 2-fold improvement in the birefringence sensitivity.

- But the acquisition time was 1.5 times longer, so the net gain is $2/1.5 = 1.33$, not a factor of 2

As discussed above, we respectfully disagree and maintain that the claim of a factor 2 in increased sensitivity is supported. Please see response to comment “Page 2”.

The edge-artifact is a dominant source of birefringence noise that is induced by a shift of the point-spread-functions in the two eigen-polarization states of the preceding system components or tissue layers (Supplementary Discussion 1). Because this shift leads to apparent diattenuation, the resulting artifacts are markedly suppressed by correctly accounting for diattenuation in the Mueller matrices of the sample.

- The authors argue that a shift of the point-spread-functions occur due to the preceding system components or tissue layers. For the system components this would be Polarization Mode Dispersion, which they mitigate, and for the tissue layers the authors argued elsewhere that the preceding layers in the guinea pig are not birefringent.

More likely a polarization dependent scattering cross section combined with an overall jump in the scattering cross section of tissue structures plays a role in the edge-artifacts, which the authors did not model.

We agree with the reviewer that the “edge-artifacts” are a result of polarization mode dispersion (PMD), as the peak signals dispersed due to the PMD of preceding media. Spectral-binning indeed

mitigates the most prominent PMD-induced artifacts, but the peak shift persists within individual bins and, in combination with ‘structural jumps’, contributes to the observed birefringence noise, which we name as “edge-artifacts”. We have clarified this point in the revised Supplementary Discussion 1.

...However, the birefringence in the superficial layers of the sample (including the cornea and optical components in the instrument), induces polarization mode dispersion and results in a differential group delay between the two principal polarization modes. Therefore, a shift in the PFSs is created and leads to a relative amplitude difference along the ranging depth which does not truly reflect the polarization states but is an artifact.

In the guinea pig retina imaging, we did use spectral binning to mitigate the PMD of system optical components. However, the birefringence of the anterior segment of the eye, including the cornea, is large enough to induce edge-artifacts. To clarify this point, we revised the text in the manuscript as follows.

...The edge-artifact (**Supplementary Discussion 1**) is a dominant source of birefringence noise that is induced by a shift of the point-spread-functions due to the polarization mode dispersion of preceding tissue layers, including the cornea....

We agree with the reviewer that the “polarization dependent scattering” may be a source of birefringence noise in our context. However, as discussed in the last round of revisions, the birefringence noise observed in dual-input PS-OCT is ‘structured’ and aligns well with the proposed noise model. Here, in our experiments, we have demonstrated clear evidence of the existence of edge-artifacts induced by polarization mode dispersion. Although polarization dependent scattering is properly modelled as sample diattenuation and does not induce artifacts in TRIPS imaging, the magnitude of polarization dependent scattering is unclear and needs further investigation. To be accurate in the statement, we revised the text as follows.

...TRIPS-OCT isolates the sample retardance from the Mueller matrices, properly separating the effects of polarization dependent scattering, sample diattenuation, and apparent diattenuation, and thus is almost free from edge-artifacts (**Ext. Data Fig. 3c**)....

Correcting for retardance and diattenuation results in fiber orientations that closely follow the anticipated profile.

- Even triple state PS-OCT can only measure a relative angle, not an absolute angle, so the absolute angle is calibrated on henly ‘s fiber direction. Please state this more clearly.

We have revised the text to state that the relative angle is measured, as follows.

...An offset existed in the measured optic axis orientation due to the unmeasurable circular birefringence of the system and the anterior segment of the eye. The offset can be estimated by minimizing the difference between the measured optic axis orientation and the assumed orientation of the HFL, i.e., radial around the fovea. Without corneal correction, the measured fiber orientation deviates significantly from the assumed radial profile. After applying the correction for corneal retardance and diattenuation, the mean error of the measurement, characterized by the residual difference between the measured optic axis orientation and the assumed orientation, was 35% lower than that of the uncorrected results (**Fig. 1f**).

Page 13

To enable clinical PSB measurements, we developed TRIPS-OCT and improved the sensitivity, accuracy, and robustness of birefringence imaging compared to traditional PS-OCT, thereby mitigating previous obstacles towards its clinical translation.

- Authors mean “compared to traditional dual input state PS-OCT”

We are convinced that TRIPS-OCT offers clear benefits compared to all current PS-OCT implementations, including single input state, dual-input state, and depth-encoded PS-OCT. Our experiments offered a direct comparison to dual-input PS-OCT. While depth-encoded PS-OCT may offer comparable diattenuation modelling, its imaging range limitations, and the need for k-clocking, especially over an extended range adds complexity. In addition, considering that previous systems have undergone clinical translation, we have revised our statement as follows.

...To enable clinical PSB measurements, we developed TRIPS-OCT which offers benefits for birefringence imaging in terms of sensitivity, accuracy, robustness, and imaging range, compared to previous PS-OCT implementations.

Page 14:

Depth-encoding PS-OCT offers reliable depth-resolved measurements independent of the sample but compromises the optimal sensitivity window along the imaging range and requires autocorrelation calibration when the images encoded at the two depths drift in position or amplitude due to environmental variation.

- Depth-encoding PS-OCT does not compromise the optimal sensitivity window. Previously the authors stated that Depth-encoding PS-OCT compromises the optical ranging window. Depth-encoding PS-OCT does not compromise the optical ranging window, but utilizes the available ranging window to multiplex polarization input states. On the other hand, TRIPS-OCT compromises speed (3 times slower than Depth-encoding PS-OCT).

Depth-encoding PS-OCT is an important implementation of PS-OCT and contributes to the advancement of birefringence imaging technology. But we also think it has several drawbacks: 1) The ranging depth is reduced by a factor of 2 at identical sampling rate. 2) The image encoded in the deeper channel suffers from a sensitivity loss (critical for sclera imaging), due to the intrinsic signal roll-off (3.6 dB at 2.1 mm as reported by B. Braaf et al.). Additionally, 3) depth-encoding systems need an encoding optical unit and a k-clock device.

B. Braaf, K. A. Vermeer, M. de Groot, K. V. Vienola, and J. F. de Boer, "Fiber-based polarization-sensitive OCT of the human retina with correction of system polarization distortions," Biomed. Opt. Express 5(8), 2736 (2014).

Although the typical imaging range may be sufficient for accommodating depth-encoding in fully cooperative volunteers, we have found this to be a challenge in patients, especially with pathological eyes. Increased imaging range can be achieved by increasing the acquisition rate, but this results in an increased noise-floor. Furthermore, extending the imaging range with laser sources that come with a fixed k-clock required electronic clock doubling, adding further system complexity.

We have revised the text to clearly state the drawbacks of depth-encoding systems.

...Depth-encoding PS-OCT offers reliable depth-resolved measurements independent of the sample but compromises the optimal sensitivity window because the image encoded in the deeper channel suffers from a sensitivity loss (usually around 3 dB which depends on the encoding depth, digitization rate and laser linewidth) due to the sensitivity roll-off....

We have clarified that speed is not a major issue in modern OCT devices, because repetitive scans are routinely averaged to improve the image quality.

The dual-input EOM-based PS-OCT is robust to environmental variation, but it assumes measurements to represent pure retardance, hence, is readily impacted by the presence of diattenuation and edge-artifacts induced by both the system components and the sample. TRIPS-OCT does not have the aforementioned limitations but requires a longer acquisition time because of

triple repetitive scans at the same location of the sample.

- Drift of the polarization state in the fibers before the polarizer in the EOM will also lead to environmental variations, which will require (auto) calibration.

Drift in the polarization state before the polarizer and the EOM simply results in amplitude modulation and impacts the intensity SNR but has no direct impact on the PS signal. Dual-input PS-OCT and TRIPS, based on Stokes and Mueller formalism, do not require phase stability and there is no need for calibration. In comparison, depth-encoding PS-OCT requires phase stability to reconstruct the depth-encoded Jones matrix. This can be achieved with the help of a calibration signal to track the variation in the phase offset between the depth-encoded signals. We have further clarified this point in the revised Discussion.

...In addition, drifts of the encoding depth due to environmental variation as well as timing jitter may lead to a relative phase offset between the pathlength-offset signals in depth-encoding PS-OCT, requiring k-clocking and autocorrelation calibration to ensure correct reconstruction of Jones matrices....

Page 14:

TRIPS-OCT is not sensitive to sample motion.

- Axial sample motion on the order of tens of microns between three sequentially acquired depth profiles will affect TRIPS-OCT. Autocalibration methods need to be implemented to realign in depth the three sequentially acquired depth profiles to have the Stokes vectors of corresponding tissue structures overlap.

To be accurate, we revised the text as follows.

...TRIPS-OCT is not sensitive to sample motion within a few micrometers....

Page 14:

Overall, TRIPS-OCT mitigates some drawbacks of traditional PS-OCT implementations and makes it more suitable for clinical translation.

- Authors mean traditional dual state PS-OCT implementations

As discussed above, depth-encoding PS-OCT is an important implementation of PS-OCT and contributes to the advancement of birefringence imaging technology. But we also think it has several drawbacks: 1) The ranging depth is reduced by a factor of 2 at identical sampling rate. 2) The image encoded in the deeper channel suffers from a sensitivity loss (critical for sclera imaging), due to the intrinsic signal roll-off (3.6 dB at 2.1 mm as reported by B. Braff et al.). Additionally, 3) depth-encoding systems need an encoding optical unit and a k-clock device.

Page 18 figure 3:

e, Numerical model of the dependency of birefringence noise and laser jitter. Laser jitter is modelled by a random shift of the simulated fringes in the fringe domain. The random shift, modelled by a Gaussian distribution, induces a relative phase variation between the images encoded in shallow and deep channels of a depth-encoding PS-OCT system.

- The model the authors use for laser jitter consists of a catastrophic breakdown of the depth encoded method, i.e., significant shifts in the acquired data (several missed k-clocks at the start). This is an unrealistic model. This comparison should be removed from the paper.

According to the report "Boy Braaf, Koenraad A. Vermeer, Victor Arni D.P. Sicam, Elsbeth van Zeeburg, Jan C. van Meurs, and Johannes F. de Boer, "Phase-stabilized optical frequency domain imaging at 1- μm for the measurement of blood flow in the human choroid," Opt. Express 19, 20886-

20903 (2011)”, a proper k-clock would minimize the laser jitter and stabilize the phase. However, the paper also reports that if there is no k-clock, the phase standard deviation is around 0.1 rad (6°) at a depth of 1 mm.

In another report “B. Baumann, B. Potsaid, M. F. Kraus, J. J. Liu, D. Huang, J. Hornegger, A. E. Cable, J. S. Duker, and J. G. Fujimoto, “Total retinal blood flow measurement with ultrahigh speed swept source/Fourier domain OCT,” *Biomed. Opt. Express* 2(6), 1539–1552 (2011)”, the phase standard deviation is reported to be 97 mrad (5.6°).

Our simulation assumes phase standard deviation is in the range of 1°-5°, which is a good model of laser phase noise under the condition of no k-clock is used. We regard “no need for a k-clock device” as an advantage of TRIPS over depth-encoding system as TRIPS do not require phase stability for birefringence reconstruction.

To clarify this point, we have clearly labelled in Ext. Data Fig. 3e that the model is a “jitter model without a k-clock device”.

To emphasis this point, we have revised the text of the caption of Ext. Data Fig. 3e.

...The random shift, modelled by a Gaussian distribution with standard deviation of 1°-5°, induces a relative phase variation, ...Of note, the simulation assumes no k-clock signal is used. An additional k-clock device and proper post-processing can minimize the jitter noise and keep the induced phase standard deviation below 0.3°⁶⁹

69. Braaf, B. *et al.* Phase-stabilized optical frequency domain imaging at 1- μ m for the measurement of blood flow in the human choroid. *Opt. Express* 19, 20886 (2011).

We have clarified that no k-clock is used in our TRIPS-OCT in the revised Method section.

...In a swept cycle, the digitization was triggered by the laser trigger signal with a constant sampling rate of 1 GHz,...

Page 34:

These edge-artifacts cannot be removed by kernel averaging as they are associated with intensity variation, which is the information carrier of the sample structure.

- The artifact manifests itself on a depth scale of 10 micron (sup fig 1b), from 172 to 182 micron in depth. This creates a wiggle in the stokes vector evolution (fig 1c) over a depth trace of 10 micron in depth. However the depth resolution is 30 micron, as stated earlier. How come the full width at half max is about 6 micron in this figure?

In our simulation of edge artifacts, spectral binning is not performed, and thus the resolution is not reduced from 6 to 30 microns. This simulation is an illustration of edge-artifacts induced by the

sample, so we assume no PMD exists in the system. To clarify this point, we have added one sentence in the Supplementary Discussion 1 as follows.

We assume that the imaging system does not have polarization mode dispersion; therefore, spectral binning is not performed in the simulation.

Reviewer #3 (Remarks to the Author):

All comments have satisfactorily been addressed.

We thank the reviewer for the supportive comments.

Reviewer #4 (Report for the authors (Required)):

The authors have addressed all of my comments very well.

We thank the reviewer for the supportive comments.

Rebuttal 3

Reviewer #2 (Remarks to the Author):

Reviewer #2 (Report for the authors (Required)):

Review of Myopia biomarker revealed by triple-input polarization sensitive imaging of the posterior sclera by Xinyu Liu et al.

The manuscript has improved considerably, but still contains a few inaccurate statements. To this reviewer, the arguments about superiority of triple input state PS-OCT over all other PS-OCT implementations are not convincing and distract from the main message of the paper, namely the myopia results and the suppression of edge artifacts by triple input state versus dual input state PS-OCT.

We thank the reviewer again for the valuable comments which helped us to improve our manuscript. To address the reviewer's comments, we have revised our manuscript. Please evaluate our revision in the following point-by-point response. In this rebuttal letter, our response is highlighted in orange and the amendment in the manuscript is highlighted in blue. Key revisions are underlined.

A) The noise analysis comparing dual input OCT with Triple input OCT is still incorrect.

A triple input state measurement with the system of the authors takes 1.5 times longer than a dual input OCT measurement.

A measurement that takes 1.5 times longer should have a SNR improvement by a factor of 1.5. Similarly, a measurement in the same amount of time with 1.5 times more sample arm power should have a SNR improvement by a factor of 1.5.

To address this difference in the acquisition time of the two methods, the authors propose to average three (3) dual state input OCT images and two (2) triple input state OCT images. The authors assume that through this averaging process the SNR of the dual state image is improved by a factor of 3 and the triple input state image is improved by a factor of 2. Thus, according to the authors, the birefringence results calculated after this averaging method can be compared, since the time difference for the two acquisition methods has been compensated for.

However, in the averaging method chosen by the authors, the SNR of the images does not improve by a factor of 3 by averaging three dual state input images, but by a factor of square root of 3, nor does the SNR improve by a factor of 2 by averaging two triple state images, but by a factor of square root of 2.

In order to achieve the SNR improvement of a factor of 3 for the dual state images, the fringe data of three images need to be averaged (known as coherent averaging, averaging the complex depth profiles before taking the square to obtain an intensity image).

The same argument holds for the triple state images.

It is clear to this reviewer that triple input state OCT reduces the edge artifacts as compared to dual input state, and thus is an important improvement. However, the presented quantification in terms of birefringence sensitivity improvement by a factor of 2 is incorrect. Based on the presented analysis, the improvement is a factor of 1.63

We understand the reviewer's calculation that the SNRs are different under coherent averaging and incoherent averaging, and we agree that this issue was not clarified in the manuscript. To be accurate, we have removed the statement regarding the SNRs being identical, and instead explain that we use identical overall imaging time to compare dual and triple input state signals. We

understand that the factor of improvement may be lower than 2 when edge artifacts is not dominating the birefringence noise and thus SNR is another noise contributor (in the range of average SNR < 15 dB). As such, we have also removed the statement of 2-fold sensitivity improvement according to the reviewer's suggestion.

In Introduction

Original: We demonstrate that TRIPS-OCT provides a 2-fold improvement in birefringence sensitivity and an improvement in accuracy of optic axis measurement compared to dual-input PS-OCT.

Revised: We demonstrate that TRIPS-OCT improves birefringence sensitivity and accuracy of optic axis measurement compared to dual-input PS-OCT.

In Results

Original: In this comparison, we ensured that the signal-to-noise ratios (SNRs) and the sampling time of the original signals used by the two methods were identical (**Ext. Data Figs. 2a-c**)....The standard deviation of the noise, or noise floor, of TRIPS-OCT was 48% lower than that of the conventional dual-input method (**Fig. 1c, Ext. Data Figs. 2d, e**), corresponding to a 2-fold improvement in the birefringence sensitivity.

Revised: In this comparison, we ensured that the sampling time of the signals used by the two methods were identical (**Ext. Data Fig. 2a**)....The standard deviation of the noise, or noise floor, of TRIPS-OCT was 48% lower than that of the conventional dual-input method (**Fig. 1c, Ext. Data Figs. 2b, c**).

In Caption of Ext. Data Fig. 2

Added: ...scans modulated by the same polarization state were averaged before birefringence reconstruction. The averaging was performed on the intensity images without considering the phase....

As a minor point, figure Ext Data 2 c also indicates that something is wrong. To the understanding of the reviewer, figure b ("Averaged intensity B-scans used for comparing the two methods") shows 2 and 3 fold averaging for triple and dual input state images, respectively. The authors argue that figure c (based on the images in b) proves that the SNR of the images are the same. However, the SNR of the 3 times averaged dual input state image should be 1.5 times better than the SNR of the 2 times averaged triple input state image according to the reasoning of the authors to compensate for the difference in required acquisition time of the two methods for a fair comparison.

We have removed Ext. Data Figs. 2b, c and revised the caption of this figure.

New Ext. Data Fig. 2:

In Caption

Original: Ext. Data Fig. 2 | Signal-to-noise ratio (SNR) analysis of image data for comparison between dual-input and triple-input PS-OCT (TRIPS-OCT) methods.... b, Averaged intensity B-scans used for comparing the two methods. Upper, dual input, lower, triple input. c, Pixel-based correlation analysis of intensity signal-to-noise ratio values. Background noise was calculated from the regions in white boxes in b. This analysis confirms that the SNRs of the original signals used by the dual-input and triple-input methods for the birefringence reconstruction are identical.

Revised: Ext. Data Fig. 2 | Comparison between dual-input and triple-input PS-OCT (TRIPS-OCT) methods under the condition of the same acquisition time....The averaging was performed on the intensity images without considering the phase. This averaging process confirms that the acquisition time of the data used by the dual-input and triple-input methods for the birefringence reconstruction are identical.

B) For the comparison with Depth Encoded PS-OCT the authors do not choose the best phase jitter performance reported in the literature (state of the art) but choose to restrict the comparison to a jitter model without a k-clock device. This results in a restricted comparison. For a previously reported phase jitter of 0.3 degrees (as acknowledged by the authors in the figure caption), both methods perform equally well. In addition, the manuscript does not provide enough information of the simulation to be able to reproduce the results of Ext data figure 3e. It is recommended to remove Ext data figure 3e

We understand the concern of the reviewer that the statement of depth-encoding PS-OCT is not fair in terms of the state of the art. To address this comment, we have removed Ext. Data Figs. 3e and removed the statement about laser jitter in the caption, according to the reviewer's suggestion.

It is recommended to remove the following statement on line 615:

Laser jitter affects the Jones matrix-based method because the relative phase between the two columns of the Jones matrix is used in the reconstruction.

We have removed this statement as the reviewer suggested.

The depth range of swept sources has significantly improved over the last 10 years. Depth ranges of 12 millimeter with negligible sensitivity roll-off are currently available (e.g. Thorlabs SL100060).

It is recommended to modify the following statement on line 510 and further:

Depth-encoding PS-OCT offers reliable depth resolved measurements independent of the sample but compromises the optimal sensitivity window because the image encoded in the deeper channel suffers from a sensitivity loss (usually around 3 dB which depends on the encoding depth, digitization rate and laser linewidth) due to the sensitivity roll-off.

We have revised this statement as follows.

Original: Depth-encoding PS-OCT offers reliable depth-resolved measurements independent of the sample but compromises the optimal sensitivity window because the image encoded in the deeper channel suffers from a sensitivity loss (usually around 3 dB which depends on the encoding depth, digitization rate and laser linewidth) due to the sensitivity roll-off. In addition, drifts of the encoding depth due to environmental variation as well as laser jitter may lead to a relative phase offset between the pathlength-offset signals in depth-encoding PS-OCT, requiring k-clocking and autocorrelation calibration to ensure correct reconstruction of Jones matrices. The dual-input EOM-based PS-OCT is robust to environmental variation,...

Revised: Depth-encoding PS-OCT offers reliable depth-resolved measurements independent of the sample but requires a doubled ranging depth to achieve the same imaging depth as a time multiplexing system and a k-clock device to ensure phase stability⁶⁹. The dual-input EOM-based PS-OCT is robust to environmental variation as it does not require phase stability for birefringence reconstruction,...

It is unclear to this reviewer why the authors are so adamant about stating that triple input OCT is superior to depth encoded PS-OCT. The authors do not provide convincing arguments and have to resort to restricted implementations (a jitter model without a k-clock device) and out of date depth range capabilities to make this point. These statements should be removed from the manuscript. The authors show that triple state input OCT is superior to dual state input OCT. This is a nice result.

We agree with the reviewer and have removed the statements of comparison between TRIPS and depth-encoding PS-OCT. Considering the remarkable achievements of depth-encoding PS-OCT has made in this field, we must acknowledge it in our manuscript. The reason we develop TRIPS-OCT is that depth-encoding PS-OCT requires a doubled imaging range. We agree that this issue can be overcome by lasers with narrower linewidth and faster digitizers as pointed out by the reviewer. Again, we thank the reviewer for these valuable comments, which significantly helped us improve our manuscript.